# BENCHMARKING OPTIMIZERS
# FOR LARGE LANGUAGE MODEL PRETRAINING

https://anonymous.4open.science/r/llm-opt-bench-7B81

## ABSTRACT

The recent development of Large Language Models (LLMs) has been accompanied by an effervescence of novel ideas and methods to better optimize the loss of deep learning models. Claims from those methods are myriad: from faster convergence to removing reliance on certain hyperparameters. However, the diverse experimental protocols used to validate these claims make direct comparisons between methods challenging. This study presents a comprehensive evaluation of recent optimization techniques across standardized LLM pretraining scenarios, systematically varying model size, batch size, and training duration. Through careful tuning of each method, we provide guidance to practitioners on which optimizer is best suited for each scenario. For researchers, our work highlights promising directions for future optimization research. Finally, by releasing our code and making all experiments fully reproducible, we hope our efforts can help the development and rigorous benchmarking of future methods.

## 1 INTRODUCTION

Over the past five years, Large Language Models (LLMs) (DeepSeek-AI, 2024b; OpenAI, 2024; Gemini Team, 2024; Llama Team, 2024) have shown growth in performance and size, demonstrating proficiency in various downstream tasks (Snell et al., 2024; Brown et al., 2020; Wei et al., 2023). The success of LLM pretraining hinges on three key pillars: high-quality data (Penedo et al., 2024b; Li et al., 2024), architectural innovations (Jiang et al., 2024; DeepSeek-AI, 2024b), and scalable optimization techniques (Jaghouar et al., 2024; Shah et al., 2024; Charles et al., 2025).

Among these, the choice of optimizer has remained notably consistent in recent years, with `Adam(W)` (Kingma & Ba, 2017; Loshchilov & Hutter, 2019) dominating deep learning for nearly a decade. However, recent advances (Jordan et al., 2024b; Liu et al., 2025; Vyas et al., 2024; Pagliardini et al., 2024; Pethick et al., 2025; Frans et al., 2025; Defazio et al., 2024b) challenge this status quo, offering alternatives that surpass `AdamW` in speed, communication efficiency (Ahn & Xu, 2025) or final downstream performance on various benchmarks (Dahl et al., 2023; Karpathy, 2022), particularly for autoregressive language modeling (Radford & Narasimhan, 2018). Despite these innovations, current benchmarks and ablation studies (Zhao et al., 2025; Morwani et al., 2025; Kaddour et al., 2023) remain narrow in scope, often examining only isolated aspects of optimizer design (Kasimbeg et al., 2025). This lack of systematic comparison makes it difficult to obtain trustworthy insights for practitioners or identify the next promising research directions.

In this work, our goal is to revisit the problem of benchmarking optimizers for LLM pretraining. We do so through standardized experiments which vary important parameters such as batch size, model size, and the number of training iterations. This allows us to formulate an up-to-date list of best-performing methods for the community of researchers and practitioners. We demonstrate the efficiency of each considered method through careful tuning, and present insightful ablations along the way. Furthermore, we provide a set of best practices for LLM pretraining that are applicable regardless of the optimizer chosen.

We summarize our contributions as follows:

(**Contribution 1**) We conduct the first large-scale, controlled benchmark of 11 different optimization methods across diverse LLM training scenarios. A fair comparison is ensured by precise accounting

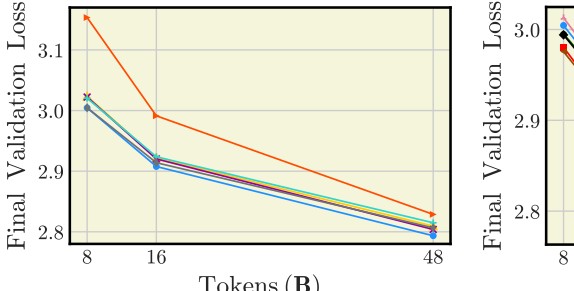 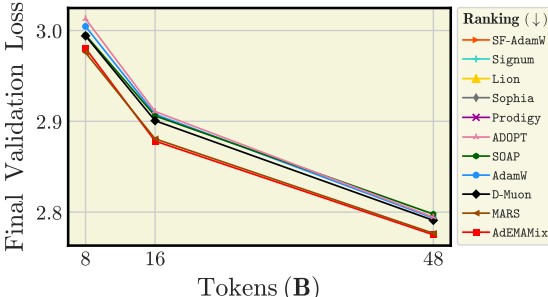

Figure 1: **Ranking of optimizers for 720M Llama-based models.** We plot the *final validation loss* obtained by the best-tuned optimizers on the FineWeb dataset. We use a batch size of $1\text{M}$ tokens and train multiple methods beyond and below the Chinchilla optimal duration, which is $14.4\text{B}$ for model of this size. `AdEMAMix` and `MARS` are the best optimizers in this setup, with a noticable gap in performance compared to other methods. We also plot the `AdamW` baseline in both figures to distinguish the group of methods that consistently perform worse than `AdamW` from the group of optimizers that outperform it for some training durations. See § 3 and Appendix G for a detailed description of our experimental setup, including hyperparameters.

for compute costs, and extensive hyperparameter tuning. We identify optimal optimizer choices in several relevant training regimes, for both dense and Mixture of Experts (MoE) architectures.

(**Contribution 2**) We perform comprehensive ablations of critical training hyperparameters—including warmup duration, initialization schemes, gradient clipping, final learning rates, and learning rate scheduler choices—providing actionable insights for optimizing LLM training in practice.

(**Contribution 3**) We open-source our full benchmarking toolkit, including training scripts, evaluation pipelines, and hyperparameter configurations, to enable reproducible research and facilitate future optimizer development.

**For practitioners**, our work provides an evidence-based answer to the burning question: "*Is* `Adam` *still the most effective optimizer in the age of LLMs, or can we achieve better performance at scale with novel optimizers?*".

**For researchers**, our work delivers a unified benchmarking framework for LLM pretraining, along with extensive ablation studies which systematically evaluate both popular and overlooked optimizer designs—revealing previously unexplored tradeoffs between efficiency, stability, and final model performance. Overall, our findings not only challenge long-held assumptions about optimizer selection but also establish a foundation for future advances in large-scale model training. By bridging the gap between theoretical innovation and practical deployment, this work aims to accelerate progress in both research and industry applications of LLM training.

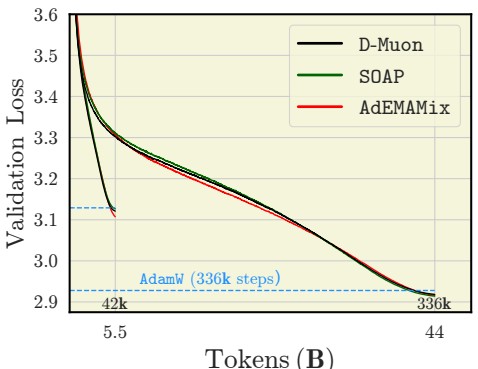

Figure 2: **Training dynamics of leading optimizers on 520M MoE model pretraining.** We use a batch size of $131\text{k}$ tokens, and train models for both short runs, i.e., less than Chinchilla optimal duration, and for extended runs beyond this regime. The dashed blue lines correspond to the final validation loss of `AdamW` baselines trained for both $42\text{k}$ and $336\text{k}$ steps.

## 2 BACKGROUND & RELATED WORK

**Optimizers.** While computer vision models often show comparable performance between `SGD` (Robbins & Monro, 1951) and `AdamW` (Zhang et al., 2020b), the landscape differs dramatically in LLM training (Srećković et al., 2025). Recent work (Zhang et al., 2024b) demonstrates that adaptive methods like `AdamW` provide substantially better optimization characteristics for transformer-based language models. The question of why `AdamW` works so well has been a long-standing topic of research (Balles & Hennig, 2020; Orabona, 2020; Zhang et al., 2020a; Kunstner et al., 2024; Kunstner, 2024). Modern methods often inherit `AdamW`'s core ideas in their structure, such as `ADOPT` (Taniguchi et al., 2024) and `AdEMAMix` (Pagliardini et al., 2024). `ADOPT` has been motivated by solving long-standing convergence issues in `AdamW`. By normalizing the second-order moment prior to the momentum update, they eliminate the non-convergence issues of `AdamW` on

smooth non-convex functions. Meanwhile `AdEMAMix` extends `AdamW` with an additional slower momentum buffer, i.e., a slower exponential moving average (EMA), which allows the use of much larger momentum values, accelerating convergence. *One interpretation of `AdamW`'s effectiveness lies in its sign-based update* (Kunstner et al., 2023): without the exponential moving average (EMA), `AdamW` resembles `signSGD` (Bernstein et al., 2018). Recent works (Zhao et al., 2025; Karimireddy et al., 2019) has shown that `Signum` (`signSGD` with momentum), can perform comparably to `AdamW`. The community also discussed `Lion` (Chen et al., 2023), a method with a similar sign-based structure. `Signum` and `Lion` offer memory benefits due to the use of only a single instead of Adam's two buffers for optimizer states. *Another family of methods stems from `AdamW`'s approximate second-order structure.* This idea has given rise to `Sophia` (Liu et al., 2024), where the diagonal of the Fisher information matrix is used as the second moment estimate. Exploiting the matrix structure of model weights and optimizer states has led to methods such as `SOAP` (Vyas et al., 2024), `Muon` (Jordan et al., 2024b) and `Scion` (Pethick et al., 2025), including their extentions (Liu et al., 2025; Riabinin et al., 2025; Ahn & Xu, 2025). *The parameter-free concept* (Orabona & Pál, 2016) has led to the development of `Schedule-Free AdamW` (`SF-AdamW`) (Defazio et al., 2024b) and `Prodigy` (Mishchenko & Defazio, 2024). These optimizers do not require a decreasing learning rate schedule, making them relevant for continual training. Last but not least, `MARS` (Yuan et al., 2024), builds upon this line of research and incorporates a variance reduction mechanism in its update rule.

**Benchmarks.** To a large extent, the benchmarking setup determines the final conclusions. Some benchmarks are designed for short speedruns in terms of training or validation loss (Jordan et al., 2024a), while others focus on a downstream target metric after training (Zhao et al., 2025; Dahl et al., 2023; Schmidt et al., 2021). Methods that perform well in short speedruns might not be optimal for longer training horizons as in real LLM training runs (see Figure 3 **(a)**, or Figures 37 and 40 **(b)**). "*But what constitutes a sufficiently long horizon?*" "*What should be the compute budget for LLM training?*" These are questions explored by scaling laws (Kaplan et al., 2020). Early benchmarks for optimizers and other ablation studies often rely on Chinchilla scaling laws (Hoffmann et al., 2022) with a ratio of roughly 20 tokens per parameter needed for pretraining. However, recent research (Li et al., 2025a; Porian et al., 2024; Sardana et al., 2024) argues that this is far from sufficient for production-ready models. Another important issue is the choice of loss function. Recent setups have used an auxiliary $z$-loss (Yang et al., 2023; Chowdhery et al., 2022) in addition to cross-entropy, which requires further investigation. We believe that this choice is influenced by the use of the OLMo (Team OLMo, 2024b) codebase, which we also address in our work. Additionally, we found that previous setups for comparing optimizers do not align with recent best practices regarding weight decay, learning rate decay, and overall hyperparameter tuning. All of these questions are revisited in our work. We also encourage the reader to refer to the concurrent work of Wen et al. (2025). Overall, our findings are well aligned, we both find `Muon` and `SOAP` outperforming `AdamW` in many settings. A few differences: we find `MARS` beating `D-Muon` at $1\mathrm{M}$ tokens batch size, and also `AdEMAMix` is the best in the majority of our runs. We attribute differences regarding `MARS` to discrepancies in the batch size, Wen et al. (2025) uses a large sequence length of 4096. In addition, the authors do not try `AdEMAMix`, therefore such a comparison in their setup remains questionable.

## 3 EXPERIMENTAL SETUP

**Notations.** We use the following notations. Let $\gamma$ be the learning rate, $\lambda$ the weight decay coefficient, and $T$ the total number of iterations. Momentum-related parameters are represented by $\beta$.

**Optimizers.** Here is a list of the optimizers we considered in our work. For each algorithm, we write in parentheses the optimizer-specific hyperparameters we tuned: `AdamW`$(\beta_1, \beta_2)$, `ADOPT`$(\beta_1, \beta_2)$, `AdEMAMix`$(\beta_1, \beta_2, \beta_3, \alpha)$, `Lion`$(\beta_1, \beta_2)$, `Signum`$(\beta)$, `Muon`$(\gamma^{\mathrm{M}}, \beta, \beta_1, \beta_2)$, `D-Muon`$(\beta, \beta_1, \beta_2)$ (Liu et al., 2025), `SOAP`$(\beta_1, \beta_2)$ and preconditioning frequency, `Sophia`$(\rho, \beta_1, \beta_2)$, `SF-AdamW`$(\beta_1, \beta_2)$, `Prodigy`$(\beta_1, \beta_2)$, `MARS`$(\eta, \beta_1, \beta_2)$. When an optimizer has several momentum variants e.g. Nesterov (Nesterov, 1983) or Polyak (Polyak, 1964), we try both. When optimizers use the Newton-Schulz orthogonalization (Bernstein & Newhouse, 2024; Higham, 2008), we vary the number of steps for this procedure. In addition, we tune the learning rate $\gamma$ extensively for all methods. We also try different gradient clipping levels, warmup steps, weight decay values, weights initialization, and learning rate schedulers. A summary of the hyperparameters tested and selected for each model size is in Appendix G. All optimizers are described in depth in Appendix C.

**Models & Data.** For most experiments, we use a Llama-like transformer (Llama Team, 2024) architecture with weight tying (Press & Wolf, 2017), including SwiGLU activations (Shazeer, 2020),

RMSNorm (Zhang & Sennrich, 2019), and RoPE embeddings (Su et al., 2023). We experiment with four sizes of models: 124M, 210M, 583M, 720M. In addition to our dense models, we also benchmark optimizers on a Llama-based 520M MoE model, the corresponding setup is described in § 4.4 and Appendix G. We train on a 100B tokens[1] subset of FineWeb (Penedo et al., 2024a). It consists of a cleaned and deduplicated corpus for LLM pretraining, which we tokenize using the GPT-2 tokenizer prior to splitting into train and validation sequences.

**Iterations & Batch size.** Throughout our experiments, we use a sequence length of 512 tokens. For clarity, we often report the batch size in tokens by writing *batch size × sequence length*. For the 124M model, we use batch sizes of $32 \times 512 = 16k$, $256 \times 512 = 131k$, and $512 \times 512 = 262k$ tokens; for the 210M model and 520M MoE model, we use a batch size of $256 \times 512 = 131k$; for the 583M model, we leverage the batch size of $3936 \times 512 = 2M$ tokens, finally, we use a batch size of $1984 \times 512 = 1M$ tokens for the 720M model. Depending on the model size, we vary the number of iterations—also measured in tokens for compatibility with scaling laws and to accommodate different batch size settings. We train 124M and 210M models for equal durations of $\{1, 2.1, 4.2, 6.3, 8.4, 16.8\}B$ tokens. This corresponds to $T \in \{64, 128, 256, 384, 512, 1024\}k$ iterations for a batch size of 32, and $T \in \{8, 16, 32, 48, 64, 128\}k$ iterations for a batch size of 256. For 583M models, we train on 13B tokens, corresponding to 6.5k iterations. In the setup with 720M model, we have $T \in \{8, 16, 48\}k$ iterations for a batch size of 1M tokens. Thus, for all model scales, we include both Chinchilla optimal lengths of training and beyond. More details are available in Appendix E.

**Loss.** We train using the classical cross-entropy next token prediction loss. Some prior works introducing optimizers (Vyas et al., 2024), benchmarkings (Zhao et al., 2025), or pretraining recipes for LLMs (Jaghouar et al., 2024; Chowdhery et al., 2022; Yang et al., 2023; Brandfonbrener et al., 2024), use a $z$-loss regularizer in addition to cross-entropy. We found that this has little impact and, therefore, do not use $z$-loss. An ablation showing results with and without $z$-loss is in § 4.

**Hyperparameter Tuning.** Training LLMs is a computationally intensive task (Erdil, 2024). As a guidance, practitioners often rely on insights gathered at lower scales, scaling laws (OpenAI, 2024; DeepSeek-AI, 2024a; Sardana et al., 2024; Li et al., 2025b), and other rules (Yang et al., 2022; Dey et al., 2024; Blake et al., 2025; Kumar et al., 2024). It is also commonplace to run experiments for only a shorter duration of training, as a way to test certain hyperparameters prior to extending the training horizon to more iterations. Because a full grid search over every hyperparameter, for each setting and optimizer, would be too costly, we resort to a similar approach. More precisely, for each model size, batch size, and optimizer, we extensively tune optimization hyperparameters for a number of training tokens which are near-Chinchilla optimal, e.g., we pick $\{2.1, 16\}B$ tokens for tuning $\{124, 720\}M$ models (see Appendix G). We then keep those hyperparameters when we increase the number of iterations. While we found that the sensitivity to several hyperparameters can change as we increase the training horizon—see Figure 13—we found this approach simple and yet effective. The hyperparameters being considered depend on the optimizer. We proceeded from small to large model scale, and used insights gathered at smaller scales to guide the hyperparameter search at larger scales. Our hyperparameter sweeps are summarized in Appendix G. We present the clarifications regarding the connection between the number of iterations and tokens for different batch size settings, as well as the Chinchilla optimal training durations for our models in Tables 3, 4, 5, 6, and 48. As learning rate schedulers, we compare cosine (Loshchilov & Hutter, 2017), linear and warmup-stable-decay (WSD) (Hu et al., 2024; Zhai et al., 2022; Hägele et al., 2024). Unless specified, we use a cosine scheduler. Results with WSD and linear schedulers are discussed in § 4. Recent works also emphasize the importance of sufficiently decaying the learning rate (Bergsma et al., 2025; Schaipp et al., 2025; Hägele et al., 2024; Li et al., 2025b; DeepSeek-AI, 2024b). As such, we take care to decay to $0.01 \times \gamma_{max}$ instead of the often used $0.1 \times \gamma_{max}$ (Hoffmann et al., 2022; Touvron et al., 2023; Biderman et al., 2023; Workshop, 2023; Team OLMo, 2024b;a; Zhao et al., 2025). To give an idea of how much effort was put into tuning each method, across all model sizes, batches and iterations, we trained a total of 2900 models, and have spent roughly 30000 GPU hours. See more details in Appendices D and G.

## 4 RESULTS

We structure our story starting with smaller models and batch sizes, and gradually scaling up to larger configurations. In some instances, we complement the core benchmarking results with additional ablations and possible best-practices.

---

[1] https://huggingface.co/datasets/HuggingFaceFW/fineweb

## 4.1 Benchmarking & Ablations at Small Scale: Training 124M Models

**Results with "small" batches.** We first report results when using batches of $32 \times 512$ tokens in Figures 3 **(a)** and 12 **(a)**. We tune the hyperparameters by training for 2.1**B** tokens (128**k** iterations) and then keep those hyperparameters for all other training durations. The best hyperparameters are reported in Appendix G.1. We observe how, for the smallest number of iterations we considered (1**B** tokens ≡ 64**k**), `SOAP`, `ADOPT`, `AdEMAMix`, `D-Muon`, `Prodigy`, and `SF-AdamW` all outperform `AdamW`, with `D-Muon` being the best. As we increase the number of iterations, `AdEMAMix` takes the lead while `AdamW` becomes a second, and closes the gap with `D-Muon` and `SOAP`. A sign-based methods such as `Lion` and `Signum` are expected to perform poorly when the batch size is small. Intuitively, this is due to the sign($\cdot$) operator being sensitive to gradient noise (Tomihari & Sato, 2025; Kornilov et al., 2025). As described in its original paper, `MARS` also performs poorly when the batch size is small. We found `Prodigy`, the basic `Muon` (see Figures 16 and 17 **(a)**) and `SF-AdamW` to underperform in this setting compared to `AdamW`. On this scale, `Prodigy` suffers from the lack of bias correction of the learning rate, as well as being sensitive to $(\beta_1, \beta_2)$ (see Figure 30). Importantly, when the batch size is sufficiently small, we observe that `Sophia` diverges when increasing the number of iterations, even if decreasing the learning rate (see Figure 27). Thus, we decided not to include `Sophia` at this stage of our benchmarking.

**Results with "large" batches.** We now report results when using batches of $256 \times 512$ tokens—$8\times$ larger than for our "small" batch setting. Results in Figures 3 **(b)** and 12 **(b)** show how `Signum`, `MARS`, `Lion`, `Prodigy` greatly benefit from the increased batch size. Remarkably, we observe that the `Prodigy` method scales similarly to `AdamW`. We emphasize the possible community interest in this algorithm, as its effective learning rate—determined by two EMA sequences—emulates the learning rate behavior of `AdamW`. When the scheduler is applied and $\gamma_{\max}$ of `Prodigy` is set to 1 (its default value), these EMAs result in the maximal effective learning rate, which closely matches that of `AdamW`—see Figure 35. For a small number of iterations (e.g. $T \in \{8\mathbf{k}, 16\mathbf{k}\}$ corresponding to 1**B** and 2**B** tokens), all methods outperform `AdamW` except for `SF-AdamW` and `Sophia`. As we increase the number of iterations `ADOPT`, `D-Muon`, `SOAP`, and `AdEMAMix` take the lead. In particular, `AdEMAMix` has a consistent lead over other methods. While we anticipated—in accordance with Vyas et al. (2024)—that `SOAP` would greatly benefit from the larger batch size, its behavior remains relatively consistent compared to our previous small batch setting.

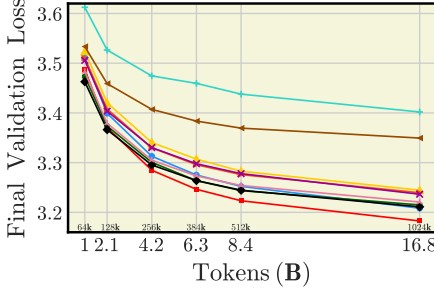

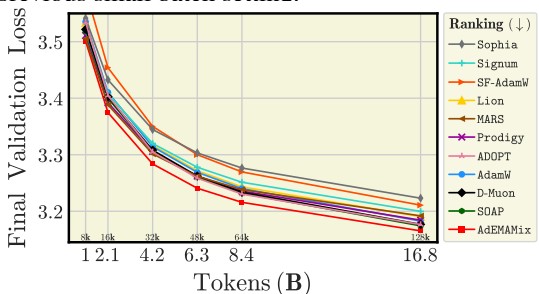

(a) Batch size $32 \times 512$ tokens.  (b) Batch size $256 \times 512$ tokens.

Figure 3: **Ranking of optimizers for 124M models with "small" and "large" batch sizes.** In both **(a)** and **(b)**, we show the *final validation loss* for different training durations, corresponding to different numbers of tokens. Above each token number, we write the number of training iterations corresponding. In **(a)**, we use a "small" batch size of $32 \times 512$ tokens. In **(b)**, we use a larger batch size of $256 \times 512$ tokens.

> **Takeaway 1.** After experimenting with both "small" and "large" batch settings, we conclude that: (I) `AdEMAMix` consistently achieves state-of-the-art performance and robust scaling with training duration; (II) sign-based methods (`Signum`, `Lion`), and `MARS` greatly benefit from the increased batch size; (III) `Sophia` diverges in the small-batch setting, when trained beyond the Chinchilla optimal horizon, even with sufficiently small learning rate; (IV) `SOAP` show a surprisingly consistent performance in both settings.

**Stability across training horizons.** As mentioned in § 3, we tune hyperparameters training on 2.1**B** tokens and keep those hyperparameters when extending the training horizon. However, when increasing the length of training or scaling batch size, critical hyperparameters of optimizers such as learning rate, betas might change (Busbridge et al., 2023). Thus, we additionally *re-tune the methods* for 16.8**B** length of training to show the best results. We found that previously widely

adopted (DeepSeek-AI, 2024b; Wortsman et al., 2023; Zhao et al., 2025; Jaghouar et al., 2024; Hägele et al., 2024; Li et al., 2025b) for `AdamW` ($\beta_1 = 0.9$, $\beta_2 = 0.95$) parameters give worse results than ($\beta_1 = 0.9$, $\beta_2 = 0.999$). We point that it would be beneficial to further increase the $\beta_2$ for `AdamW`-like optimizers when increasing the length of training. The same applies to $\beta_3$ parameter of `AdEMAMix`, which we increase from $0.999$ to $0.9999$ when training on $16.8$**B** tokens and beyond (see Appendix F.1 for a detailed ablation on that matter and references therein). Importantly, from Figure 12 **(b)**, we see that `SOAP` and `D-Muon` narrow the gap with `AdEMAMix`. It is interesting to see how the situation changes when the training horizon is extended to $33.6$**B** tokens ($\equiv 256$**k** iterations). For this experiment, we use the batch size of ($256 \times 512$), and keep the re-tuned hyperparameters we found for $16.8$**B** tokens run, simply reusing them for longer training. We report insights gathered from this ablation in Figure 4 (*right*). As in the "small" batch ablation, we emphasize that `Sophia` exhibits convergence issues when extending the training run, and diverges shortly after $130$**k** steps (Figure 26). Regarding other optimizers, we observe a consistent behavior compared to the one from Figure 3 **(b)**—all methods remain at the same position in our tier-list. The results suggest that the best hyperparameters found at $16.8$**B** scale are also consistent w.r.t. doubling the number of steps. "But what can one say about scaling batch size while keeping the same amount of tokens seen?"

**Increasing the batch size further.** We also run an experiment with batches of $512 \times 512 = 262$**k** tokens, training for $64$**k** iterations, thus, we keep the total amount of tokens to train on. We show the results of this ablation in Figure 4 (*left*). Noticeably `MARS` becomes the second best-performing method behind `AdEMAMix`, followed closely by `Prodigy`, `Lion`, `ADOPT`, and `SOAP`. Interestingly, `Signum` performs comparably to `AdamW`. Our results with batches of $\{131, 262\}$**k** tokens show an evidence that sign-based methods greatly benefit from increased batch size, as noticed in many prior works (Chen et al., 2023). Furthermore, the hyperparameter sweeps from (Zhao et al., 2025; Zhang et al., 2024a) suggest that `Lion`, `Signum`, `AdamW` stay consistent w.r.t tuning all hyperparameters except for batch size, where they notice a worsens in performance at large batch sizes above ours $256 \times 512$, while we observe a quite opposite results in our setup.

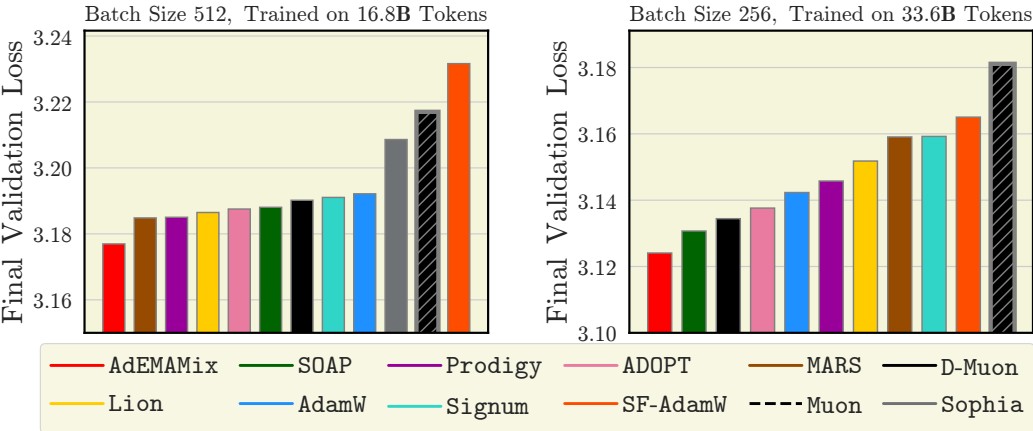

Figure 4: **Scaling batch size vs. scaling the number of iterations.** Our results demonstrate that: (*left*) scaling the batch size significantly improves `MARS`, `Signum`, `Lion` and `Prodigy` making them as good as `AdamW` even for a long training for $16.8$**B** tokens. Which was not the case in Figure 3 **(b)**, where we still observed a significant gap in performance; and (*right*): indeed, with scaling of the number of iterations, the gap between `SOAP` and `AdEMAMix` narrow and, finally, increases. However, with increase of the `AdEMAMix` $\beta_3$, the performance gap with `SOAP` reappears.

> **Takeaway 2.** (I) Suprisingly, many methods, especially `MARS`, `Prodigy`, and sign-based ones, can outperform `AdamW` while trained on a sufficiently large batches. (II) We also found that in our setup, once optimizers are properly re-tuned for the maximal length of training considered, doubling of number of iterations does not affect the ranking of methods.

**Ablations at small scale: 124M models.** In this section, we present *ablations with relatively short descriptions* and the most meaningful figures to highlight them. We complement our results from this section in Appendix F.1. We start sequentially with smaller models ablating: **weight decay**, **learning rate sensitivity**, **warmup**, **learning rate schedulers**. For each ablation, we provide the corresponding takeaway—see Takeaway 10, 11, 12, 13. Details on hyperparameter searches are provided in Appendix G.

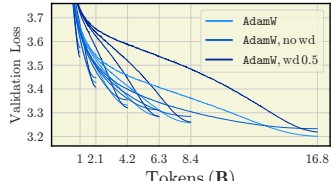 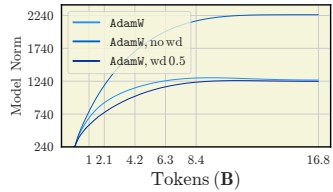

(a) Use large $\lambda$ for short training.    (b) Use $\lambda = 0.1$ for long training.    (c) Norm grows with smaller $\lambda$.

Figure 5: **Larger weight decay achieves significantly better results when training on fewer tokens.** In **(a)** we observe that runs of AdamW, Signum, and Lion with the large weight decay of $0.5$ consistently outperform the baseline AdamW with weight decay of $0.1$ for all training durations except for the last one. Notably, Signum and Lion with large weight decay perform even better than AdamW with the same learning rate. In **(b)**, we also consider a setting without weight decay. We observe that this is suboptimal not only for AdamW, but also for the majority of other optimizers (see Appendix F.1), while the typical weight decay of $0.1$ remains the best for longer training. In **(c)**, we ablate the impact od weight decay on the model's $\ell_2$ norm. See Figure 15 for detailed ablation.

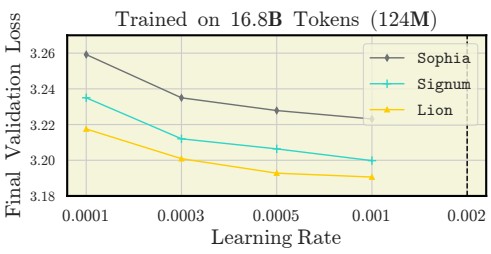 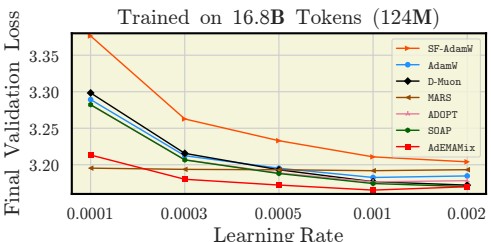

(a) Sign-based, and Sophia diverge with large $\gamma$.    (b) Parabolic shape for most optimizers.

Figure 6: **Optimal learning rate stability across optimizers.** The optimal learning rate determined during tuning on $2.1$B tokens remains consistent after a learning rate sweep on $16.8$B tokens for most optimizers. In **(a)**, we observe that sign-based methods and similar to them Sophia diverge with increasing learning rate. Interestingly, in **(b)**, SF-AdamW, SOAP, and D-Muon demonstrate their best performance with a large learning rate of $0.002$, while MARS maintains remarkably consistent performance across the learning rate sweep. See Figure 18 and Appendix F.1 for more details.

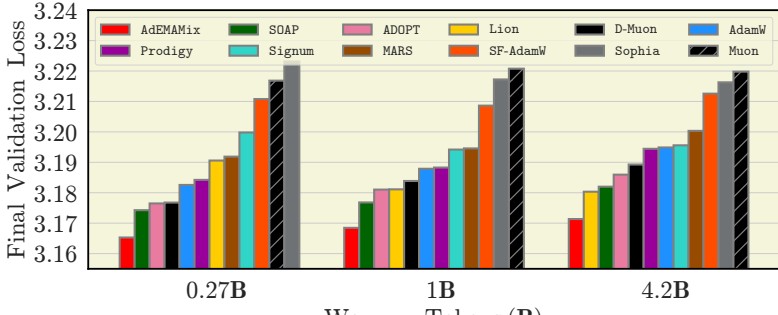

Figure 7: **Warmup ablation.** For $124$M model trained on the batches of $256 \times 512$ tokens, we perform a sweep over the linear warmup durations of $\{1.56\%, 6.25\%, 25\%\}$ of the length of training, which corresponds to $\{2, 8, 32\}$k steps, respectively. Clearly, sign-based optimizers, Sophia, and SF-AdamW benefit from the increased warmup.

## 4.2 Benchmarking & Ablations at Medium Scale: Training **210M** Models

We report the complete ablation and benchmarking of $210$M models in Section 4.2. Figure 36 demonstrated the ranking of optimizers in this setup, and Figure 37 depicts the training dynamics. We use the same $256 \times 512$ batch size for these models.

> **Takeaway 3.** We do not observe a much of a change in ranking of optimizers for $210$M model, compared to benchmarking on $124$M. At the same time, we replicated almost identical hyper-parameters for all optimizers, except for the learning rate for sign-base methods. We also point out that sign-based methods are more sensitive to the learning rate while scaling the model size. As that, we changed the peak learning rate from $10^{-3}$ to $5 \cdot 10^{-4}$ for Lion and Signum.

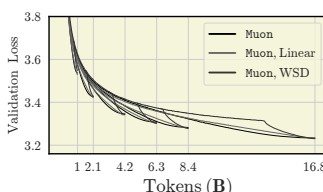 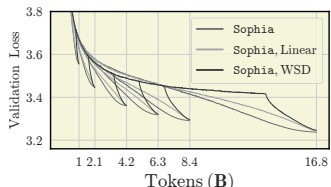 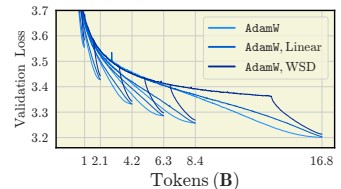

(a) `Muon` "prefers" WSD.     (b) No preference for `Sophia`.     (c) Cosine & `AdamW`.

Figure 8: **Comparisons between cosine, WSD, and the linear schedulers.** Notably, schedulers behave differently with respect to optimizer. In **(a)**, the `Muon` optimizer shows a preference for WSD across most training durations. `Sophia` exhibits an almost perfect match between all three schedulers. However, for `AdamW`, along with the majority of other optimizers studied (see Figure 20), we get a better performance with cosine. We also report a detailed comparison for all optimizers in Appendix F.1, and cover additional ablations on another dataset (see Figure 19). We also study their gradient norm patterns and report results in Figure 21, and in Figure 22.

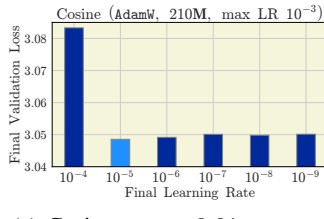 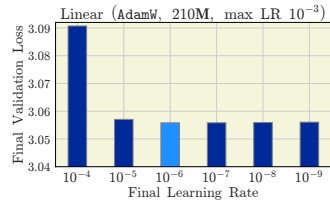 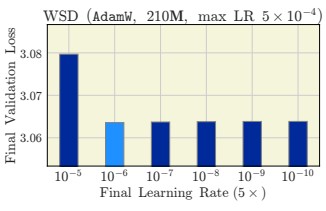

(a) Cosine, $\gamma_{end} = 0.01 \times \gamma_{max}$.    (b) Linear, $\gamma_{end} = 0.001 \times \gamma_{max}$.    (c) WSD, $\gamma_{end} = 0.01 \times \gamma_{max}$.

Figure 9: **Decaying the learning rate down to $0.01 \times \gamma_{max}$ and beyond, instead of only to 10%.** We run a 210M Llama model, and observe a common pattern for different schedulers that decreasing the learning rate to moderate $0.01 \times \gamma_{max}$ value is a better choice than decreasing it down to zero. Interestingly, the linear learning rate scheduler for models at a given scale, requires $0.001 \times \gamma_{max}$. See Figure 25 for corresponding ablation for 124M model.

> **Takeaway 4.** Decaying the learning rate further than 10% of the maximal significantly improves the results. However, for different schedulers, the best final learning rate is different.

### 4.3 SCALING UP: BENCHMARKING MODELS OF 583M AND 720M PARAMETERS

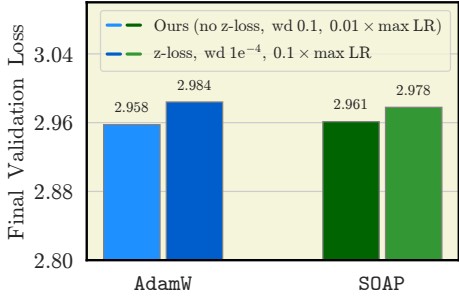

Figure 10: **Ablation of $z$-loss regularization.** Incorporating the $z$-loss regularizer does not improve the final loss or reduce the spikiness of the loss curves. Moreover, combining $z$-loss with small weight decay and decaying $\gamma$ down to 10%, further degrades overall performance. Notably, these changes can reverse the relative ranking of optimizers compared to the results reported by Vyas et al. (2024).

**Comparison between our setting and Vyas et al. (2024).** We pick two methods: `AdamW`, `SOAP`, and run experiments with a larger model of 583M parameters, and a large batch size of 2M tokens. The goal is to get closer to one of the settings described in (Vyas et al., 2024), i.e., train for the Chinchilla optimal amount of tokens and use the same batch size. Therefore, we train for 6500 iterations, corresponding to 13B tokens. We found several key differences between our codebase and (Team OLMo, 2024a), used by Vyas et al. (2024): (I) we decay the learning rate to $0.01 \times \gamma_{max}$ instead of $0.1 \times \gamma_{max}$, with $\gamma_{max}$ being the maximum learning rate, (II) we use typical weight decay values of e.g. 0.1 instead of smaller values such as 0.01 or 0.0001, (III) we do not use a $z$-loss in addition to ours. Our ablations in Figures 9 and 25 already confirm that properly decaying the learning rate has an important effect on optimization. Regarding $z$-loss and weight decay, we run an ablation to compare both settings and conclude that removing the $z$-loss and increasing the weight decay to 0.1 improves the results. We remind that hyper-

parameter choice in (Vyas et al., 2024) has been suggested by popular codebases for LLM pretraining (Team OLMo, 2024a;b; Biderman et al., 2023). In that view, we pose the following observation to practitioners.

> **Takeaway 5.** Hyperparameter choices commonly imposed by popular codebases—such as final learning rate, $z$-loss regularization, and weight decay—substantially affect both absolute performance and the relative ranking of optimizers at Chinchilla scale.

**Results on 720M parameter model & 1M batch size.** To expand our ablations towards more practical scales, we train a 720M parameter model with a batch size of 1M tokens. As previously, we include both the Chinchilla optimal horizon and beyond, following the setup in § 3. Our goal is to characterize how optimizer performance evolves with increased model size, batch size, and total tokens seen. The best optimizers are reported in Figure 1. In Appendix F.3 we report training dynamics for all optimizers—see Figure 40.

> **Takeaway 6.** (I) At larger scale of model and batch size, `AdEMAMix` and `MARS` dominate, by far outperforming others—see Figure 1. (II) Despite training with large batches, `Signum` and `Lion` scale poorly. (III) `D-Muon` is consistent across all our benchmarking setups.

### 4.4 EXTENSION TO MoEs

The goal of ablating optimizer performance on MoE models is to assess whether our previous benchmarking results transfer smoothly to a new type of architecture. To show this smooth transition, we utilize an old batch size setup and keep untuned all optimizer-related hyperparameters found for the corresponding dense model—simulating a situation as one would do in practice without much time for running dozens of experiments on new architectures.

**Setup & Comparison.** Besides training dense Llama-like transformers, we also cover a comparison for MoE architectures (Shazeer et al., 2017). Our variant of MoE is based on the Switch-Transformer implementation (Fedus et al., 2022). We use a classical linear gating with softmax and top-$k$ routing ($k = 2$) and 8 experts. The activation functions remains the same as for the dense base model from § 3. Given configuration of 124M dense Llama model, we result in approximately 520M parameter MoE model. In this setting, we train with a batch size of $256 \times 512$ for $T \in \{42, 336\}$k iterations ($\{5.5, 44\}$B tokens). If we assume that Chinchilla scaling law is applicable to this model, then it results in 10.4B tokens. See Appendix G.5 for more details.

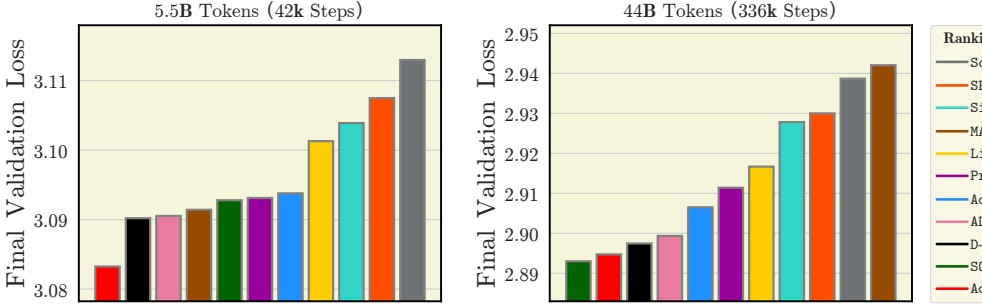

Figure 11: **Ranking optimizers for 520M MoE models with $256 \times 512$ batch size.** We report results for models trained for both 42k iterations (*left*), and 336k (*right*). MoE configuration correspond to one of the 124M dense model. Optimizer rankings closely mirror those in Figure 3 **(b)**, indicating that our benchmarking results transfer smoothly from dense models to MoEs. We also see that `SOAP` outperforms `AdEMAMix` in 336k steps run (see also Figure 2), however, with re-tuned beta parameters we might expect the opposite results in longer training (see Figures 4 and 13 **(b)**).

> **Takeaway 7.** Benchmarking results obtained for dense models transfer to corresponding MoEs.

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

# A   APPENDIX

## CONTENTS

# B  DISCUSSION

**A summary of results.** In this work, we benchmarked many interesting to community optimizers across architectural changes, model scales, training durations and batch sizes. After an extensive hyperparameter tuning, we revealed ablations for each optimizers showing their sensitivity to certain of them. We questioned flaws in popular code base for LLM pretraining—so important for careful benchmarking and the overall model performance. Regarding the benchmarking setup, we built a rankings of optimizers in each setup considered, if consider the global result and the question of the effectiveness of AdamW for LLMs, we point that there are new reliable optimizers that would be beneficial at scale—AdEMAMix, D-Muon, MARS. We point that methods such as ADOPT and Prodigy scale similarly to AdamW, and also worth a try for production purposes.

**Our advices on tuning each method.** Overall, we validate both widely used hyperparameters such as $\lambda = 0.1$ and $T_{\text{warmup}} \approx 2\text{k}$, and explore the sensitivity of optimizers to $\gamma$-schedulers, $\gamma$-decay, and optimizer-related hyperparameters. Notably, a large weight decay ensures faster convergence when training for fewer iterations, and large warmup of $25\%$ of the total training duration $T$ is beneficial for sign-based methods, Sophia, and SF-AdamW. For Lion—as mentioned in (Chen et al., 2023)—we find that the best value for $\beta_1$ is consistently 0.99. The mechanism for Lion appears similar to AdEMAMix, suggesting that Lion could perform better with larger $\beta_1$, which would require schedulers. We also pose an interesting observation toward Prodigy: while it may not be so efficient with very small batch sizes, with scaling of the model size and the batch size, it becomes almost as competitive as AdamW. MARS also benefits from large batches and continues to improve performance as the model size scales. For MARS, when optimizing 1D parameters with AdamW, we found that it is better to keep ($\beta_1$, $\beta_2$) of AdamW; for our largest models, $\beta_1 = 0.8$ performs slightly better than $\beta_1 = 0.9$. Additionally, MARS betas determined for 2D parameters in (Yuan et al., 2024) also seem to be the best in our settings. Basic Muon performs poorly at relatively small batch sizes (32, 256) across different model sizes and training lengths; however, applying weight decay to 2-dimensional parameters, as in D-Muon, resolves this and yields a robust optimizer across all benchmarking scenarios we considered. AdEMAMix remains the best optimizer overall, scaling efficiently with bath size, model size, and training horizons. Importantly, increasing $\beta_2$ for longer training substantially benefits AdEMAMix and other AdamW-like methods. Moreover, AdEMAMix allows using a large weight decay term $\lambda$ during prolonged training, e.g., runs of 128k iterations with $\lambda = 0.5$ still slightly outperform those with $\lambda = 0.1$. Beyond optimizer-specific hyperparameters, we show that the choice of $\gamma$-scheduler also depends on the optimizer selected. Regarding the learning rate, decaying $\gamma$ below $0.1 \times \gamma_{\text{max}}$ is important, as it significantly improves the optimization performance.

**Limitations.** We conduct our benchmarking experiments on models of up to 720M parameters, with long training runs of almost 50B tokens. The insights we find vary across scales, and training behavior may change further at practical scales and with extremely long training durations (Wei et al., 2022; Tay et al., 2022). Especially when certain optimizers are not widely supported by modern sharding frameworks (Zhao et al., 2023; Rajbhandari et al., 2020; Rasley et al., 2020) at the moment. Throughout this work, we study the loss dynamics, leaving aside downstream performances. Although these often scale reliably with loss  (Du et al., 2025; Gadre et al., 2024), there are also counterexamples (Xu et al., 2025; Liu et al., 2022). Bridging the gap between loss minimization and downstream task performance is important, as downstream abilities are ultimately the main metric of interest. We leave a deeper investigation of this connection to future research. We also do not cover previously explored Adan (Xie et al., 2024), NAdam(W) (Dozat, 2016), Shampoo (Gupta et al., 2018) optimizers, as well as recently introduced Scion (Pethick et al., 2025), novel variations of Muon (Qiu et al., 2025; An et al., 2025; Ahn & Xu, 2025), and others (Peng et al., 2024; Defazio, 2025; Guan, 2023; Wang et al., 2025). In addition, it is important to come up with a unified benchmark of optimizers for memory-efficient pretraining (Glentis et al., 2025; Zhu et al., 2025; Ma et al., 2025; Su et al., 2025), as they become more popular and argue that they might even outperform the AdamW baseline. We emphasize that there is still a huge branch of research on optimizers left to explore.

## C   OPTIMIZERS WE STUDY

In this section, we describe all considered algorithms, presenting them in a unified formalism. We start with notation and then discuss the algorithms according to their logical grouping:

1. `Adam`-like methods: `AdamW` (Algorithm 1), `ADOPT` (Algorithm 2), and `AdEMAMix` (Algorithm 3).

2. Sign-based methods: `Lion` (Algorithm 4), `Signum` (Algorithms 5 and 6).

3. Approximate second-order optimizers: `Muon` (Algorithm 8), `SOAP` (Algorithm 10), and `Sophia` (Algorithm 11).

4. Learning rate / scheduler-free learning algorithms: `Schedule-Free AdamW` (Algorithm 12), `Prodigy` (Algorithm 13).

5. `MARS` methods: (Algorithms 14, 15, 16).

**Notation.** In our work, we denote vectors and matrices in bold, and scalars in regular type. Let $\mathcal{L} : \mathcal{D} \to \mathbb{R}$ be an empirical loss function parameterized by $x$ and mapping a batch of inputs $\boldsymbol{\xi} \subset \mathcal{D}$ to $\mathbb{R}$. As $g = \nabla_x \mathcal{L}(x, \boldsymbol{\xi})$, we denote a stochastic gradient of the loss w.r.t. parameters $x$. For brevity, we omit $x$ in $\nabla$ and write $\nabla \mathcal{L}(x, \boldsymbol{\xi})$. We use the following standardized notation for specific symbols in our work: batch size—$|\boldsymbol{\xi}|$, learning rate—$\gamma$, weight decay—$\lambda$, momentum—$\beta$, iteration counter $t$ with the total number of iterations—$T$. And basic notation for symbols in the algorithms: $m, v$—are first and second moment estimates, respectively, with their bias corrected versions $\hat{m}, \hat{v}$, and beta parameters—$(\beta_1, \beta_2)$. We denote the dot product of two vectors $z, y$ as $\langle z, y \rangle$, while $z \odot y$ stands for their element-wise product. All division and addition operations in the described algorithms are element-wise.

### C.1   ADAMW, ADOPT, ADEMAMIX

**AdamW.** Our baseline—`Adam(W)`, has become a de facto optimizer for deep learning, demonstrating impressive performance across diverse domains—from tabular data to diffusion and language models.

The method originated from the ideas of `Adagrad` (Duchi et al., 2011) and `RMSProp` (Graves, 2014), which utilize a second moment estimate $v$ in their update rule. However, `Adam(W)` enhanced this prior scheme by incorporating momentum (Nemirovskii & Nesterov, 1985; Sutskever et al., 2013), establishing itself as a state-of-the-art method for a wide range of tasks. All other algorithms we consider also employ a similar, if not identical, momentum scheme.

A key difference between `Adam` and `AdamW` is the use of decoupled weight decay (Loshchilov & Hutter, 2019) in the latter. We adopt the decoupled weight decay scheme for all methods to ensure consistency, as correct weight decay is critical for optimizer comparison, hyperparameter tuning, and final performance. The importance of the correct weight decay implementation is clearly observable, e.g., for `Signum`.

---

**Algorithm 1** `AdamW`

1: **Input:** Initial parameters $x_0$, number of iterations $T$, learning rate $\gamma_t$, weight decay $\lambda$, $\beta_1$, $\beta_2$, $\varepsilon$.
2: **Initialize:** $m_0 \leftarrow 0$, $v_0 \leftarrow 0$
3: **for** $t \in [T]$ **do**
4:     $g_t \leftarrow \nabla \mathcal{L}(x_t, \boldsymbol{\xi}_t)$
5:     $m_t \leftarrow \beta_1 m_{t-1} + (1 - \beta_1) g_t$
6:     $v_t \leftarrow \beta_2 v_{t-1} + (1 - \beta_2) g_t \odot g_t$
7:     $\hat{m}_t \leftarrow m_t / (1 - \beta_1^t)$,   $\hat{v}_t \leftarrow v_t / (1 - \beta_2^t)$
8:     $x_{t+1} \leftarrow x_t - \gamma_t \left( \frac{\hat{m}_t}{\sqrt{\hat{v}_t} + \varepsilon} + \lambda x_t \right)$
9: **end for**
10: **Return:** $x_T$

---

**ADOPT.** Recently, Taniguchi et al. (2024) proposed a modification of `Adam`, by removing the current gradient $\boldsymbol{g}_t$ from the second moment estimate $\boldsymbol{v}_t$ and altering the order of the momentum update $\boldsymbol{m}_t$ and normalization. As shown in `line 8` of Algorithm 2, the parameter update depends only on the previous value of the second moment estimate $\boldsymbol{v}_{t-1}$. The authors analyze the convergence of ADOPT with the following update rule:

$$\boldsymbol{m}_t \leftarrow \beta_1 \boldsymbol{m}_{t-1} + (1 - \beta_1)\frac{\boldsymbol{g}_t}{\max\{\sqrt{\boldsymbol{v}_{t-1}}, \varepsilon\}},$$

$$\boldsymbol{x}_{t+1} \leftarrow \boldsymbol{x}_t - \gamma_t \boldsymbol{m}_t.$$

However, the practical implementation differs in a few details. To tackle instabilities caused by near-zero gradients during the early stages of training, the authors propose using a clipping on $\boldsymbol{g}_t / \max\{\sqrt{\boldsymbol{v}_{t-1}}, \varepsilon\}$, which we formalize as the `clamp` operation. Given a vector $\boldsymbol{g}$ and a positive scalar $c$, it is defined as:

$$\texttt{clamp}\,(\boldsymbol{g}, c)^{(I)} = \min\left\{\max\left\{g^{(I)}, -c\right\}, c\right\}. \tag{1}$$

Thus, the element-wise `clamp` operation preserves $\boldsymbol{g}_t$ from the division by near-zero values.

The authors theoretically claim that ADOPT achieves the optimal convergence bound for smooth non-convex objectives, regardless of the choice of the $\beta_2$ parameter. We empirically investigate this claim and observe that, contrary to the theoretical results, there is a significant performance gap for different choices of $\beta_2$ in practice—see Figure 29. Also, the effect of $\varepsilon$ in Algorithm 2 is intriguing: in contrast to the typical $\varepsilon = 10^{-8}$ value for `AdamW`, the authors pose that for ADOPT mechanism the smaller value of $10^{-6}$ is more suitable. We notice that this also holds in practice for the method, and we provide the corresponding ablation in Figure 38.

---

**Algorithm 2** ADOPT

---

1: **Input:** Initial parameters $\boldsymbol{x}_0$, number of iterations $T$, learning rate $\gamma_t$, weight decay $\lambda$, $\beta_1$, $\beta_2$, $\varepsilon$.
2: **Initialize:** $\boldsymbol{m}_0 \leftarrow \boldsymbol{0}$, $\boldsymbol{v}_0 \leftarrow \nabla\mathcal{L}(\boldsymbol{x}_0, \boldsymbol{\xi}_0) \odot \nabla\mathcal{L}(\boldsymbol{x}_0, \boldsymbol{\xi}_0)$
3: **for** $t \in [T]$ **do**
4: $\quad \boldsymbol{g}_t \leftarrow \nabla\mathcal{L}(\boldsymbol{x}_t, \boldsymbol{\xi}_t)$
5: $\quad c_t \leftarrow t^{1/4}$ $\hfill \triangleright$ Update clipping value schedule
6: $\quad \boldsymbol{m}_t \leftarrow \beta_1 \boldsymbol{m}_{t-1} + (1 - \beta_1)\texttt{clamp}\left(\frac{\boldsymbol{g}_t}{\max\{\sqrt{\boldsymbol{v}_{t-1}}, \varepsilon\}}, c_t\right)$
7: $\quad \boldsymbol{v}_t \leftarrow \beta_2 \boldsymbol{v}_{t-1} + (1 - \beta_2)\boldsymbol{g}_t \odot \boldsymbol{g}_t$
8: $\quad \boldsymbol{x}_{t+1} \leftarrow \boldsymbol{x}_t - \gamma_t (\boldsymbol{m}_t + \lambda \boldsymbol{x}_t)$ $\hfill \triangleright$ Update without $\boldsymbol{v}_t$
9: **end for**
10: **Return:** $\boldsymbol{x}_T$

---

**AdEMAMix.** Another `Adam`-like optimizer we study is `AdEMAMix` (Pagliardini et al., 2024). This work argues that using a single EMA to accumulate past gradients in the first moment estimate $\boldsymbol{m}$ can be suboptimal, as it cannot simultaneously prioritize both immediate past and older gradients. In Algorithm 3, the authors incorporate two EMAs: one—`Adam`-like EMA for $\boldsymbol{m}$ (fast), and another—a slow EMA $\boldsymbol{m}^{\text{slow}}$ (see `line 7`) with an additional $\beta_3$ parameter. In the update rule, fast and slow EMAs are balanced with the constant factor $\alpha$ (see `line 10` of Algorithm 3). This algorithmic design enables `AdEMAMix` to benefit from older gradients, resulting in smoother loss curves during training.

To mitigate the effect of early instabilities, the authors use two additional schedulers for $\alpha$ and $\beta_3$ – `alpha_scheduler` and `beta_scheduler`, formalized in our work as follows:

$$\texttt{alpha\_scheduler}(t, \alpha, T_\alpha) = \min\left\{\frac{t\alpha}{T_\alpha}, \alpha\right\},$$

$$\texttt{beta\_scheduler}(t, \beta_3, \beta_{\text{start}}, T_{\beta_3}) = \min\left\{\exp\left(\frac{\log(\beta_{\text{start}})\log(\beta_3)}{\left(1 - \frac{t}{T_{\beta_3}}\right)\log(\beta_3) + \frac{t}{T_{\beta_3}}\log(\beta_{\text{start}})}\right), \beta_3\right\}.$$

In all experiments, we set $\beta_{\text{start}} = \beta_1$, and the warmup parameters equal to the length of training: $T_\alpha = T_{\beta_3} = T$.

Although these schedulers may seem at odds with the WSD scheduler (Hu et al., 2024), setting $T_\alpha, T_{\beta_3}$ longer than the first WSD checkpoint does not noticeably harm performance. Thus, `AdEMAMix` can still be combined with recent findings regarding the WSD scheduler.

---

**Algorithm 3** `AdEMAMix`

---

1: **Input:** Initial parameters $\boldsymbol{x}_0$, number of iterations $T$, learning rate $\gamma_t$, weight decay $\lambda$, $\beta_1$, $\beta_2$, $\beta_3$, $\beta_{\text{start}}$, $\alpha$, `beta_scheduler`, `alpha_scheduler`, warmup parameters $T_{\beta_3}$ and $T_\alpha$, $\varepsilon$.
2: **Initialize:** $\boldsymbol{m}_0 \leftarrow \boldsymbol{0}, \boldsymbol{m}_0^{\text{slow}} \leftarrow \boldsymbol{0}, \boldsymbol{v}_0 \leftarrow \boldsymbol{0}$
3: **for** $t \in [T]$ **do**
4:     $\beta_3(t) \leftarrow \texttt{beta\_scheduler}(t, \beta_3, \beta_{\text{start}}, T_{\beta_3}), \alpha(t) \leftarrow \texttt{alpha\_scheduler}(t, \alpha, T_\alpha)$     ▷ Update $\beta_3$ and $\alpha$ schedulers
5:     $\boldsymbol{g}_t \leftarrow \nabla\mathcal{L}(\boldsymbol{x}_t, \boldsymbol{\xi}_t)$
6:     $\boldsymbol{m}_t \leftarrow \beta_1 \boldsymbol{m}_{t-1} + (1 - \beta_1)\boldsymbol{g}_t$
7:     $\boldsymbol{m}_t^{\text{slow}} \leftarrow \beta_3(t)\boldsymbol{m}_{t-1}^{\text{slow}} + (1 - \beta_3(t))\boldsymbol{g}_t$     ▷ Update slow EMA
8:     $\boldsymbol{v}_t \leftarrow \beta_2 \boldsymbol{v}_{t-1} + (1 - \beta_2)\boldsymbol{g}_t \odot \boldsymbol{g}_t$
9:     $\hat{\boldsymbol{m}}_t \leftarrow \boldsymbol{m}_t/(1 - \beta_1^t), \quad \hat{\boldsymbol{v}}_t \leftarrow \boldsymbol{v}_t/(1 - \beta_2^t)$
10:     $\boldsymbol{x}_{t+1} \leftarrow \boldsymbol{x}_t - \gamma_t \left( \frac{\hat{\boldsymbol{m}}_t + \alpha(t)\boldsymbol{m}_t^{\text{slow}}}{\sqrt{\hat{\boldsymbol{v}}_t} + \varepsilon} + \lambda \boldsymbol{x}_t \right)$
11: **end for**
12: **Return:** $\boldsymbol{x}_T$

---

### C.2 SIGN-BASED METHODS: LION AND SIGNUM

Another branch of methods includes sign-based methods, represented by `Lion` and `Signum`. To some extent, one can classify `Adam` as a sign-based optimizer also, but we mention only `Lion` and `Signum` as they explicitly incorporate the `sign` operation in the update rule.

These methods, particularly `Signum`, have been unfairly overlooked in the context of LLM pre-training. However, our results demonstrate that, with sufficiently large batch sizes and at moderate model scales, these optimizers perform on par with `Adam`, and in some cases even outperform it.

**Lion.** The first sign-based method we study is `Lion` (Chen et al., 2023). This optimizer is symbolically discovered in the program space of first-order optimization primitives. `Lion` updates its EMA of $\boldsymbol{m}$ after updating the parameters and has additional term $(1 - \beta_1)\boldsymbol{g}$ which adds to the momentum. This interpolation $\beta_1 \boldsymbol{m}_{t-1} + (1 - \beta_1)\boldsymbol{g}_t$ (see line 6 of Algorithm 4) makes the symbolic-discovered idea behind `Lion` similar to the idea of the `AdEMAMix` optimizer.

---

**Algorithm 4** `Lion`

---

1: **Input:** Initial parameters $\boldsymbol{x}_0$, number of iterations $T$, learning rate $\gamma_t$, weight decay $\lambda$, $\beta_1$, $\beta_2$.
2: **Initialize:** $\boldsymbol{m}_0 \leftarrow \boldsymbol{0}$
3: **for** $t \in [T]$ **do**
4:     $\boldsymbol{g}_t \leftarrow \nabla\mathcal{L}(\boldsymbol{x}_t, \boldsymbol{\xi}_t)$
5:     $\boldsymbol{m}_t \leftarrow \beta_2 \boldsymbol{m}_{t-1} + (1 - \beta_2)\boldsymbol{g}_t$     ▷ Update EMA of $\boldsymbol{g}_t$
6:     $\boldsymbol{x}_{t+1} \leftarrow \boldsymbol{x}_t - \gamma_t \left( \texttt{sign}\left( \beta_1 \boldsymbol{m}_{t-1} + (1 - \beta_1)\boldsymbol{g}_t \right) + \lambda \boldsymbol{x}_t \right)$
7: **end for**
8: **Return:** $\boldsymbol{x}_T$

---

**Signum.** Another sign-based method, which is the adoptation of `signSGD`—`Signum` (Bernstein et al., 2018) (or, alternatively, `signSGD` with momentum). This method differs from `Lion` in the interpolation term between the EMA of momentum and the current gradient, as well as in the `Signum`'s update rule, where a current EMA is used.

Importantly, while `Signum` is not yet as widespread for LLM pretraining and has largely remained a theoretical artifact, recent studies have begun to adopt `Signum` for scalable training (Zhao et al., 2025), primarily due to its memory efficiency compared to `AdamW`.

In this regard, we would like to highlight that many recent PyTorch (Paszke et al., 2019) implementations of the `Signum` optimizer are unlikely to be suitable for this method, which impairs its potential performance.

The main issue with open-source implementations is the use of decoupled weight decay in the PyTorch implementation of SGDM (SGD with momentum) (Sutskever et al., 2013). Indeed, with decoupled weight decay, the update in Algorithm 5 transforms into:

$$\boldsymbol{x}_{t+1} \leftarrow \boldsymbol{x}_t - \gamma_t \texttt{sign}\left(\beta \boldsymbol{m}_{t-1} + (1-\beta)\boldsymbol{g}_t - \lambda(1-\beta)\boldsymbol{g}_t\right),$$

which affects the sign of the update, potentially leading to the wrong optimization direction if the weight decay is sufficiently large. See Figures 5 **(a)** and 15 for the impact of the correct weight decay implementation for sign-based methods like Signum and Lion.

Another popular failure while using Signum is the incorrectly tractable PyTorch implementation of SGD with momentum. It does not include such EMA as line 5 in Algorithm 5, on the other hand, in PyTorch, the momentum update depends on the dampening parameter $\tau$:

$$\boldsymbol{m}_t \leftarrow \beta \boldsymbol{m}_{t-1} + (1-\tau)\boldsymbol{g}_t,$$

where $\tau$ is zero by default. Therefore, the typical update rule, reflecting the actual Signum behavior in practice, corresponds to the following update:

$$\boldsymbol{x}_{t+1} \leftarrow \boldsymbol{x}_t - \gamma_t \left(\texttt{sign}\left(\beta \boldsymbol{m}_{t-1} + (1-\tau)\boldsymbol{g}_t\right) + \lambda \boldsymbol{x}_t\right),$$

where the weight decay is decoupled and, consequently, does not affect $\texttt{sign}$.

However, we found out that the PyTorch implementation of Nesterov momentum (Nesterov, 1983)

$$\boldsymbol{g}_t \leftarrow \boldsymbol{g}_t + \beta \boldsymbol{m}_t,$$

improves Signum. Since enabling Nesterov momentum requires zero dampening $\tau$, we revisited the description of Algorithm 5 and propose more practical, PyTorch-compatible version of Signum in Algorithm 6. We study the role of dampening and Nesterov momentum in our variant of Signum in Figure 33.

| **Algorithm 5** Signum (basic) | **Algorithm 6** Signum (our PyTorch variant) |
|---|---|
| 1: **Input:** Initial parameters $\boldsymbol{x}_0$, number of iterations $T$, learning rate $\gamma_t$, weight decay $\lambda$, momentum $\beta$. | 1: **Input:** Initial parameters $\boldsymbol{x}_0$, number of iterations $T$, learning rate $\gamma_t$, weight decay $\lambda$, momentum $\beta$. |
| 2: **Initialize:** $\boldsymbol{m}_0 \leftarrow \mathbf{0}$ | 2: **Initialize:** $\boldsymbol{m}_0 \leftarrow \mathbf{0}$ |
| 3: **for** $t \in [T]$ **do** | 3: **for** $t \in [T]$ **do** |
| 4: $\quad \boldsymbol{g}_t \leftarrow \nabla \mathcal{L}(\boldsymbol{x}_t, \boldsymbol{\xi}_t)$ | 4: $\quad \boldsymbol{g}_t \leftarrow \mathcal{L}(\boldsymbol{x}_t, \boldsymbol{\xi}_t)$ |
| 5: $\quad \boldsymbol{m}_t \leftarrow \beta \boldsymbol{m}_{t-1} + (1-\beta)\boldsymbol{g}_t$ | 5: $\quad \boldsymbol{m}_t \leftarrow \beta \boldsymbol{m}_{t-1} + \boldsymbol{g}_t$ |
| 6: $\quad \boldsymbol{x}_{t+1} \leftarrow \boldsymbol{x}_t - \gamma_t \left(\texttt{sign}\left(\boldsymbol{m}_t\right) + \lambda \boldsymbol{x}_t\right)$ | 6: $\quad \boldsymbol{x}_{t+1} \leftarrow \boldsymbol{x}_t - \gamma_t \left(\texttt{sign}\left(\beta \boldsymbol{m}_t + \boldsymbol{g}_t\right) + \lambda \boldsymbol{x}_t\right)$ |
| 7: **end for** | 7: **end for** |
| 8: **Return:** $\boldsymbol{x}_T$ | 8: **Return:** $\boldsymbol{x}_T$ |

Moreover, to prevent other researchers and practitioners from the possible wrong use of Signum, and to ensure reproducibility, we provide our Python code.

Listing 1: **Signum code skeleton using PyTorch.**

```python
from typing import Dict

import torch

class Signum(torch.optim.Optimizer):
    def __init__(
        self,
        params,
        lr=1e-3,
        momentum=0,
        dampening=0,
        weight_decay=0,
        nesterov=False,
        sign_update=True,
    ):
        if lr < 0.0:
            raise ValueError(f"Invalid learning rate: {lr}")
        if momentum < 0.0:
            raise ValueError(f"Invalid momentum value: {momentum}")
        if weight_decay < 0.0:
```

```
22              raise ValueError(f"Invalid weight_decay value: {weight_decay}")
23
24          defaults = dict(
25              lr=lr,
26              momentum=momentum,
27              dampening=dampening,
28              weight_decay=weight_decay,
29              nesterov=nesterov,
30              sign_update=sign_update,
31          )
32          if nesterov and (momentum <= 0 or dampening != 0):
33              raise ValueError("Nesterov momentum requires a momentum and zero dampening")
34          super().__init__(params, defaults)
35
36      def __setstate__(self, state):
37          super().__setstate__(state)
38          for group in self.param_groups:
39              group.setdefault("nesterov", False)
40
41      @torch.no_grad()
42      def _init_state(self, example, state=None):
43          assert isinstance(example, torch.Tensor)
44          assert isinstance(state, Dict) or state is None
45          if state is None:
46              state = {}
47          state["step"] = 0
48          state["momentum_buffer"] = torch.clone(example).detach()
49          return state
50
51      @torch.no_grad()
52      def _compute_update(
53          self, grad, state, lr, momentum, nesterov, dampening, sign_update, **kwargs
54      ):
55          if momentum != 0:  # Signum check
56              buf = state["momentum_buffer"]
57              buf.mul_(momentum).add_(grad, alpha=1 - dampening)
58
59              if nesterov:
60                  grad = grad.add(buf, alpha=momentum)
61              else:
62                  grad = buf
63
64          if sign_update:
65              grad = grad.sign()
66
67          return grad * (-lr)
68
69      @torch.no_grad()
70      def step(self, closure=None):
71          """Performs a single optimization step.
72
73          Args:
74              closure (Callable, optional): A closure that reevaluates the model
75                  and returns the loss.
76          """
77          loss = None
78          if closure is not None:
79              with torch.enable_grad():
80                  loss = closure()
81
82          for group in self.param_groups:
83              for p in group["params"]:
84                  if p.grad is None:
85                      continue
86
87                  grad = p.grad
88                  state = self.state[p]
89
90                  if group["weight_decay"] != 0:
91                      p.mul_(1 - group["lr"] * group["weight_decay"])
92
93                  if len(state) == 0:
94                      self._init_state(example=p, state=state)
95                      if not group["momentum"]:
96                          state.pop("momentum_buffer", None)
97
98                  state["step"] += 1
99
100                 update = self._compute_update(
101                     grad,
102                     state,
103                     group["lr"],
104                     group["momentum"],
105                     group["nesterov"],
106                     group["dampening"],
107                     group["sign_update"],
108                 )
109
110                 p.add_(update)
111
112         return loss
```

## C.3 MUON & D-MUON, SOAP, SOPHIA

The next page of the methods covers algorithms that rather aim to use more information about the problem's curvature (SOAP (Vyas et al., 2024), Sophia (Liu et al., 2024)) or perform fast updates of matrix parameters involving higher order computations (Muon (Jordan et al., 2024b)).

Contrary to chronological order, we discuss them starting from the recent one—Muon and end up with Sophia.

**Muon & D-Muon.** Specifically designed for speedrun comparisons, Muon surpasses the AdamW baseline on the nanoGPT pretraining benchmark (Jordan et al., 2024a). Claims from the Muon project extend to faster learning, lower memory usage and better sample-efficiency, with a small overhead in wall-clock time.

The reason why Muon is a good option for speedrun pretraining lies in its structure—Muon treats different parameters based on their tensor dimensionality. One-dimensional (1D) parameters, large embedding layers, Layer Norm (or RMSNorm) parameters, and the output layer of LLM (lm_head) are optimized by AdamW. And all parameters with two or more dimensions (e.g., Multi-Head Attention layers) are optimized by Algorithm 7, which we call MuonNon1D.

Inspired by Shampoo's preconditioners (Gupta et al., 2018), the authors of MuonNon1D incorporated an orthogonalization step to compute SVD transformation of the gradient matrix. Before the orthogonalization step, MuonNon1D resembles SGD with Nesterov momentum. To ensure a fast orthogonalization procedure, the authors, inspired by (Bernstein & Newhouse, 2024), use the Newton-Schulz procedure (Higham, 2008). As the number of Newton-Schulz iterations increases, the resulting matrix becomes closer to $UV^\top$ from SVD transformation. The authors also mention that Muon can be considered an alternative method of smoothing spectral steepest descent (Carlson et al., 2015), offering a distinct set of memory and runtime trade-offs compared to Shampoo.

---

**Algorithm 7** MuonNon1D (for non-1D parameters)

1: **Input:** Initial non-1D parameters $x_0$, number of iterations $T$, learning rate $\gamma_t$, momentum $\beta$, number of Newton-Schulz iterations $T_{\text{NS}}$, $a$, $b$, $c$ coefficients.
2: **Initialize:** $m_0 \leftarrow 0$
3: **for** $t \in [T]$ **do**
4:      $g_t \leftarrow \nabla\mathcal{L}(x_t, \xi_t)$
5:      $m_t \leftarrow \beta m_{t-1} + g_t$
6:      $g_t \leftarrow \beta m_t + g_t$            ▷ Practical implementation of Nesterov momentum
7:      **Set:** $w_0 \leftarrow g_t / \|g_t\|_F$
8:      **for** $n \in [T_{\text{NS}}]$ **do**
9:          $w_{n+1} \leftarrow a w_n + b w_n w_n^\top + c \left( w_n w_n^\top \right)^2 w_n$      ▷ Newton-Schulz iteration
10:      **end for**
11:      $x_{t+1} \leftarrow x_t - \gamma_t w_{T_{\text{NS}}}$
12: **end for**
13: **Return:** $x_T$

---

Importantly, we noticed that the original algorithmic description of Muon optimizer, provided in the official repository[2], differs from the actual one, presented in Algorithm 7. In the original code, as well as in our benchmarking, weight decay does not apply to the matrix parameters in the optimizer state of MuonNon1D, meaning that the only weight decay used during training is AdamW's weight decay. From this perspective, we observe that the gap between the final loss values for runs with weight decay of 0.1 and 0 almost disappears, while the run with a weight decay of 0.5 becomes the worst, which is not the case for other optimizers. See Figures 5 and 15 regarding these ablations.

Noticeably, the weight decay issue was addressed in the recent paper (Liu et al., 2025), in which the authors also present a scheme for sharing the learning rate and weight decay between the matrix and non-matrix parameters of the model. They do this via the RMS heuristic: since AdamW has the property of keeping its RMS updates close to 1 (Hinton et al., 2012), particularly around 0.2-0.4 in the practice of LLM training (Liu et al., 2025; AI et al., 2025), they scale the RMS update of

---

[2]https://github.com/KellerJordan/modded-nanogpt

---

**Algorithm 8** Muon (general scheme)

---

1: **Input:** Initial parameters $x_0$, number of iterations $T$. Muon's parameters: learning rate $\gamma_t^{\text{M}}$, momentum $\beta$, number of Newton-Schulz iterations $T_{\text{NS}}$, $a$, $b$, $c$ coefficients. AdamW's parameters: learning rate $\gamma_t^{\text{A}}$, weight decay $\lambda$, $\beta_1$, $\beta_2$, $\varepsilon$.
2: **for** $t \in [T]$ **do**
3:     **if** $x_t \in \{\texttt{embeds}, \texttt{scalar\_params}, \texttt{lm\_head}\}$ **then**
4:         $x_t^{\text{A}} \leftarrow x_t$
5:         $x_{t+1}^{\text{A}} \leftarrow \text{AdamW}\left(x_t^{\text{A}}, \gamma_t^{\text{A}}, \lambda, \beta_1, \beta_2, \varepsilon, T=1\right)$         ▷ One iteration of AdamW
6:     **else**
7:         $x_t^{\text{M}} \leftarrow x_t$
8:         $x_{t+1}^{\text{M}} \leftarrow \text{MuonNon1D}\left(x_t^{\text{M}}, \gamma_t^{\text{M}}, T_{\text{NS}}, \beta, a, b, c, T=1\right)$   ▷ One iteration of MuonNon1D
9:     **end if**
10: **end for**
11: **Return:** $x_T^{\text{A}}, x_T^{\text{M}}$

---

Muon to this range. With these adjustments, practitioners do not need to tune the learning rate and weight decay for 1D and non-1D parameters separately, which is a significant bonus of the newer Muon-like algorithm. We include this variation of Muon under the D-Muon naming.

Our ablations demonstrate that D-Muon scales better than the basic Muon in all settings we have considered so far (see Figures 1, 3, 12, 36, 37 and 40). We also report a detailed comparison of these two similar methods in Figures 16 and 17, and discuss their close connection with the weight decay applied to non-1D parameters in the D-Muon algorithm. Refer to this ablation in § 4.

Another interesting aspect of Muon is the effect of the Newton-Schulz orthogonalization procedure (Bernstein & Newhouse, 2024; Higham, 2008) on optimization. We show how the number of Newton-Schulz steps impacts the performance of Muon in Figure 32. Furthermore, we pose that improving the orthogonalization procedure in methods like Muon, Scion, MARS-Shampoo (see Algorithm 16) could substantially improve their overall performance. Recent work has already begun to explore this avenue (Amsel et al., 2025; Grishina et al., 2025), but a deeper investigation remains an open research challenge.

**SOAP.** (Vyas et al., 2024) proposed a new, improved modification of Shampoo (Gupta et al., 2018). SOAP reduces the computational overhead by optimizing only two-dimensional layers (2D) via Algorithm 9, while running AdamW for 1D layers. At initialization, the preconditioners are computed via the eigenvector decomposition of the initial gradient matrices $\texttt{eigenbasis}\left(\nabla\mathcal{L}(x_0, \xi_0)\nabla\mathcal{L}(x_0, \xi_0)^\top\right)$: $\nabla\mathcal{L}(x_0, \xi_0)\nabla\mathcal{L}(x_0, \xi_0)^\top = q\Lambda q^{-1}$, where $\Lambda$ stands for the diagonal matrix whose diagonal elements are the corresponding eigenvalues. For subsequent iterations, SOAPNon1D rotates gradients into this slowly changing basis, maintains second-moment statistics in that basis, and periodically updates the basis via QR decomposition (see lines 15, 16 of Algorithm 9) for all 2D layers (except for embeds and lm_head). This is the main computational part of the method.

A key idea behind the SOAP optimizer is:

1. Given the slowly changing coordinate basis provided by eigenvectors $l$ and r, SOAP updates its second moment estimates in this basis; that is to say, it runs AdamW in another, a rotated space.

2. To update the eigenvectors of $l$ and r, SOAP runs QR decomposition with the preconditioning frequency $\phi$.

In Algorithm 9, setting both $q_l$ and $q_r$ to the identity matrix would result in AdamW.

The overall SOAP algorithm can be formalized as Algorithm 10.

**Sophia.** Despite being named a second-order optimizer, Sophia (Liu et al., 2024) performs an update that is quite similar to Adam's. It also leverages the diagonal preconditioner $h$, but not the curvature information of the optimization problem, which depends on the non-diagonal terms of the Hessian. One should notice that Sophia was introduced with two types of preconditioner—Hutchinson (Bekas et al., 2007) and Gauss-Newton-Bartlett (Martens, 2020). Since the latter shows more promising performance, we only consider this type of preconditioner for Sophia.

**Algorithm 9** `SOAPNon1D` (for non-1D parameters)

---

1: **Input:** Initial parameters $x_0$, number of iterations $T$, learning rate $\gamma_t$, weight decay $\lambda$, $\beta_1$, $\beta_2$, preconditioning frequency $\phi$, $\varepsilon$.
2: **Initialize:** $m_0 \leftarrow \mathbf{0}$, $v_0 \leftarrow \mathbf{0}$
3: **Initialize preconditioners:** $q_l, q_r \leftarrow \texttt{eigenbasis}\left(\nabla\mathcal{L}(x_0, \xi_0)\nabla\mathcal{L}(x_0, \xi_0)^\top\right)$
4: **for** $t \in [T]$ **do**
5:      $g_t \leftarrow \nabla\mathcal{L}(x_t, \xi_t)$
6:      $g'_t \leftarrow q_l^\top g_t q_r$                                       ▷ Rotate $g_t$
7:      $m_t \leftarrow \beta_1 m_{t-1} + (1 - \beta_1)g_t$
8:      $m'_t \leftarrow q_l^\top m_t q_r$                    ▷ Compute `Adam`'s statistics in rotational space
9:      $v_t \leftarrow \beta_2 v_{t-1} + (1 - \beta_2)g'_t \odot g'_t$
10:     $\gamma_t \leftarrow \gamma_t \frac{\sqrt{1-\beta_2^t}}{1-\beta_1^t}$                           ▷ Optional: use bias correction
11:     $x_{t+1} \leftarrow x_t - \gamma_t \left(q_l \frac{m'_t}{\sqrt{v_t}+\varepsilon}q_r^\top + \lambda x_t\right)$       ▷ Perform update in original space
12:     $l_t \leftarrow \beta_2 l_{t-1} + (1 - \beta_2)g_t g_t^\top$                     ▷ Update preconditioners
13:     $r_t \leftarrow \beta_2 r_{t-1} + (1 - \beta_2)g_t^\top g_t$
14:     **if** $t \equiv 1 \pmod{\phi}$ **then**
15:        $q_l \leftarrow \texttt{QR}\left(l_t q_l\right)$
16:        $q_r \leftarrow \texttt{QR}\left(r_t q_r\right)$
17:     **end if**
18: **end for**
19: **Return:** $x_T$

---

**Algorithm 10** `SOAP` (general scheme)

---

1: **Input:** Initial parameters $x_0$, number of iterations $T$, learning rate $\gamma_t$, weight decay $\lambda$, $\beta_1$, $\beta_2$, preconditioning frequency $\phi$, $\varepsilon$.
2: **for** $t \in [T]$ **do**
3:      **if** $x_t \in \{\texttt{embeds}, \texttt{scalar\_params}, \texttt{lm\_head}\}$ **then**
4:         $x_t^\texttt{A} \leftarrow x_t$
5:         $x_{t+1}^\texttt{A} \leftarrow \texttt{AdamW}\left(x_t^\texttt{A}, \gamma_t, \lambda, \beta_1, \beta_2, \varepsilon, T=1\right)$       ▷ One iteration of `AdamW`
6:      **else**
7:         $x_t^\texttt{S} \leftarrow x_t$
8:         $x_{t+1}^\texttt{S} \leftarrow \texttt{SOAPNon1D}\left(x_t^\texttt{S}, \gamma_t, \lambda, \beta_1, \beta_2, \varepsilon, T=1\right)$    ▷ One iteration of `SOAPNon1D`
9:      **end if**
10: **end for**
11: **Return:** $x_T^\texttt{A}, x_T^\texttt{S}$

---

Every $\phi$ iterations, `Sophia` updates its second moment estimate by computing the gradient $\hat{g}$ of the empirical loss $\mathcal{L}$ given `softmax` of the logits instead of the true logits. Multiplying by the batch size, we obtain $\hat{h}$, after that, `Sophia` updates the EMA of $\hat{h}$.

Importantly, we found that the algorithmic description of `Sophia` in the original paper differs in minor details from the code implementation[3]. Indeed, the update rule in their work is formulated as follows:

$$x_{t+1} \leftarrow x_t - \gamma_t \texttt{clamp}\left(\frac{m_t}{\max\{\rho h_t, \varepsilon\}}, 1\right),$$

where `clamp` is defined as in Equation (1).

On the other hand, the code from the official repository suggests:

Listing 2: **`Sophia` update skeleton using PyTorch.**

```
1 # update step
2 step_t += 1
3
```

---

[3]https://github.com/Liuhong99/Sophia

```
4  # Perform stepweight decay
5  param.mul_(1 - lr * weight_decay)
6
7  # Decay the first and second moment running average coefficient
8  exp_avg.mul_(beta1).add_(grad, alpha=1 - beta1)
9
10 else:
11     step_size_neg = -lr
12
13     ratio = (exp_avg.abs() / (rho * bs * hess + 1e-15)).clamp(None, 1)
14     param.addcmul_(exp_avg.sign(), ratio, value=step_size_neg)
```

Therefore, the update rule of `Sophia` is misstated in the original paper and should be corrected to match `line 16` of Algorithm 11.

> **Takeaway 8.** The actual update rule of `Sophia` does not match its description in the original paper.

---

**Algorithm 11** `Sophia`

---

1: **Input:** Initial parameters $x_0$, number of iterations $T$, learning rate $\gamma_t$, weight decay $\lambda$, $\beta_1$, $\beta_2$, estimator frequency $\phi$, scaling factor $\rho$, $\varepsilon$.
2: **Initialize:** $m_0 \leftarrow 0$, $h_0 \leftarrow 0$
3: **for** $t \in [T]$ **do**
4:     $g_t \leftarrow \nabla \mathcal{L}(x_t, \xi_t)$
5:     $m_t \leftarrow \beta_1 m_{t-1} + (1 - \beta_1) g_t$
6:     **if** $t \equiv 1 \pmod{\phi}$ **then**
7:         $p_t \leftarrow \xi_t$                                    ▷ Obtain logits from batch
8:         $p_t \leftarrow \text{softmax}(p_t)$                        ▷ Sample from logits
9:         $\hat{\mathcal{L}}(x_t, \xi_t) \leftarrow p_t$              ▷ Loss, where $p_t$ are labels
10:         $\hat{g}_t \leftarrow \nabla \hat{\mathcal{L}}(x_t, \xi_t)$
11:         $\hat{h}_t \leftarrow |\xi_t| \hat{g}_t \odot \hat{g}_t$
12:         $h_t \leftarrow \beta_2 h_{t-\phi} + (1 - \beta_2) \hat{h}_t$
13:     **else**
14:         $h_t \leftarrow h_{t-1}$
15:     **end if**
16:     $x_{t+1} \leftarrow x_t - \gamma_t \left( \text{sign}(m_t) \min \left\{ \frac{|m_t|}{\rho h_t + \varepsilon}, 1 \right\} + \lambda x_t \right)$
17: **end for**
18: **Return:** $x_T$

---

### C.4 SCHEDULE-FREE ADAMW, PRODIGY

In this section, we outline two more players—`Schedule-Free AdamW` (Defazio et al., 2024b) and `Prodigy` (Mishchenko & Defazio, 2024). Both of them have a promising advantages and require less hyperparameter tuning which paves the road to parameter-free optimizers.

**Schedule-Free AdamW.** Defazio et al. (2024b) introduced the notion of schedule-free optimizers. The underlying idea behind `Schedule-Free SGD` and `Schedule-Free AdamW` (`SF-AdamW`) is to eliminate learning rate schedulers by replacing them with iterate averaging. Specifically, the schedule-free method uses interpolation between Polyak-Ruppert averaging (Polyak, 1990; Ruppert, 1988) and Primal averaging (Nesterov & Shikhman, 2015) for the momentum update, rather than the usual EMA (see `line 4` of Algorithm 12). To stabilize scalable training, the authors also propose an internal warmup mechanism (see `line 7` of Algorithm 12), which gradually increases the learning rate while ensuring `Adam`-style bias correction.

An interesting result we observe, is that `SF-AdamW` shows the best performance with a larger number of warmup iterations compared to other methods—see Figure 7.

Another key point—training with `SF-AdamW` is sensitive to the choice of beta parameters. Unlike in `AdamW`, these parameters play distinct roles in `SF-AdamW`: $\beta_1$ determines the interpolation between the $z_t$ and $x_t$ sequences, which acts as a form of schedule-free momentum. Specifically, the term $(1 - \beta_1) g_t$ is immediately incorporated into the iterate sequence $y_t$, while the remainder of $g_t$ is

gradually incorporated through averaging—a mechanism analogous to the momentum EMA, but with a longer delay for the residual contribution. By contrast, $\beta_2$ controls the EMA of the second moment estimate with respect to $\boldsymbol{y}_t$ (rather than directly with $\boldsymbol{x}_t$; see `line 6` of Algorithm 12).

For `Adam` it is common to analyze in theory the case, when $\beta_2 = 1 - 1/T$ (Reddi et al., 2019; Zaheer et al., 2018; Chezhegov et al., 2024), i.e., the choice of the "optimal" $\beta_2$ parameter depends on the length of training. Which, presumably, is also the case for `SF-AdamW`, making it not fully schedule-free. Hägele et al. (2024) observed this sensitivity to beta parameters, and we go beyond this ablation also (Figure 31).

Importantly, the authors mention that disabling gradient norm clipping is crucial for schedule-free runs; however, we do not observe this in practice and instead find the opposite effect—see Figure 28.

---

**Algorithm 12** `SF-AdamW`

---

1: **Input:** Initial parameters $\boldsymbol{x}_0$, number of iterations $T$, learning rate $\gamma$, weight decay $\lambda$, $\beta_1$, $\beta_2$, warmup iterations $T_{\text{warmup}}$, $\varepsilon$.
2: **Initialize:** $\boldsymbol{z}_0 \leftarrow \boldsymbol{x}_0$, $\boldsymbol{v}_0 \leftarrow \boldsymbol{0}$
3: **for** $t \in [T]$ **do**
4: $\quad \boldsymbol{y}_t \leftarrow (1 - \beta_1)\boldsymbol{z}_t + \beta_1 \boldsymbol{x}_t$
5: $\quad \boldsymbol{g}_t \leftarrow \nabla \mathcal{L}(\boldsymbol{y}_t, \boldsymbol{\xi}_t)$
6: $\quad \boldsymbol{v}_t \leftarrow \beta_2 \boldsymbol{v}_{t-1} + (1 - \beta_2)\boldsymbol{g}_t \odot \boldsymbol{g}_t$
7: $\quad \gamma_t \leftarrow \gamma \sqrt{1 - \beta_2^t} \min\{1, t/T_{\text{warmup}}\}$
8: $\quad \boldsymbol{z}_{t+1} \leftarrow \boldsymbol{z}_t - \gamma_t \left(\boldsymbol{g}_t / (\sqrt{\boldsymbol{v}_t} + \varepsilon) + \lambda \boldsymbol{y}_t\right)$
9: $\quad c_{t+1} \leftarrow \frac{\gamma_t^2}{\sum_{i=0}^{t} \gamma_i^2}$
10: $\quad \boldsymbol{x}_{t+1} \leftarrow (1 - c_{t+1})\boldsymbol{x}_t + c_{t+1}\boldsymbol{z}_{t+1}$
11: **end for**
12: **Return:** $\boldsymbol{x}_T$

---

**Prodigy.** Mishchenko & Defazio (2024) extended the `D-Adaptation` framework. Drawing inspiration from the `AgaGrad` (Duchi et al., 2011) theory, the authors derived an alike step-size rule, giving rise to a new family of methods. While studying the convergence (in the deterministic case) of several proposed algorithms that are based on the gradient descent and dual averaging, the authors also introduced an `Adam`-like version of their methods—the `Prodigy` optimizer (Algorithm 13)—that effectively removes the need for hand-tuned learning rates through an intrinsic, adaptive step-size scheme. The EMA of `Prodigy` specifically includes $d_t \boldsymbol{g}_t$ sequence rather than the raw gradients $\boldsymbol{g}_t$ (see `lines 5, 6, 8, 9` of Algorithm 13). The new term $d_t$ is determined by two additional EMA sequences, which are also responsible for the adaptive rescaling of the learning rate according to `line 10`. Mishchenko & Defazio (2024) evaluate `Prodigy` in practice on language models by running a shallow nanoGPT transformer on the Shakespeare (over-training regime) and BookWiki datasets. We extend the experiments with `Prodigy` to a larger scale and a greater variety of LLM pretraining settings.

Crucially, `Prodigy` does not require extensive learning rate tuning. Typically, we initialize $\gamma = 1$, as suggested by the authors, and it remains remarkably stable, as demonstrated in our $\gamma$-sweeps (Figures 6 and 18). However, `Prodigy` is still be compatible with learning rate schedules, which we verify experimentally (Figures 8 and 20). We further show that, without any schedulers, $d_t$ sequence behaves similarly to the constant learning rate with warmup (see Figure 35 and related ablations). Moreover, `Prodigy` scales reliably similar to `AdamW`, making it a promising choice for future development of parameter-free methods.

C.5 `MARS`

Very recently, Yuan et al. (2024) introduced `MARS`—a family of optimizers incorporating modern adaptive (Loshchilov & Hutter, 2019; Chen et al., 2023) and approximate second-order methods (Gupta et al., 2018) methods with a variance reduction update style.

This optimization framework gave a rise to: `MARS-AdamW`, our main baseline, which we call simply `MARS`; `MARS-Lion`; and `MARS-Shampoo`. We mainly include `MARS-AdamW` in our ablation studies, but also report results for the other two optimizers (see Figure 34).

---

**Algorithm 13** `Prodigy`

---

1: **Input:** Initial parameters $\boldsymbol{x}_0$, number of iterations $T$, learning rate $\gamma$, weight decay $\lambda$, $\beta_1$, $\beta_2$, $\varepsilon$.
2: **Initialize:** $d_0 \leftarrow 10^{-6}$, $\gamma \leftarrow 1$, $\boldsymbol{m}_0 \leftarrow \boldsymbol{0}$, $\boldsymbol{v}_0 \leftarrow \boldsymbol{0}$, $r_0 \leftarrow 0$, $\boldsymbol{s}_0 \leftarrow \boldsymbol{0}$ ▷ Optional: use scheduler on $\gamma$
3: **for** $t \in [T]$ **do**
4: $\quad \boldsymbol{g}_t \leftarrow \nabla\mathcal{L}(\boldsymbol{x}_t, \boldsymbol{\xi}_t)$
5: $\quad \boldsymbol{m}_t \leftarrow \beta_1 \boldsymbol{m}_{t-1} + (1-\beta_1)d_t \boldsymbol{g}_t$
6: $\quad \boldsymbol{v}_t \leftarrow \beta_2 \boldsymbol{v}_{t-1} + (1-\beta_2)d_t^2 \boldsymbol{g}_t \odot \boldsymbol{g}_t$
7: $\quad \gamma_t \leftarrow \gamma\sqrt{1-\beta_2^t}/(1-\beta_1^t)$ ▷ Optional: use bias correction
8: $\quad r_t \leftarrow \sqrt{\beta_2}r_{t-1} + (1-\sqrt{\beta_2})\gamma_t d_t^2 \langle \boldsymbol{g}_t, \boldsymbol{x}_0 - \boldsymbol{x}_t \rangle$
9: $\quad \boldsymbol{s}_t \leftarrow \sqrt{\beta_2}\boldsymbol{s}_{t-1} + (1-\sqrt{\beta_2})\gamma_t d_t^2 \boldsymbol{g}_t$
10: $\quad d_{t+1} \leftarrow \max\left\{d_t, \frac{r_t}{\|\boldsymbol{s}_t\|_1}\right\}$
11: $\quad \boldsymbol{x}_{t+1} \leftarrow \boldsymbol{x}_t - \gamma_t d_t \left(\boldsymbol{m}_t/\left(\sqrt{\boldsymbol{v}_t} + d_t\varepsilon\right) + \lambda\boldsymbol{x}_t\right)$
12: **end for**
13: **Return:** $\boldsymbol{x}_T$

---

The authors modified the variance reduction update by introducing a scaling parameter $\eta$, which we call variance reduction scaling in the outlined algorithms and experiments. This parameter controls the scale of gradient correction—see `line 5` of Algorithms 14, 15, and 16.

Importantly, we follow only the approximate scheme of `MARS`-like optimizers, i.e., we evaluate the gradient $\boldsymbol{g}_t$ in different stochasticity, meaning that

$$\boldsymbol{g}_t = \nabla\mathcal{L}\left(\boldsymbol{x}_t, \boldsymbol{\xi}_t\right),$$
$$\boldsymbol{g}_{t-1} = \nabla\mathcal{L}\left(\boldsymbol{x}_{t-1}, \boldsymbol{\xi}_{t-1}\right).$$

In the same spirit as for `SOAP` and `Muon`, the authors use `MARS`-like algorithms for layers with two or more dimensions. *For 1D layers, embeds, scalar parameters and the final layer of neural network, this method utilizes* `AdamW`. *This design choice enables efficient and fast training with* `MARS`. Following the practices from the original work, we also use `MARS` only for 2D layers. Importantly, for `MARS`-based methods, one need to tune both the `AdamW`'s learning rate $\gamma_t^{\mathtt{A}}$, and the learning rate for 2D parameters, which we denote as $\gamma_t^{\mathtt{M}}$ for compatibility with the `Muon` pseudocode 8.

**MARS (`MARS-AdamW`).** For the `AdamW`-like algorithm, the difference occurs in the computation of $\boldsymbol{m}_t$ and $\boldsymbol{v}_t$, where the variance reduction update $\boldsymbol{c}_t$ is used instead of the gradient.

---

**Algorithm 14** `MARS` (`MARS-AdamW`)

---

1: **Input:** Initial parameters $\boldsymbol{x}_0$, number of iterations $T$, learning rate $\gamma_t$, weight decay $\lambda$, $\beta_1$, $\beta_2$, variance reduction scaling $\eta$, $\varepsilon$.
2: **Initialize:** $\boldsymbol{m}_0 \leftarrow \boldsymbol{0}$, $\boldsymbol{v}_0 \leftarrow \boldsymbol{0}$
3: **for** $t \in [T]$ **do**
4: $\quad \boldsymbol{g}_t \leftarrow \nabla\mathcal{L}(\boldsymbol{x}_t, \boldsymbol{\xi}_t)$
5: $\quad \boldsymbol{c}_t \leftarrow \boldsymbol{g}_t + \eta\frac{\beta_1}{1-\beta_1}\left(\boldsymbol{g}_t - \boldsymbol{g}_{t-1}\right)$
6: $\quad$ **if** $\|\boldsymbol{c}_t\|_2 > 1$ **then**
7: $\quad\quad \boldsymbol{c}_t \leftarrow \boldsymbol{c}_t/\|\boldsymbol{c}_t\|_2$
8: $\quad$ **end if**
9: $\quad \boldsymbol{m}_t \leftarrow \beta_1 \boldsymbol{m}_{t-1} + (1-\beta_1)\boldsymbol{c}_t$
10: $\quad \boldsymbol{v}_t \leftarrow \beta_2 \boldsymbol{v}_{t-1} + (1-\beta_2)\boldsymbol{c}_t \odot \boldsymbol{c}_t$
11: $\quad \hat{\boldsymbol{m}}_t \leftarrow \boldsymbol{m}_t/(1-\beta_1^t), \quad \hat{\boldsymbol{v}}_t \leftarrow \boldsymbol{v}_t/(1-\beta_2^t)$
12: $\quad \boldsymbol{x}_{t+1} = \boldsymbol{x}_t - \gamma_t\left(\frac{\hat{\boldsymbol{m}}_t}{\sqrt{\hat{\boldsymbol{v}}_t}+\varepsilon} + \lambda\boldsymbol{x}_t\right)$
13: **end for**
14: **Return:** $\boldsymbol{x}_T$

---

We point out once again that, for LLM training, we only run Algorithm 14 for 2D parameters, resulting in the following two updates at each iteration:

$$\boldsymbol{x}_{t+1}^{\texttt{M}} \leftarrow \texttt{MARS}\left(\boldsymbol{x}_t^{\texttt{M}}, \gamma_t^{\texttt{M}}, \lambda^{\texttt{M}}, \beta_1^{\texttt{M}}, \beta_2^{\texttt{M}}, \varepsilon, T = 1\right) \quad \text{for 2D parameters},$$

$$\boldsymbol{x}_{t+1}^{\texttt{A}} \leftarrow \texttt{AdamW}\left(\boldsymbol{x}_t^{\texttt{A}}, \gamma_t^{\texttt{A}}, \lambda^{\texttt{A}}, \beta_1^{\texttt{A}}, \beta_2^{\texttt{A}}, \varepsilon, T = 1\right) \quad \text{for 1D parameters},$$

i.e., in the same way as in Algorithms 8 and 10. The same holds for two more versions—`MARS-Lion` and `MARS-Shampoo`, which we discuss below.

**MARS-Lion.** Similarly to the `Lion` algorithm, the authors use scaled gradient correction $\boldsymbol{c}_t$ with the current gradient. Importantly, Algorithm 15 does not leverage second moment estimates to update 2D parameters. Instead, the updates rely on the sign-based characteristic of `Lion` integrated with the variance reduction framework.

---

**Algorithm 15** `MARS-Lion`

---

1: **Input:** Initial parameters $\boldsymbol{x}_0$, number of iterations $T$, learning rate $\gamma_t$, weight decay $\lambda$, $\beta_1$, variance reduction scaling $\eta$, $\varepsilon$.
2: **Initialize:** $\boldsymbol{m}_0 \leftarrow \boldsymbol{0}$, $\boldsymbol{v}_0 \leftarrow \boldsymbol{0}$
3: **for** $t \in [T]$ **do**
4: $\quad \boldsymbol{g}_t \leftarrow \nabla\mathcal{L}(\boldsymbol{x}_t, \boldsymbol{\xi}_t)$
5: $\quad \boldsymbol{c}_t \leftarrow \boldsymbol{g}_t + \eta\frac{\beta_1}{1-\beta_1}(\boldsymbol{g}_t - \boldsymbol{g}_{t-1})$
6: $\quad$ **if** $\|\boldsymbol{c}_t\|_2 > 1$ **then**
7: $\quad\quad \boldsymbol{c}_t \leftarrow \boldsymbol{c}_t/\|\boldsymbol{c}_t\|_2$
8: $\quad$ **end if**
9: $\quad \boldsymbol{m}_t \leftarrow \beta_1 \boldsymbol{m}_{t-1} + (1-\beta_1)\boldsymbol{c}_t$
10: $\quad \boldsymbol{x}_{t+1} = \boldsymbol{x}_t - \gamma_t\left(\texttt{sign}\left(\boldsymbol{m}_t\right) + \lambda\boldsymbol{x}_t\right)$
11: **end for**
12: **Return:** $\boldsymbol{x}_T$

---

**MARS-Shampoo.** The same holds for `MARS-Shampoo`. A key point to note is that, to compute SVD of the first moment estimate, the authors also perform the Newton-Schulz steps (Bernstein & Newhouse, 2024; Higham, 2008). In our experiments, we use 5 iterations of this orthogonalization scheme for `MARS-Shampoo`.

---

**Algorithm 16** `MARS-Shampoo`

---

1: **Input:** Initial parameters $\boldsymbol{x}_0$, number of iterations $T$, learning rate $\gamma_t$, weight decay $\lambda$, $\beta_1$, variance reduction scaling $\eta$, $\varepsilon$.
2: **Initialize:** $\boldsymbol{m}_0 \leftarrow \boldsymbol{0}$, $\boldsymbol{v}_0 \leftarrow \boldsymbol{0}$
3: **for** $t \in [T]$ **do**
4: $\quad \boldsymbol{g}_t \leftarrow \nabla\mathcal{L}(\boldsymbol{x}_t, \boldsymbol{\xi}_t)$
5: $\quad \boldsymbol{c}_t \leftarrow \boldsymbol{g}_t + \eta\frac{\beta_1}{1-\beta_1}(\boldsymbol{g}_t - \boldsymbol{g}_{t-1})$
6: $\quad \boldsymbol{m}_t \leftarrow \beta_1 \boldsymbol{m}_{t-1} + (1-\beta_1)\boldsymbol{c}_t$
7: $\quad \boldsymbol{U}_t, \boldsymbol{\Sigma}_t, \boldsymbol{V}_t \leftarrow \texttt{SVD}(\boldsymbol{m}_t)$ $\qquad\qquad\qquad$ ▷ Use Newton-Schulz orthogonalization
8: $\quad \boldsymbol{x}_{t+1} = \boldsymbol{x}_t - \gamma_t\left(\boldsymbol{U}_t\boldsymbol{V}_t^\top + \lambda\boldsymbol{x}_t\right)$
9: **end for**
10: **Return:** $\boldsymbol{x}_T$

---

## D  IMPLEMENTATION

Our code is based on an extension of nanoGPT[4] and uses PyTorch (Paszke et al., 2019) as well as FlashAttention (Dao et al., 2022). We incorporate mixed-precision training (Micikevicius et al., 2018), i.e., we train in `bfloat16` precision, except for normalization modules and softmax which we train in `float32`. The optimizer states are also stored in `float32`. The majority of experiments were performed using a cluster of A100-SXM4-80GB, H100-HBM3-80GB GPUs as well as

---

[4]https://github.com/karpathy/nanoGPT

GH200-120GB. We trained both in a single GPU regime and in DDP (Li et al., 2020) (from 2 to 8 GPUs per one run). We estimate that the full cost of all experiments for our project to roughly 30000 GPU hours. To give an idea of how much effort was put into tuning each method, across all model sizes, batches and iterations, we trained a total of 2900 models. This includes includes *nearly*: 750 AdamW, 145 ADOPT, 238 AdEMAMix, 158 Lion, 165 Signum, 231 Muon, 135 D-Muon, 354 SOAP, 199 Sophia, 133 SF-AdamW, 195 Prodigy, 217 MARS-AdamW, 26 MARS-Lion, and 20 MARS-Shampoo models. See Appendix G for details about hyperparameter tuning.

# E    MODEL & DATA

**Architecture details.** In our project, we use the Llama-like family of models (Llama Team, 2024). We implement the popular in the community decoder-only transformer with SwiGLU activation functions (Shazeer, 2020), RoPE embeddings (Su et al., 2023), RMSNorm (Zhang & Sennrich, 2019). The vocabulary is based on the GPT2 (Radford et al., 2019) tokenizer [5] and contains 50304 tokens. Importantly, our variant of the Llama-based architecture employs weight tying (Press & Wolf, 2017).

The number of parameters in our models is fully configurable, and we present the exact configurations used in our experiment in Table 1.

Table 1: **Configurations for our Llama-like models.**

| # Parameters | 124M | 210M | 583M | 720M |
|---|---|---|---|---|
| Hidden size | 768 | 768 | 1920 | 2048 |
| # Attention heads | 12 | 12 | 15 | 16 |
| # Layers | 12 | 24 | 11 | 12 |
| Init. std | 0.02 | 0.02 | 0.02 | 0.02 |
| Use bias | no | no | no | no |
| RMSNorm epsilon | 0.00001 | 0.00001 | 0.00001 | 0.00001 |
| Positional encoding | RoPE | RoPE | RoPE | RoPE |

**Dataset.** Our main findings are obtained on the subset of FineWeb (Penedo et al., 2024a) with 100**B** tokens [6], cleaned and deduplicated corpus for LLM pretraining, which we split into train and validation sequences. During training, we evaluate the models with a fixed set of 32 batches of our chosen sequence length (512 for almost all experiments, the same context length as training) to establish the validation loss curves. At the end of the training, we compute the full validation loss and perplexity (this loss is reported as Final Validation Loss in the figures). We also performed our initial results on the subset of the OpenWebText2 dataset (Gao et al., 2020b).

# F    ADDITIONAL RESULTS

In this section, we complement our results from the main part with extended experiments. We start sequentially with smaller models of 124**M** and 210**M** parameters, ablating: **warmup**, **weight decay**, **learning rate schedulers**, **gradient norm patterns**, **learning rate decaying**, and other optimizer-related phenomena. We finalize this section with the **wall-clock performance of optimizers**. Details on hyperparameter searches are provided in Appendix G.

## F.1    ABLATIONS FOR 124M MODEL

At first, we systematically gather all ablations with 124**M** parameter models. As in § 4.1, we study: the effect of scaling the number of iterations and hyperparameter dependence on $T$; warmup; the importance of weight decay for optimizers; $\gamma$-sensitivity; a comparison of $\gamma$-schedulers; gradient norm patters during training; learning rate decaying; and optimizer-specific phenomena for Sophia, SF-AdamW, Prodigy, Muon, Signum, and MARS-based methods.

---

[5]https://github.com/openai/tiktoken

[6]https://huggingface.co/datasets/HuggingFaceFW/fineweb

**Complementing benchmarking results of 124M models.** In addition to results from the main part, we show the dynamics of the validation loss in Figure 12. The presented runs correspond to those in Figure 3.

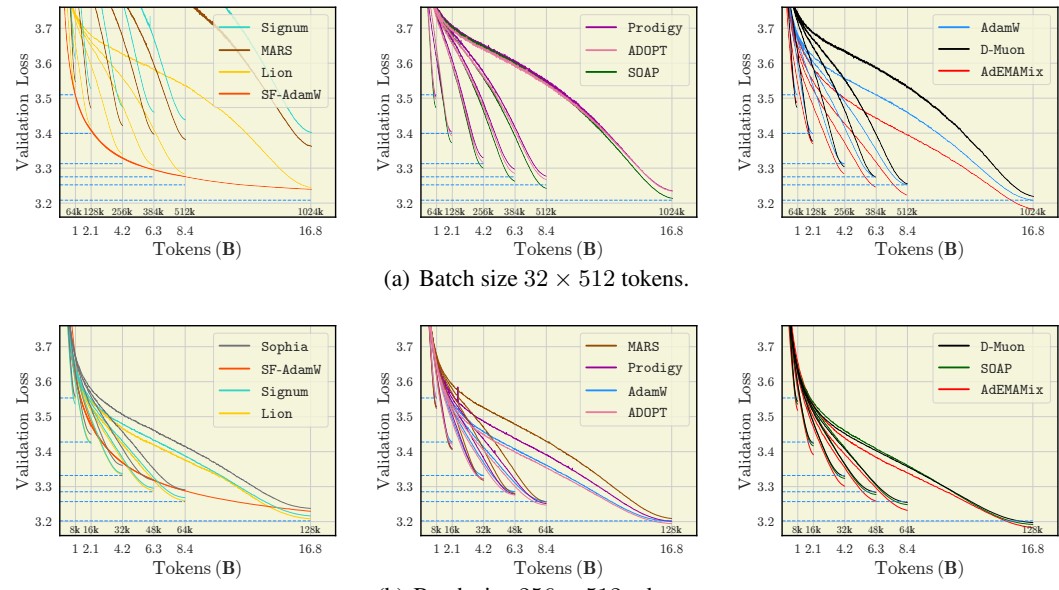

(a) Batch size $32 \times 512$ tokens.

(b) Batch size $256 \times 512$ tokens.

Figure 12: **Comparing optimizers for training a 124M parameter LLM.** We plot the *validation loss* dynamics obtained by considered optimizers. In **(a)**, we train methods with a "small" batch size of 16k tokens for $\{64, 128, 256, 384, 512, 1024\}$k iterations. In **(b)**, we train methods with nearly $8\times$ larger batch size of 131k tokens for $\{8, 16, 32, 48, 64, 128\}$k iterations. Thus, in both settings, we result in the same number of tokens models see during the training: $\{1, 2.1, 4.2, 6.3, 8.4, 16.8\}$**B**. We observe that: (I) many methods outperform AdamW in the short runs for 1**B** or 2.1**B** tokens; (II) as training on more tokens, AdamW narrows the gap with SOAP and D-Muon, while AdEMAMix emerges as the best-performing method; (III) Signum, MARS, Lion, Prodigy benefit from the increased batch size.

**Scaling the number of iterations.** We stay in the setup from § 3, training 124**M** model with batches of $256 \times 512$ tokens. Our training runs in Figure 12 **(b)** demonstrate the the gap between SOAP and AdEMAMix narrows as the training horizon extends. As for 124**M** model, we tuned all optimizers on 2.1**B** tokens length of training, some hyperparameters, particularly those sensitive to the training duration, may become suboptimal for longer runs of 16.8**B** tokens. For example, the beta parameter ($\beta_2$) of the second moment estimate in AdamW-like methods should arguably be re-tuned when the number of iterations is increased (Orvieto & Gower, 2025; Marek et al., 2025; Busbridge et al., 2023), which also makes theoretical claims (Reddi et al., 2019; Défossez et al., 2022; Wang et al., 2023; Zaheer et al., 2018; Taniguchi et al., 2024; Chezhegov et al., 2024). Our extensive tuning on 2.1**B** yielded the result that for AdamW, SOAP, and AdEMAMix optimizers, $\beta_2$ should be set to 0.999. Importantly, Pagliardini et al. (2024) suggest increasing $\beta_3$ of AdEMAMix, which controls the slow EMA (see line 7 of Algorithm 3), for longer training.

As such, we conducted two experiments.

First, we keep the best hyperparameters found for 2.1**B** tokens horizon, and extend them to $2\times$ loner duration than the maximum one (16.8**B** tokens), resulting in a total of 33.6**B** tokens—it is interesting to observe whether the gap between SOAP and AdEMAMix finally closes in a longer run. Secondly, we re-tune beta parameters of SOAP and AdEMAMix for 16.8**B** and 33.6**B** runs, and compare results. Our re-tuned values are $\beta_2 = 0.9999$ for SOAP, and $\beta_3 = 0.9999$ for AdEMAMix.

This ablation is described in Figure 13 **(a,b)**. We see that, indeed, without re-tuning, SOAP ends up outperforming AdEMAMix when extending the training horizon further to 33.6**B** tokens ($\equiv 256$**k** iterations). However, with re-tuning of $\beta_2$ for SOAP and $\beta_3$ for AdEMAMix, the latter optimizer

still takes the lead. Notably, in our experiments, $\beta_3 = 0.999$ is only better than $\beta_3 = 0.9999$ when the number of training iterations is less than 32**k**. Surprisingly, given the many theoretical claims from works analyzing Adam, that $\beta_2$ depends on the number of iterations: $\beta_2 = \beta_2(T) = 1 - 1/T$, we do not observe this to be a common rule in practice, as many influential settings regarding LLM pretraining (DeepSeek-AI, 2024b; Brown et al., 2020; Touvron et al., 2023; Team OLMo, 2024b) utilize a typical ($\beta_2 = 0.95$) even for very long training for trillions of tokens. Therefore, we highlight this oversight in Takeaway 9, proposing to re-tune $\beta_2$ hyperparameter of Adam-like methods when changing of the training horizon.

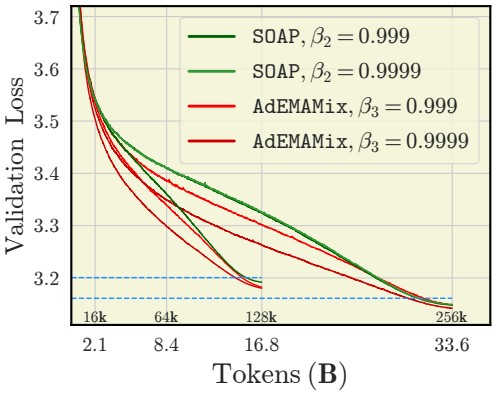 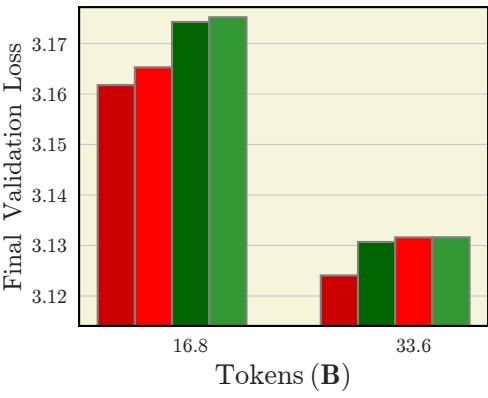

(a) Training dynamics with different $\beta_3$.  (b) Increase $\beta_3$ for AdEMAMix for longer training.

Figure 13: **Re-tuning beta parameters is significant for longer training.** This experimental setup coincides with the one from Figure 4—where we consider an impact of increasing batch size or number of iterations. We elaborate more on the impact of beta parameters for AdEMAMix: in **(a)**, we show a training dynamics of AdEMAMix and SOAP for $T \in \{128, 256\}$**k** steps, and **(b)** demonstrates the final loss. Throughout this experiment we use a batch size of $256 \times 512$ tokens and train a 124**M** model. Our results reveal that increasing $\beta_3$ for AdEMAMix is crucial for long runs. Without these changes, SOAP with $\beta_2 = 0.999$ found via tuning on 16**k** steps runs ends up outperforming AdEMAMix with $\beta_3 = 0.999$. Re-tuning $\beta_2$ of SOAP does not change the results much, and give almost identical loss curves, however, the training dynamics if AdEMAMix changes dramatically with $\beta_3$ as noticed in (Pagliardini et al., 2024), and gives best results with larger $\beta_3 = 0.9999$.

> **Takeaway 9.** We highlight the overlooked claim that $\beta_2$ parameters of Adam-like methods should be re-tuned with training durations. One needs to increase $\beta_2$ for longer training. This re-tuning significantly improves the optimizer performance.

**Warmup ablation.** In this section, we supplement the experiments on warmup from § 4.1. We study the impact of warmup on the final validation loss. Replicating our setup (§ 3), we use the batch size of $256 \times 512$ tokens and reuse the best hyperparameters found through tuning, except for $T_{\text{warmup}}$. For all methods, we sweep over $T_{\text{warmup}} \in \{1.56\%, 6.25\%, 25\%\}$ of the total training duration $T$ to examine each method's sensitivity to warmup. Additionally, for AdamW, we extend this sweep to $T_{\text{warmup}} \in \{1.56\%, 5\%, 6.25\%, 10\%, 25\%\}$ of $T$. We specifically consider $1.56\%$ and $6.25\%$ percentages because the former represents a typical number of warmup steps (2000) for models of our scale, while the latter ($6.25\%$ of 128000 steps) aligns with the warmup strategy used in Llama (Llama Team, 2024).

Contrary to the insights from (Zhang et al., 2024a), we observe that $25\%$ of the Cinchilla optimal duration (620**M** tokens for 124**M** model) is far from being the best warmup for pretraining. We emphasize that their results were obtained for 85**M** models and then extrapolated to larger scales. However, in our setting, we found that the basic 2000 steps were a more suitable warmup option for most optimizers; exceptions include sign-based methods (Signum, Lion), and Sophia with SF-AdamW.

We provide the warmup sweep for AdamW in Figure 14.

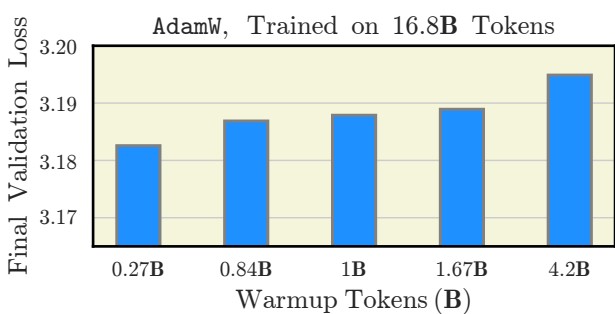

Figure 14: **Warmup sweep for `AdamW`.** We observe that the smaller yet reasonable warmup value is the best. However, this is not the case for other methods like `Signum`, `Lion`, `Sophia`, and `SF-AdamW`—see Figure 7.

> **Takeaway 10.** As usual, a warmup duration in LLM pretraining is around 2k steps. However, we reveal that the warmup duration is optimizer-dependent and should be tuned: for `SF-AdamW`, `Sophia`, and `Signum`, longer warmup results in improved final performance, while `Lion` with increased warmup also surpasses strong baselines such as `AdamW`.

**Weight decay ablation.** Prior work analyzing the importance of weight decay $\lambda$ in model training suggests tuning both $\lambda$ and the learning rate $\gamma$ so that their product $\lambda\gamma$ remains constant. D'Angelo et al. (2024) argue that, across different pairs of $\gamma$ and $\lambda$, the lowest error of the model is observed along the contour in the hyperparameter space where $\lambda\gamma = $ const. The authors also establish a connection between the quantity of $\lambda\gamma$ product and an effect of regularization and noise scale in the over-training regime, such as for small computer vision models trained over multiple epochs. Kosson et al. (2024b) highlight that if $\gamma$ and $\lambda$ are chosen to result in constant product, the model achieves the same equilibrium rotation value, reflecting a similar effective update size for the weights. While previous studies have analyzed the rule of keeping $\lambda\gamma = $ const primarily on image classification tasks with ResNet-based models (He et al., 2015), Pagliardini et al. (2024) also used this heuristic when tuning hyperparameters for LLM pretraining.

In our study, which focuses solely on language modelling, we demonstrate that using a relatively large weight decay term with a fixed learning rate can significantly accelerate short training runs. Throughout our weight decay ablation experiments, we fix the best value of $\gamma$ found via tuning on near-Chinchilla optimal $T$, and sweep the weight decay across $\lambda \in \{0, 0.1, 0.5\}$, where $\lambda = 0.1$ is the standard value of the decoupled weight decay term in our work. Our results are consistent across optimizers and training horizons: runs with large $\lambda$ dominate for a small number of iterations, but as the training length increases to $\{8.4, 16.8\}$B tokens, runs with a moderate $\lambda = 0.1$ begin to outperform (Figures 5 and 15). An important example is `Muon`. As this optimizer does not use weight decay for 2D parameters, we observe that runs with $\lambda = 0.5$ underperform those with $\lambda \in \{0, 0.1\}$ even in short training on $\{1, 2.1, 4.2, 6.3\}$B tokens. However, when we consider an implementation of the `D-Muon` optimizer with learning rate and weight decay shared across all parameters, we again observe a similar pattern to that seen with other methods—larger weight decay dominates when training on fewer tokens.

We highlight these observations for practitioners and suggest that this approach may be useful for short training runs. Our main claim from this section is summarized in Takeaway 11.

**Impact of weight decay on `Muon`.** We complement our claims regarding the old implementation of `Muon` where weight decay has not been applied to matrix parameters. Through comparison with modified `D-Muon` Liu et al. (2025), we clearly demonstrate the impact of the enabled weight decay for longer training in Figures 16 and 17.

> **Takeaway 11.** The use of weight decay, particularly a large decoupled weight decay term (0.5 and above), can significantly impact the final loss value and optimizer behavior. However, for extended training horizons, a moderate, non-zero weight decay of 0.1 proves to be a robust option.

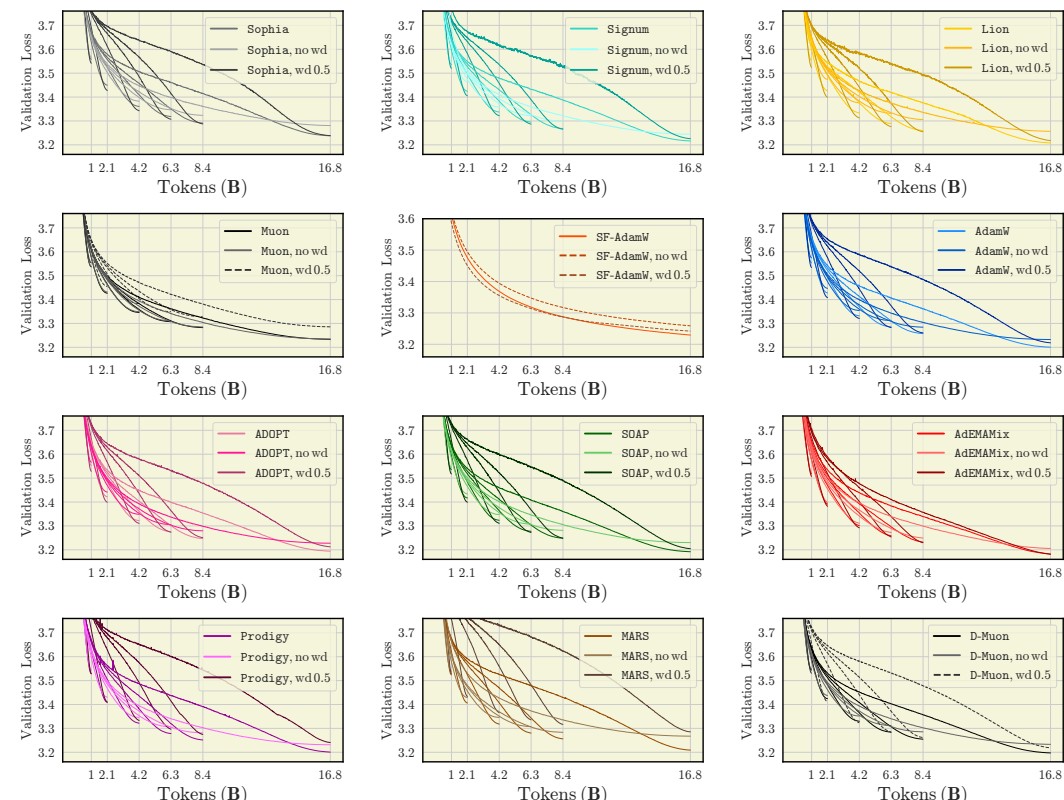

Figure 15: **Larger weight decay achieves significantly better results when training on fewer tokens.** We observe that the majority of runs with the large weight decay of 0.5 consistently outperform those with weight decay of 0.1 for all training durations except for the long training on 16.8**B** tokens. Notably, Signum and Lion with large weight decay perform even better than AdamW with the same learning rate—see Figure 5. We also consider a setting without weight decay. We observe that this is suboptimal for most of other optimizers, while the typical weight decay of 0.1 remains the best for long training durations. An interesting pattern emerges for optimizers that treat one-dimensional and two-dimensional parameters differently, such as Muon and MARS. For these, runs with large weight decay (0.5) consistently underperform those with 0.1 and, in some cases, even those without weight decay. For Muon, we attribute this effect to its algorithmic design, in which weight decay is not employed to optimize matrix parameters (see Algorithm 7), in contrast to D-Muon, where the observed patterns are reliably similar to those seen with AdamW. For MARS, we only vary the weight decay corresponding to matrix parameters while keeping 0.1 for all scalar, one-dimensional and final layer parameters. In this case, we conclude that the gap between large and small weight decay values narrows significantly faster.

**Learning rate sensitivity.** In this part of the work, we meticulously replicate the learning rate sweep process and present comprehensive results. In line with our experimental setup (§ 3), our aim is to determine the true impact of the learning rate and its transferability to longer training horizons. For each optimizer, we only vary the learning rate while maintaining the best hyperparameters obtained during our initial tuning (see Appendices G and G.1) on 2.1**B** tokens for 124**M** parameter model. That is, the learning rate has been re-tuned for all optimizers on the training length of 16.8**B** tokens. We do not present $\gamma$-sensitivity for Prodigy in the main part (§ 4) because of the difference in axis scale: we sweep across $\gamma_{\max} \in \{0.5, 1, 2, 10, 100\}$ for this optimizer. We show the results of the learning rate sweep in Figure 18.

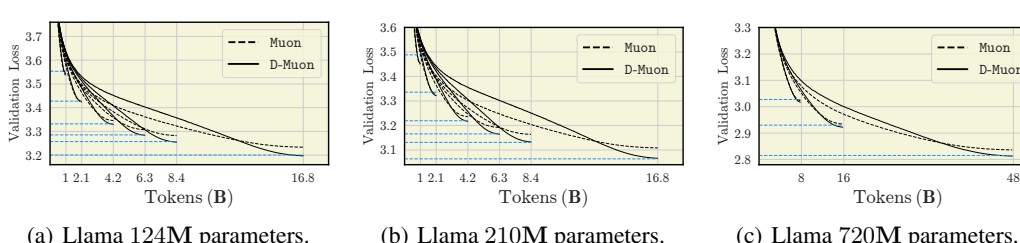

(a) Llama 124**M** parameters.   (b) Llama 210**M** parameters.   (c) Llama 720**M** parameters.

Figure 16: **Importance of weight decay for `Muon`.** We complement our weight decay ablation with comparison of two version of `Muon`: one that uses a weight decay for all parameters (`D-Muon`), and another, with weight decay being applied only to embeddings, scalar parameters, and the final layer. For three scales of models **(a,b,c)**, we show that `D-Muon` greatly outperforms the basic `Muon`. We emphasize that the main reason—weight decay, which supports our ablations in Figures 5 and 15.

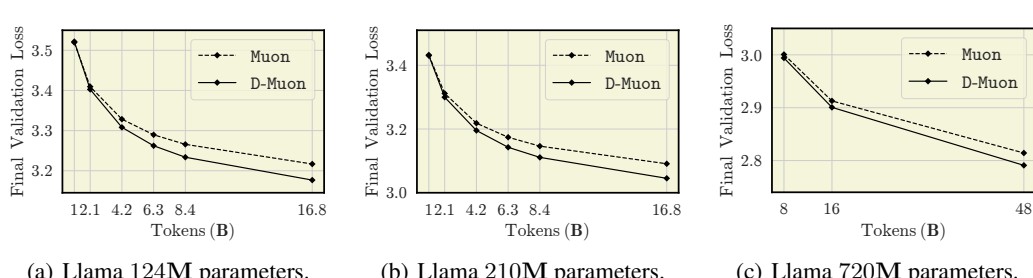

(a) Llama 124**M** parameters.   (b) Llama 210**M** parameters.   (c) Llama 720**M** parameters.

Figure 17: **Weight decay in `Muon` & `D-Muon`.** We compare two methods—basic `Muon` (Jordan et al., 2024b), and `D-Muon` (Liu et al., 2025) with a weight decay applied to all parameter groups. Across model sizes used in our benchmarking of dense LLMs, we observe a major improvement of `D-Muon` over `Muon`. We relate this observation to our ablation on the importance of weight decay across different optimizers, and training horizons. See Figure 16 and Appendix F.1 for more details.

> **Takeaway 12.** For most optimizers, the learning rate $\gamma_{\max}$ selected near the Chinchilla optimal horizon transfers smoothly to our $8\times$ longer run. Notably, we found that: (I) sign-based methods and `Sophia` diverge with larger $\gamma_{\max} = 2e^{-3}$; (II) while `SF-AdamW`, `SOAP`, and `D-Muon` achieve better performance with such a large learning rate; (III) `MARS` demonstrates a very consistent performance across $\gamma$ sweep, which is not typical for other optimizers.

**Comparison of learning rate schedulers.** In this part of our ablations, we systematically investigate the impact of $\gamma$-schedulers on optimizers. As we mention in § 3, we conduct the majority of experiments on the FineWeb dataset (Penedo et al., 2024a). However, here we also present a small ablation on another corpus for LLM pretraining—OpenWebText2 (OWT2) (Gao et al., 2020a)—as the main results of Defazio et al. (2024b) are obtained on the subset of this corpus. We show our results for two batch size settings: $32 \times 512$ for OWT2 (Figure 19), and $256 \times 512$ for FineWeb (Figure 20).

In Figure 19, we present our initial results in the small-batch setting on the OWT2 dataset (Gao et al., 2020a). We run the WSD scheduler experiments *without following* the rule of thumb from (Hägele et al., 2024); instead, use a linear decay shape during the learning rate cooldown and set $\gamma$ to the value that is near-optimal for cosine. Hence, we use $\gamma_{\max} = 0.001$ with the learning rate decay to $\gamma_{\text{end}} = 0.01 \times \gamma_{\max}$ for both cosine and WSD schedulers. This is the only experiment where we do not follow the best-practices of using WSD. Regarding hyperparameter tuning, we observe little shift compared to that found in Appendix G for FineWeb. We only pose that it may be beneficial to additionally re-tune the gradient clipping threshold, as this depends on the "cleanliness" of the dataset. Our ablations (Figure 19) reveal that `SF-AdamW` can potentially outperform the `AdamW` baseline with the WSD scheduler. However, the cosine $\gamma$-scheduler still takes the lead in this setup.

We also report the final validation loss on the FineWeb dataset (Penedo et al., 2024a) for 124**M** model trained with the batch size of $256 \times 512$ tokens. For WSD, we follow the rule of thumb

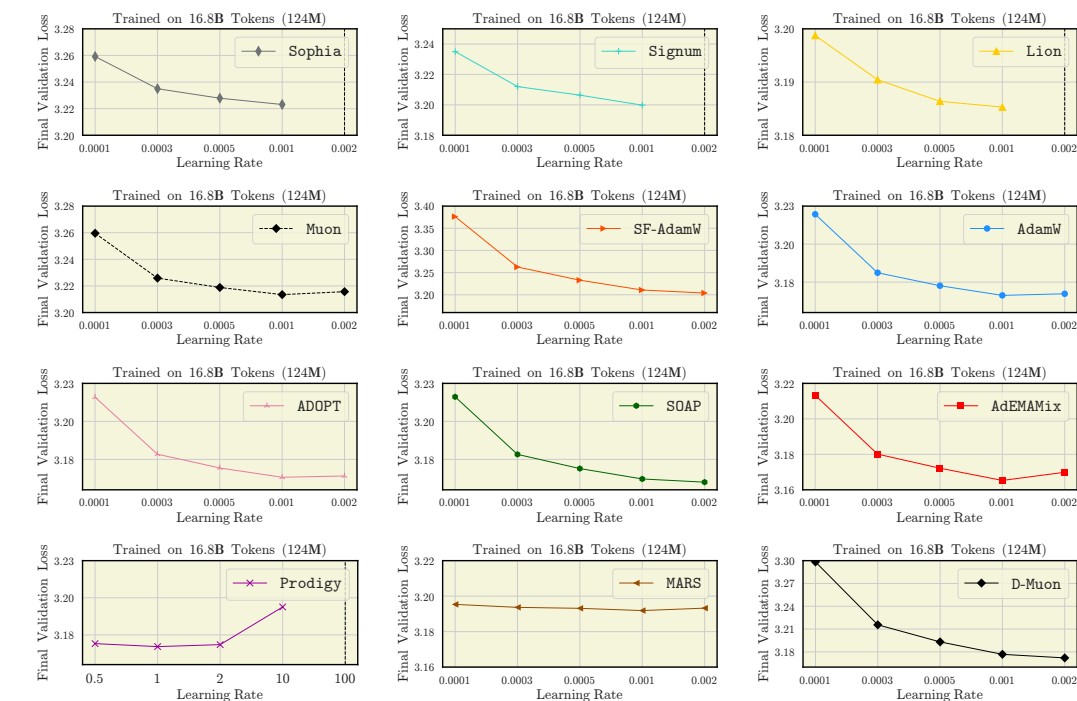

Figure 18: **Learning rate sensitivity.** In the current setting, only `SOAP`, `SF-AdamW`, and `D-Muon` reach the better performance with the large learning rate of 0.002. Conversely, `Sophia` and all sign-based methods (`Signum` and `Lion`) diverge with this learning rate value. `MARS` and `Prodigy` show a remarkably consistent performance across the learning rate sweep. And, `Prodigy` diverges for sufficiently large value of $\gamma_{max}$—see Figure 35 for more insights regarding the learning rate of `Prodigy`.

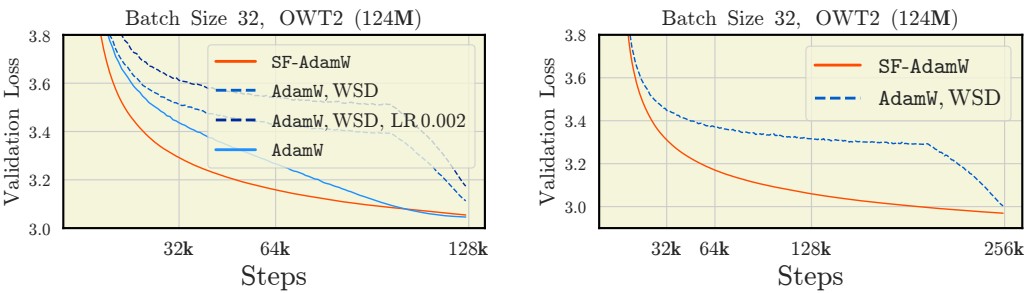

(a) Cosine scheduler outperforms WSD and `SF-AdamW`.

(b) Performance gap narrows with longer training.

Figure 19: **WSD scheduler underperforms both `AdamW` with cosine scheduler and `SF-AdamW`.** This is the only experiment we conduct on the OpenWebText2 (OWT2) dataset. We follow the small-batch setup, and replicate the best hyperparameters of each optimizer found through our tuning process. Once the learning rate and beta parameters of `SF-AdamW` and `AdamW` are properly tuned, we observe a surprisingly large performance gap between the WSD scheduler and its competitors. Figure **(b)** suggests that this gap may potentially diminish with extended training.

from Hägele et al. (2024): 20% of the steps for the cooldown, $(1 - \sqrt{x})$ decay shape, and the learning rate is half the optimal for cosine, i.e., 0.0005 if we have the best learning rate 0.001 for the method. Additionally, we point out that we do not include stochastic weight averaging (Izmailov et al., 2019) in the comparison, which might potentially enhance the overall performance. We ran the linear $\gamma$-scheduler with the same learning rates that we found through our tuning for cosine (Figures 6 and 18). We report our findings in Figure 20. All missing optimizers—`AdamW`, `Muon`, and `Sophia`—are in the main part; see Figure 8.

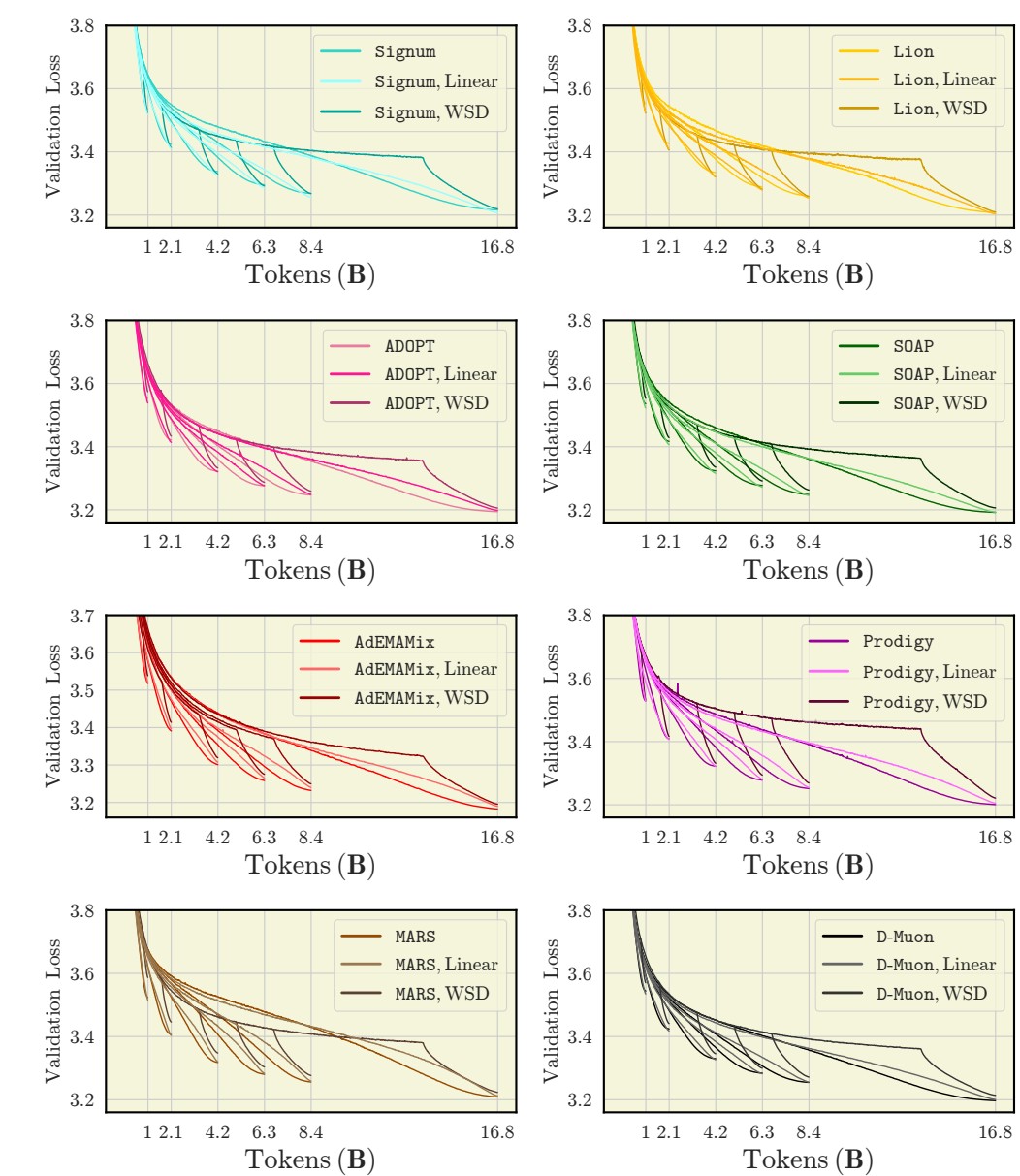

Figure 20: **Comparisons between cosine, WSD, and the linear schedulers.** We complement results in Figure 8 by extending them to all the optimizers considered in our benchmarking. In most cases, the tuned cosine baseline performs similarly to runs using the linear scheduler, with both slightly outperforming WSD. However, certain optimizers still tend to "prefer" different $\gamma$-schedulers. For example, Muon shows a preference in WSD (see Figure 8 (**a**)), AdamW performs better with the cosine scheduler, Signum and Lion appear to favor the linear scheduler. While the performance differences are not particularly large, they are still meaningful in the context of benchmarking. Therefore, we adopt the cosine scheduler as our default, as even small gaps can substantially impact our setup.

> **Takeaway 13.** A choice of the learning rate scheduler is also optimizer-related. For most methods, the cosine scheduler dominates. However, linear scheduler outperforms or matches cosine and WSD for sign-based methods, SOAP, and MARS. WSD appears to be the best option for Muon. We also study the gradient norm patterns for all optimizers and highlight it for sign-based method, who attain the completely different "bump" shape.

**Gradient norm patterns.** We systematically track the evolution of gradient norms across weight decay ($\lambda$), maximum learning rate ($\gamma_{\max}$), and learning rate scheduler sweeps in Figures 22 to 24. This analysis spans all optimizers in our benchmark, providing insight into how these hyperparameters influence gradient magnitude and stability. Our goal is to determine whether the gradient norm dynamics correlate with improved convergence and whether these trends are optimizer-specific or general. We also investigate whether deviations from expected patterns (e.g., premature flattening or explosive growth) can serve as indicators of suboptimal configuration, potentially informing better tuning heuristics.

Firstly, we study the dynamics of gradient norms while sweeping the learning rate schedulers—see Figure 22. This result complements the one in Figure 21. In general, Defazio et al. (2024a) argue that there exists an interdependence between the learning rate schedule and observed gradient norm patterns, proposing a schedule refinement for optimization algorithms. The observation that $\gamma$-scheduler can tract the gradient norm pattern and vice versa encourages us to expand experimental observations to optimizers studied in our benchmark.

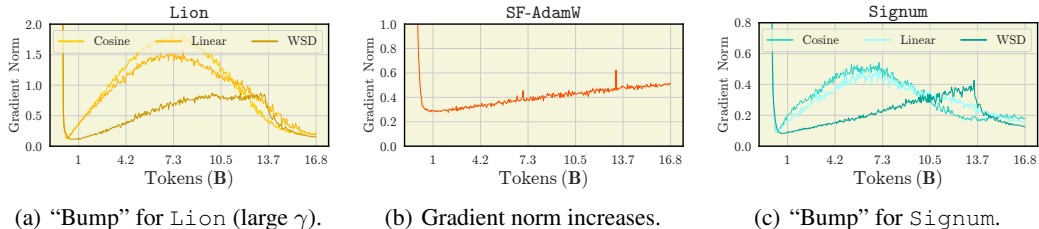

(a) "Bump" for `Lion` (large $\gamma$).  (b) Gradient norm increases.  (c) "Bump" for `Signum`.

Figure 21: **Gradient norm patterns for different schedulers.** In our experiments with learning rate schedulers (Figure 8), we also track a gradient norm changes for all optimizers, showing how it is affected by the choice of the scheduler. We use our cosine baseline, linear scheduler with the same optimal learning rate as for cosine, and the WSD scheduler with a rule of thumb described in Appendix G. We found that gradient norm evolution for majority of optimizers resembles the `SF-AdamW` pattern in **(b)**. Exceptions are sign-based methods—`Signum` **(c)**, and `Lion` **(a)**.

Prior works (Kosson et al., 2024b; Defazio, 2025) study the connection between gradient norm patterns, weight decay, and learning rate. Kosson et al. (2024b) explore how weight decay influences the update behavior of individual neurons in deep neural networks. The authors show that weight decay causes the weight norm to reach a stable equilibrium magnitude. At this equilibrium point, the opposing effects of gradient updates (which increase the norm) and weight decay (which reduce it) cancel each other out. Importantly, this study highlights the effectiveness of the decoupled weight decay for optimization over $\ell_2$ regularization, noting that the gradient norm varies between neurons or layers for $\ell_2$ regularization which is not the case for decoupled weight decay. In our experiments, we study how (decoupled) weight decay influences gradient norms—see Figure 23. We use the cosine learning scheduler and the best other hyperparameters found for optimizers, sweeping the weight decay across three critic values: $0$, the standard one of $0.1$, and the "large" weight decay of $0.5$. Basically, these gradient norms were tracked during weight decay ablation and correspond to Figures 5 and 15. We observe that runs without weight decay typically result in gradient norm curves that are more flattened and with a smaller magnitude compared to runs with $\lambda \in \{0.1, 0.5\}$. Exceptions are sign-based methods, `Muon`, `AdEMAMix`, and `Sophia`. Using the large weight decay term of $0.5$ results in a dramatic increase in the gradient norms towards the end of the training. Nevertheless, we present figures for long training runs of $7\times$ Chinchilla optimal duration for $124\mathbf{M}$ models (resp. $16.8\mathbf{B}$ tokens and $128\mathbf{k}$ steps)—where runs with $\lambda = 0.1$ outperform ones with $\lambda \in \{0, 0.5\}$—we emphasize that the same patterns of the gradient norms are also observed in shorter runs where $\lambda = 0.5$ still demonstrates the best performance.

Another key factor influencing the gradient norms is the learning rate. As with previous ablations on gradient norms (Figures 22 and 23), we follow our benchmarking setup (§ 3). During the learning rate sweep (Figures 6 and 18), we track the gradient norms presented in Figure 24. Notably, smaller learning rates result in larger gradient norm magnitudes, with exceptions for sign-based `Signum` and `Lion`. We also observe a dramatic increase in gradient norms for `Muon` with $\gamma_{\max} = 0.0001$, which we attribute to the large difference between learning rates for 1D and 2D parameters, the latter typically set around $0.01$ (see the "Learning rate `Muon`" row in Table 12). For `Prodigy` with

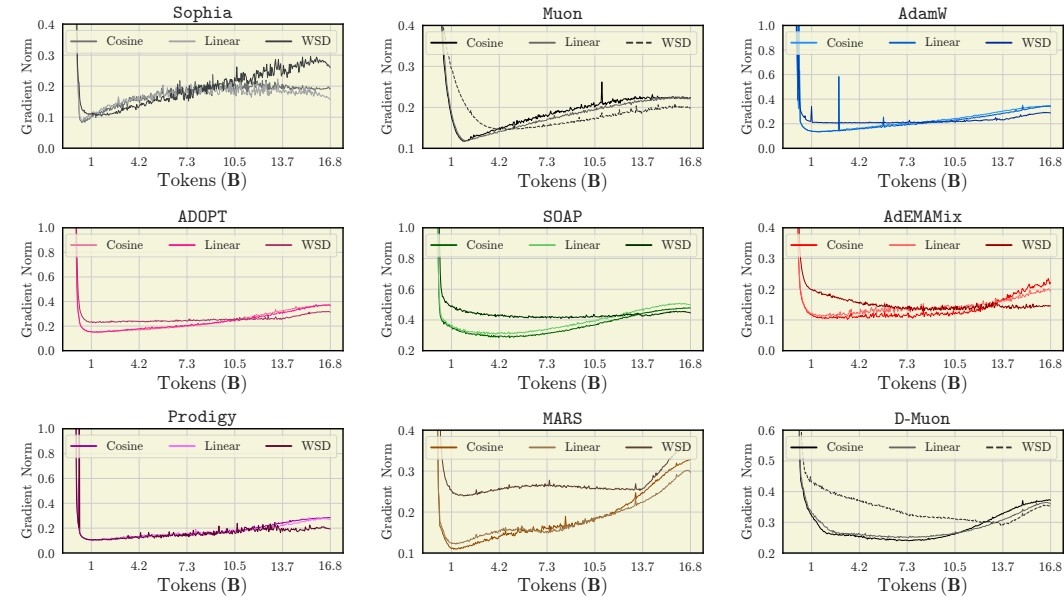

Figure 22: **Gradient norm patterns for cosine, linear, and WSD $\gamma$-schedulers.** We run all optimizers on 124M models and track the gradient norms (before clipping) for runs using different $\gamma$-schedulers. For most optimizers, we see that gradient norms tend to increase over the course of training with cosine and linear schedules. In contrast, WSD tends to produce flatter gradient norm trajectories, with consistently lower magnitudes toward the end of training compared to the other schedulers. Since the WSD scheduler maintains a constant learning rate until the cooldown phase (the final 20% of the training length), we observe a more stable gradient norm behavior in later stages. In this regard, our findings align with prior works (Kosson et al., 2024b; Defazio, 2025), which explore a connection between the learning rate schedule and gradient norm dynamics. Interestingly, Signum and Lion—see Figure 21—exhibit a pronounced drop in gradient norm during the cooldown phase, setting them apart from the other optimizers.

$\gamma_{\max} = 10$ the explosion in gradient norms might be caused by the critical value of the learning rate, which leads to divergence if increased.

> **Takeaway 14.** (I) The WSD scheduler produces stable, flat gradient norm trajectories, contrasting with the increasing norms from cosine and linear schedules. (II) The impact of weight decay is optimizer-specific, with no single value (e.g., 0 or 0.1) universally yielding optimal stability; larger decay often increases norms late in training. (III) Smaller learning rates typically lead to larger gradient norms, a trend from which sign-based methods notably deviate.

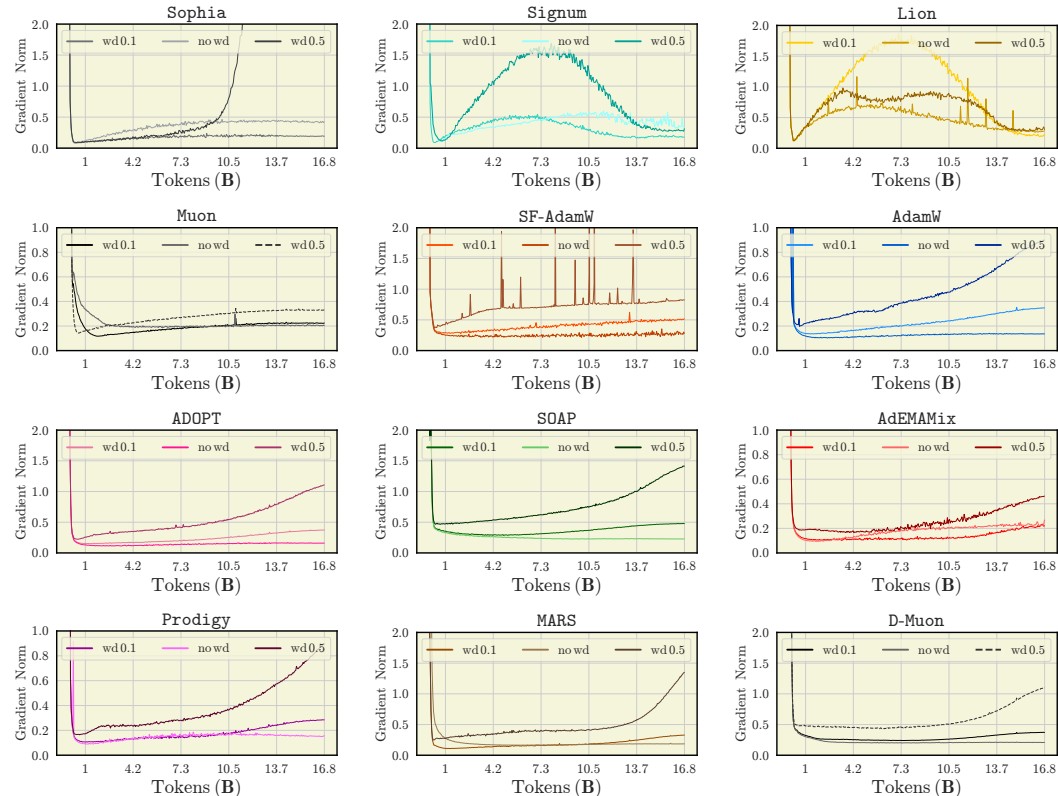

Figure 23: **Gradient norm patterns for weight decay sweep.** We complement our weight decay ablation (Figures 5 and 15) by tracking the gradient norms for all the optimizers studied in our benchmark. To highlight the effect of changing the weight decay, we use the same cosine $\gamma$-scheduler for all optimizers and keep the other best hyperparameters found, sweeping only the weight decay values as described in § 3—i.e., we fix the maximum learning rate and only change the weight decay. For `Muon`, we only sweep the weight decay for $\{$`embeds`, `scalar_params`, `lm_head`$\}$ (as in the initial implementation, the weight decay has not been applied to matrix parameters), while for `MARS`, we only sweep the weight decay of 2D parameters. Our observations reveal that, regardless of optimizer used, runs with a larger weight decay result in higher gradient norms. For `Muon`, `AdEMAMix`, `Sophia`, and sign-based methods, runs with moderate $\lambda = 0.1$ result in the most flattened and smallest gradient norms in magnitude. While for `AdamW`-like methods, `D-Muon`, `SOAP`, `Prodigy`, and `SF-AdamW`, this holds for $\lambda = 0$. We attribute the discrepancies between `D-Muon` and `Muon` to the latter's absence of weight decay for matrix parameters. As shown in Figures 5 and 15, `AdEMAMix` can benefit from large weight decay for longer training durations. Runs of `AdEMAMix` with $\lambda = 0.5$ are still outperform those with $\lambda = 0.1$. Interestingly, this is reflected in the gradient norms, as the absolute values corresponding to $\lambda = 0.5$ are much smaller than those of the respective runs of other `AdamW`-like optimizers.

**Learning rate decaying for** $124$**M model.** Prior ablation studies on $210$M models (Figure 9) demonstrated that decaying the learning rate down to $10\%$ of its maximum value underperforms compared to $0.01, 0.001 \times \gamma_{\max}$. To generalize this finding, we conduct the same ablation on a smaller $124$M model. As before, we use three $\gamma$-schedulers—cosine, linear, and WSD, utilizing the best hyperparameters for `AdamW` at this scale, training for $16.8$**B** tokens with the batch size of $256 \times 512$ tokens. We $\gamma_{\max} = 0.001$—a robust and well-adopted value—and sweep the final learning rate $\gamma_{\text{end}}$ across $\{10^{-1}, 10^{-2}, 10^{-3}, 10^{-4}, 10^{-5}, 10^{-6}\} \times \gamma_{\max}$. We present the results of this ablation in Figure 25. Recently, the question of the learning rate decaying has been an interesting topic of discussion (Bergsma et al., 2025; Schaipp et al., 2025; Hägele et al., 2024), with works focusing on the explanations of the WSD scheduler pointing to the possible impact of decaying $\gamma$ to zero (or very small magnitudes). Importantly, our ablations for models of two scales—$124$M and $210$M—suggest that the optimal choice of $\gamma_{\text{end}}$ may depend on the model scale. For example, $\gamma_{\text{end}} = 0.01 \times \gamma_{\max}$ delivers the best performance for $210$M model trained with WSD, while for

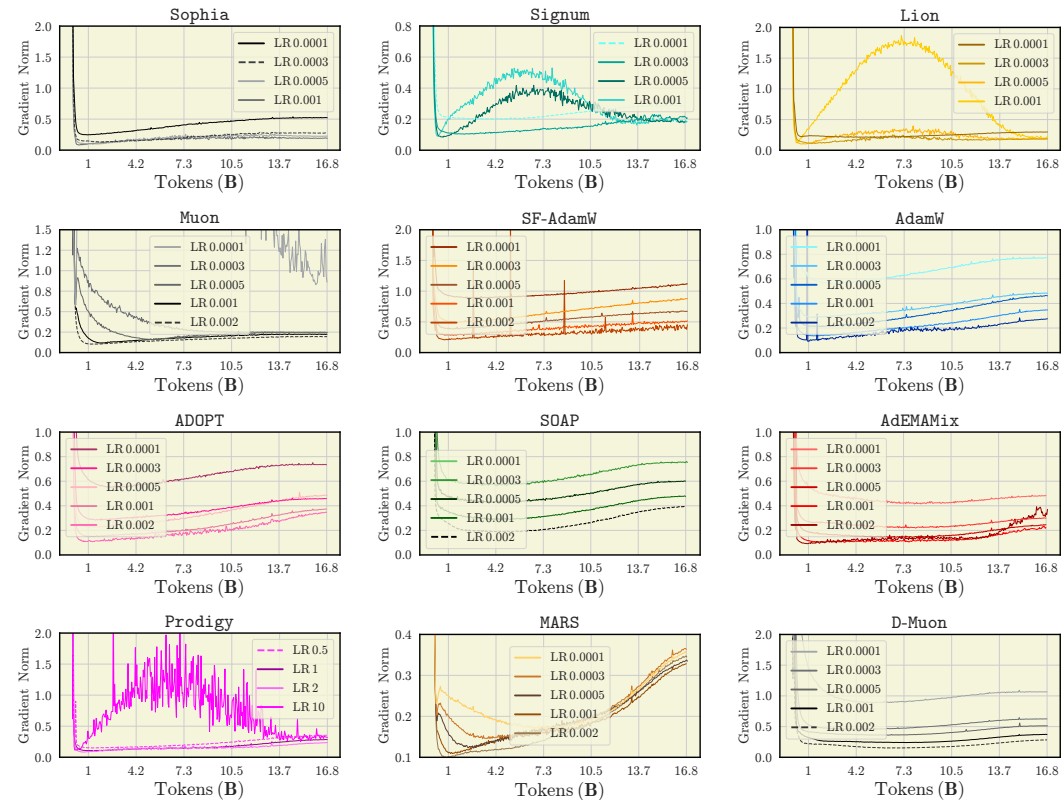

Figure 24: **Gradient norm patterns for learning rate sweep.** In this experiment, we complement the result on the learning rate sweep for optimizers (Figures 6 and 18) by tracking the gradient norms. We follow the same setup as for the $\gamma$-sensitivity ablation, varying the learning rates while training 124M language models for 16.8B tokens using a cosine $\gamma$-scheduler with $\gamma_{\text{end}} = 0.01 \times \gamma_{\text{max}}$. Except for Lion and Signum, we see that smaller $\gamma_{\text{max}}$ leads to larger magnitude of the gradient norms—unless the learning rate is high enough to nearly lead to divergence, e.g., $\gamma_{\text{max}} = 10$ for Prodigy. Interestingly, we connect the "bump" shape of the gradient norms for sign-based methods with the fact that $\gamma_{\text{max}} = 0.001$, used for them, is close to the "critical" value, an increase of which also leads to divergence—and our experiments with these optimizers on larger models support this, as we were able to decrease $\gamma$ in order to train properly.

124M model $\gamma_{\text{end}} = 10^{-6} \times \gamma_{\text{max}}$ takes the lead, which is closer to decaying to zero, as in prior works (Hägele et al., 2024; Schaipp et al., 2025). We also highlight that increasing the model size decreases the optimal learning rate for the model, thus the very small values of $\gamma_{\text{end}}$ might not affect the final performance much, while slowing the training at the latest stage, which is undesirable for modern large-scale pretraining tasks. Furthermore, we do not conduct the learning rate decaying ablations for different optimizers, utilizing only AdamW. Thus, we point out that it is possible for $\gamma_{\text{end}}$ to depend on the optimizer choice as well—this is an interesting branch of the research on optimizers to explore in future work.

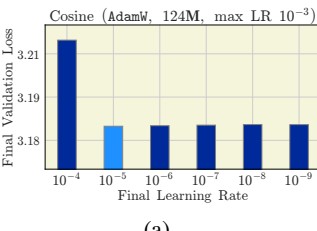 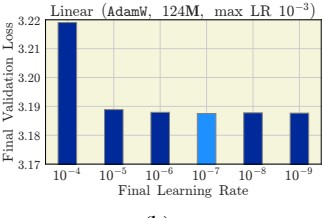 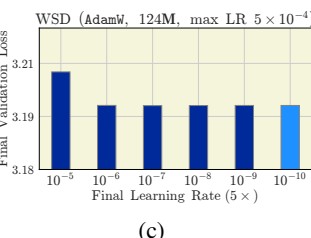

(a)                    (b)                    (c)

Figure 25: **Do not decay the learning rate down to $10\%$: ablation on 124M models.** We extend our ablation on learning rate decay from Figure 9 to studying the impact of the change of model parameters—reducing it from 210M to 124M. Consistent with our previous results, decaying the learning rate only down to $10\%$ of the maximum results in significantly worse final performance, indicating the need for further decay. Notably, for the linear **(b)** and WSD **(c)** schedulers, the best choice of $\gamma_{\text{end}}$ differs from that observed at 210M. For linear, the optimal setting shifts to $\gamma_{\text{end}} = 10^{-4} \times \gamma_{\text{max}}$ (vs. $10^{-3} \times \gamma_{\text{max}}$ at 210M), and for WSD to $\gamma_{\text{end}} = 10^{-6} \times \gamma_{\text{max}}$ (vs. $10^{-2} \times \gamma_{\text{max}}$ at 210M); see Figure 9 **(b,c)**. Overall, while the differences in final performance beyond $0.1 \times \gamma_{\text{max}}$ are relatively small, these results highlight that the optimal $\gamma_{\text{end}}$ depends on model size, which in turn influences the appropriate learning rate schedule.

**Fail of `Sophia`.** Another striking effect we observed throughout our benchmarking experiments is the convergence issues of the `Sophia` optimizer. In the main text (see Takeaway 1), we reported that "`Sophia` diverges in the small-batch setting when trained beyond the Chinchilla optimal horizon, even with sufficiently small learning rates." Later, we also noted that in the large-batch regime "`Sophia` exhibits convergence issues when extending the training run, diverging shortly after 130k steps." These phenomena are particularly puzzling, since `Sophia` does converge in long runs of 336k steps on MoE models. Figure 27 demonstrates loss curves of 124M Llama model trained with a small batch size of $32 \times 512$ tokens and using the cosine $\gamma$-scheduler. Initially, we used $\gamma_{\text{max}} = 0.001$, which proved too large for this setup, so we switched to $\gamma_{\text{max}} \in \{1e^{-4}, 3e^{-4}, 5e^{-4}\}$. For runs up to $T = 64\text{k}$ steps, training converged properly. However, increasing the number of steps beyond this point led to divergence (see Figure 27 **(a)**). Interestingly, the divergence onset occurred at almost the same iteration for both $3e^{-4}$ and $5e^{-4}$ learning rate values. For reference, training with $T = 128\text{k}$ steps in the small-batch setup results in $\sim 2.1\text{B}$ tokens, while the Chinchilla optimal horizon for this model is about 2.5B. Thus, `Sophia` fails to converge with such a small batch size even before reaching the optimal horizon. When switching to

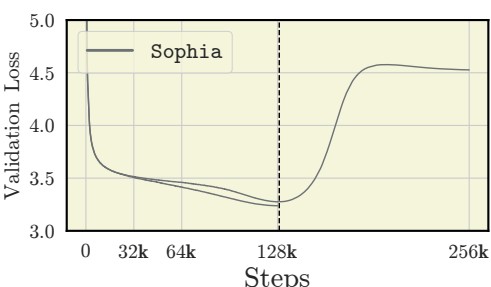

Figure 26: **`Sophia` diverges in the large-batch setup, when training for many iterations.** In the small-batch setup, we observed that `Sophia` exhibited convergence issues. With batch size $256 \times 512$, `Sophia` initially converges reliably across all training durations for 124M models used in our benchmarking. However, when extending training beyond 16.8B tokens, divergence reappears. To clearly visualize so, we present the best stable run ($T = 128\text{k}$ steps, 16.8B tokens) with the unstable one ($T = 256\text{k}$ steps, 33.6B tokens), using identical hyperparameters. The dashed line marks the iteration $t = 129720$ where divergence begins. This instability raises serious concerns about the practicality of `Sophia` for long training runs at scale.

a larger batch size of $256 \times 512$, we initially observed stable convergence across training durations from 1B to 16.8B tokens (see Figure 3 **(b)**). The same held true for an even larger batch size of $512 \times 512$ tokens, where `Sophia` converged for 64k iterations, i.e., 16.8B tokens (see Figure 4, *left*). However, doubling the training steps with the $256 \times 512$ batch size again led to divergence (see Figure 26 and Figure 4, *right*). Using the same hyperparameters that worked well for 16.8B tokens, we extended training to 33.6B tokens ($\equiv 256\text{k}$ iterations). Strangely, shortly after reaching 16.8B tokens, `Sophia` diverged, with the failure occurring precisely at $t = 129720$ (marked by the dashed line). We do not attribute these issues to implementation bugs, since `Sophia` converges in much longer runs (336k steps) with larger 520M models (see Figure 41). Instead, we caution prac-

titioners against relying on `Sophia` in its current form and emphasize that there remains substantial room for improvement. We also note that previous benchmarking work (Kaddour et al., 2023) evaluated `Sophia` only on BERT (Devlin et al., 2019) and T5 (Raffel et al., 2023) pretraining tasks (encoder-only and encoder-decoder architectures, respectively).

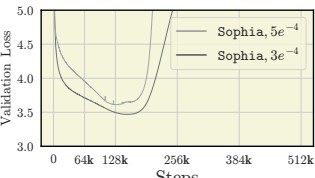 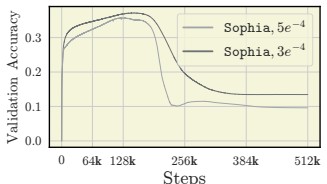 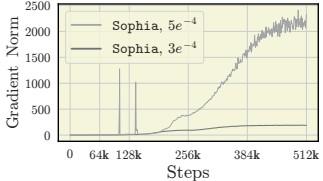

(a) Divergence after $\sim$ 2.1B tokens.    (b) Next-token prediction accuracy.    (c) Exploded gradient norms.

Figure 27: **`Sophia` diverges in the small-batch setting even with sufficiently small learning rate.** We train 124M Llama models with batch size $32 \times 512$ tokens for $T \in \{64, 128, 256, 384, 512, 1024\}$k iterations. `Sophia` diverges with the typical learning rate $\gamma_{\max} = 0.001$, and even at smaller values (e.g., $3e^{-4}, 5e^{-4}$) it still fails shortly after 2.1B tokens ($\equiv$ 128k steps). Figures **(a–c)** show loss, next-token prediction accuracy, and gradient norms, respectively. For both reported $\gamma_{\max}$ values, divergence occurs at nearly the same iteration (within 10k steps, $\sim$ 164M tokens). We do not attribute this instability to implementation bugs, since `Sophia` converges on larger MoE models for longer horizons (Figure 41). Whether this instability is related to the Chinchilla optimal horizon remains unclear; however, with a larger batch size ($256 \times 512$), `Sophia` again fails once training exceeds 16.8B tokens (see Figure 26).

> **Takeaway 15.** (I) `Sophia` diverges in the small-batch setting, even with sufficiently small learning rate. (II) When training with an increased batch size, `Sophia` starts to diverge after exceeding some limit in iterations—nearly $7\times$ Chinchilla optimal horizon in our experiments.

**Clipping & `SF-AdamW`.** Defazio et al. (2024b), when introducing the schedule-free concept, emphasized that gradient clipping should be disabled for `SF-AdamW`. Motivated by this claim, we paid particular attention to clipping during tuning. Following our setup (§ 3), we trained $124\mathbf{M}$ models with a batch size of $256 \times 512$ tokens for up to $128\mathbf{k}$ steps ($\equiv 16.8\mathbf{B}$ tokens). While sweeping the main hyperparameters of `SF-AdamW`—$(\beta_1, \beta_2)$, $\gamma$, $\lambda$, $T_{\text{warmup}}$—we also varied the gradient clipping threshold across $\{0.5, 1\}$ and tested runs without clipping, as suggested in the original paper. Our results, summarized in Figure 28, show a clear discrepancy with prior claims. Disabling clipping consistently produced unstable training, with highly spiky loss curves (Figure 28**a**). To mitigate this, we reduced the learning rate from $0.001$ to $0.0005$, which largely stabilized the runs (Figure 28**b**). However, even under this adjustment, the best clipped run—with the clipping threshold of $0.5$—still outperformed the no-clipping alternative. Thus, contrary to Defazio et al. (2024b), we find gradient clipping to be a critical hyperparameter for the stability of `SF-AdamW`.

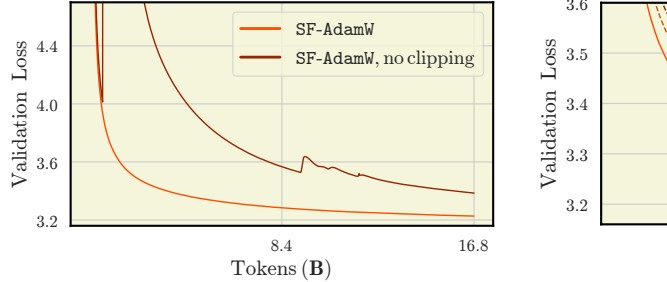

(a) Disabling clipping causes instability to `SF-AdamW`.

(b) Reducing $\gamma$ helps, but the clipped baseline is better.

Figure 28: **Clipping is significant for `Schedule-Free`**. Contrary to the claims of Defazio et al. (2024b), we find that gradient clipping remains a critical hyperparameter for `SF-AdamW`. As shown in **(a)**, disabling clipping causes severe training instabilities. To mitigate these undesired loss dynamics, we reduced the learning rate from $0.001$ to $0.0005$, which stabilized training **(b)**. However, even under this adjustment, the clipped runs still outperform those without clipping.

> **Takeaway 16.** Gradient clipping is crucial for stability of `Schedule-Free AdamW`.

**Betas sensitivity.** The impact of the beta parameters on optimizers—especially in `Adam`-like methods—has been studied both theoretically (Reddi et al., 2019; Défossez et al., 2022; Zaheer et al., 2018) and empirically (Pagliardini et al., 2024; Marek et al., 2025; Busbridge et al., 2023). However, many large-scale works in industry (DeepSeek-AI, 2024b; Brown et al., 2020; Touvron et al., 2023; Team OLMo, 2024b; Jaghouar et al., 2024) either do not tune the betas at all or simply adopt conventional defaults ($\beta_1 = 0.9$, $\beta_2 = 0.95$). Earlier in this manuscript (Takeaway 9), we argued that betas should be tuned in tandem with training duration—a conclusion supported by extensive ablations and hyperparameter sweeps. Here, we demonstrate the most striking effects of tuning beta parameters, with a particular focus on $\beta_2$. Our ablation focuses on "parameter-free" methods such as `Prodigy` (Figure 30), `SF-AdamW` (Figure 31), and the `Adam`-like optimizer `ADOPT` (Figure 29).

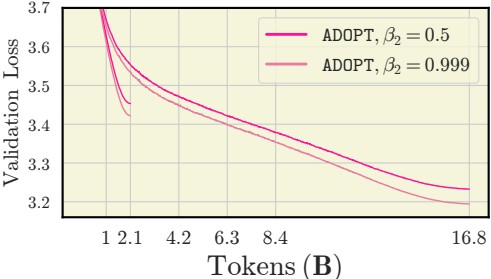

Figure 29: **ADOPT still needs $\beta_2$.** One of the main theoretical claims of Taniguchi et al. (2024)—that `ADOPT` converges with any $\beta_2$. The authors verify those on a toy problem motivated by Reddi et al. (2019). However, in LLM training, the choice of $\beta_2$ still matters significantly. Our results demonstrate that, despite theoretical guarantees, performance strongly depends on tuning $\beta_2$ in practice.

We highlight that: (I) despite the theoretical convergence guarantees of `ADOPT` for any $\beta_2$, in practice the performance gap between the best and a poorly chosen $\beta_2$ remains substantial; (II) when the batch size is small ($32 \times 512$ tokens), `Prodigy` is very sensitive to $\beta_2$, even diverging when changing it from 0.999 to 0.9999, however, applying the bias correction—see line 7 of Algorithm 13—fixes this issue; (III) prior works (Hägele et al., 2024; Song et al., 2025) question a sensitivity of `SF-AdamW` to $\beta_2$, which also studied by Defazio et al. (2024b) on image classification tasks, we confirm that changes in betas, especially $\beta_2$, highly affects the overall performance, in Figure 31 **(b)** we compare our best found ($\beta_1 = 0.9$, $\beta_2 = 0.9999$) hyperparameters with default ($\beta_1 = 0.9$, $\beta_2 = 0.95$) used by Defazio et al. (2024b), and ($\beta_1 = 0.95$, $\beta_2 = 0.99$) noticed by Hägele et al. (2024).

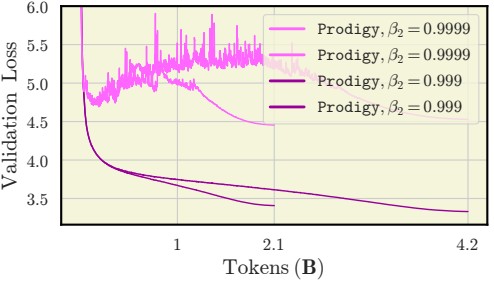

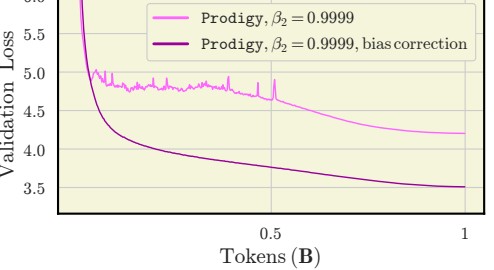

(a) Minor change in $\beta_2$ leads to divergence.    (b) Bias correction resolves issues with divergence.

Figure 30: **Prodigy is sensitive to beta parameters in the small-batch setting**. In this experiment, we follow our setup (§ 3) with a small batch size of $32 \times 512$ tokens, training 124**M** models with the best hyperparameters while sweeping $\beta_2$. Although $\beta_2 = 0.999$ yields the best results in this setting (see Table 17), even a slight change to $\beta_2 = 0.9999$ causes divergence. This occurs because ($\beta_1$, $\beta_2$) directly affect the internal statistics $r_t$, $s_t$, which determine the optimizer's effective learning rate. As shown in **(b)**, enabling bias correction (see line 7 of Algorithm 13) effectively resolves this instability.

> **Takeaway 17.** (I) `Prodigy` diverges with a minor change in $\beta_2$, when the batch size is small. Using bias correction should resolve this issue. (II) `SF-AdamW` is sensitive to ($\beta_1$, $\beta_2$); we find that typically large $\beta_2$ values (e.g., 0.9999) are beneficial for schedule-free runs. (III) Despite the established convergence theory for any $\beta_2$, `ADOPT` still requires careful tuning of this hyperparameter.

**Muon's Newton-Schulz iterations.** We briefly study the impact of Newton-Schulz iterations on the `Muon` optimizer, focusing on the first version of `Muon` with weight decay applied only to 1D param-

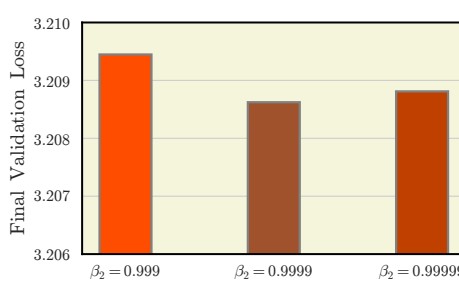
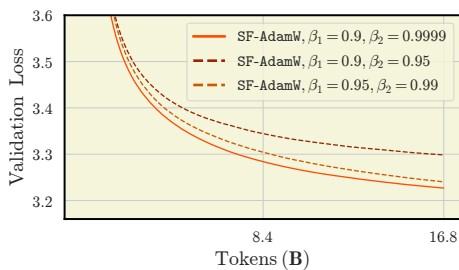

(a) $\beta_2$ sweep with fixed $\beta_1 = 0.9$.

(b) Comparison with Defazio (Defazio et al., 2024b) and Hägele (Hägele et al., 2024).

Figure 31: **Impact of beta parameters on `Schedule-Free`**. We elaborate further on the question of the sensitivity of `SF-AdamW` to $\beta_2$. For language modeling, Defazio et al. (2024b) initially suggested using default ($\beta_1 = 0.9$, $\beta_2 = 0.95$). Then, Hägele et al. (2024) revisited hyperparameter tuning of the schedule-free optimizer, proposing to apply ($\beta_1 = 0.95$, $\beta_2 = 0.99$), which improved the performance a lot. Based on our tuning, we claim that ($\beta_1 = 0.9$, $\beta_2 = 0.9999$) achieves the best performance at this scale—see **(b)**. In addition, we fix $\beta_1 = 0.9$ and report the result with sweep of $\beta_2 \in \{0.999, 0.9999, 0.99999\}$, showing that the large and unconventional value of $\beta_2 = 0.9999$ is indeed the best in schedule-free runs. We also notice that `SF-AdamW` requires a slightly larger optimal $\beta_2$, compared to all other optimizers.

eters. Recent research (Ahn & Xu, 2025; Amsel et al., 2025; Grishina et al., 2025) has extensively explored the Newton-Schulz orthogonalization procedure, examining its impact on the wall-clock speed, communication efficiency on many GPUs, and numerical precision formats. Additionally, the theoretical implications of orthogonalization procedures on optimizer convergence have been investigated in (Kovalev, 2025; Riabinin et al., 2025). In this ablation, we focus solely on the final loss performance of `Muon`, setting aside other considerations such as computational efficiency or wall-clock time. Following the tuning setup (§ 3) for smaller 124**M** parameter models with batch size of $256 \times 512$ tokens, we train for 2.1**B** tokens ($\equiv$ 16k steps), slightly below the Chinchilla optimal training horizon. Once the main hyperparameters of `Muon` are properly tuned, we sweep the number of Newton-Schulz iterations $T_{NS} \in \{1, 5, 10, 20\}$. The default setting for both `Muon` (Algorithm 8) and `D-Muon` (Liu et al., 2025) is $T_{NS} = 5$. Our results indicate that $T_{NS} \in \{5, 10\}$, and 20 yield comparable performance, with $T_{NS} = 5$ slightly outperforming the others. However, setting $T_{NS} = 1$ significantly degrades performance. These findings are summarized in Figure 32. Importantly, we always use Nesterov momentum, when running `Muon`-like methods.

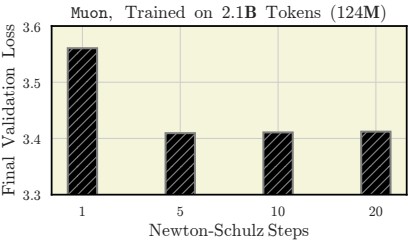

Figure 32: **`Muon`'s dependence on the number of Newton-Schulz iterations.** We perform a short ablation targeting the final loss of `Muon` (Algorithm 8) by varying the number of Newton-Schulz iterations. Training is done for 16k steps with a batch size of $256 \times 512$ tokens, sweeping $T_{NS} \in \{1, 5, 10, 20\}$. We find that increasing $T_{NS}$ beyond 5 does not improve performance, while unnecessarily increasing wall-clock time.

**`Signum` configurations.** We consider the `Signum` optimizer (Algorithm 6), which, perhaps unexpectedly, demonstrates strong performance at a small scale and competes effectively with `AdamW` when batch sizes are large (Figure 4 (*left*)). A key factor contributing to this performance is the decoupled weight decay. However, a fixed weight decay alone does not fully account for `Signum`'s efficiency. Another important ingredient is the momentum mechanism. In this ablation, we study two momentum configurations: Nesterov momentum (Nesterov, 1983) (our default, in Algorithm 6) and dampening, which is commonly used in PyTorch's implementation of `SGD`. We also compare both with the "plain" `Signum`, which uses conventional momentum without Nesterov. To give a

better understanding of these concepts, we provide a brief algorithmic description in Appendix C.2 and below:

1. Dampening update:

$$\begin{cases} \boldsymbol{m}_t \leftarrow \beta \boldsymbol{m}_{t-1} + (1-\tau)\boldsymbol{g}_t, \\ \boldsymbol{x}_{t+1} \leftarrow \boldsymbol{x}_t - \gamma_t \left( \texttt{sign}\left(\boldsymbol{m}_t\right) + \lambda \boldsymbol{x}_t \right). \end{cases}$$

2. The "plain" update of `Signum` without Nesterov momentum:

$$\begin{cases} \boldsymbol{m}_t \leftarrow \beta \boldsymbol{m}_{t-1} + \boldsymbol{g}_t, \\ \boldsymbol{x}_{t+1} \leftarrow \boldsymbol{x}_t - \gamma_t \left( \texttt{sign}\left(\boldsymbol{m}_t\right) + \lambda \boldsymbol{x}_t \right). \end{cases}$$

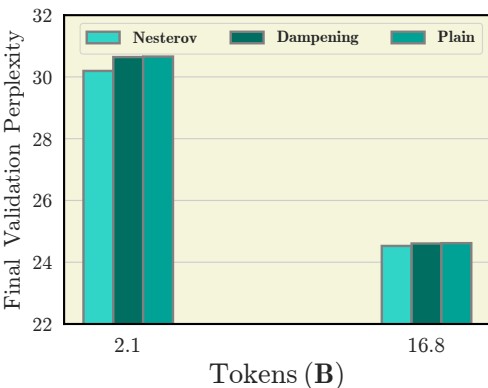

Figure 33: **Comparison of different update rules for `Signum`.** We evaluate three variants of the `Signum` update: Nesterov (our default), dampening—which resembles an EMA of $\boldsymbol{m}_t$ when the dampening parameter $\tau$ equals the momentum $\beta$—and the "plain" `Signum` without Nesterov momentum or dampening. Validation perplexity is reported for two training horizons in $(256 \times 512)$ batch size setting. The Nesterov variant corresponds to the runs included in our main benchmarking results (Figures 3 and 12). While Nesterov style momentum consistently achieves the best performance, the relative perplexity gap compared to the other variants decreases as the training horizon increases.

That is, the dampening update rule with $\tau = \beta$ resembles the basic EMA we used to see in methods such as `AdamW`—line 5 of Algorithm 1. And the "plain" `Signum` follows the conventional momentum style of `SGD` used in its PyTorch implementation [7].

The results of the comparison are shown in Figure 33. We ran three variations of the method for 2.1**B** and 16.8**B** in the "large" batch setup, and reported the final perplexity (PPL). For the Nesterov momentum version (our default), we use $\beta = 0.95$ found through careful tuning. For the damping version, we found that $\tau = 0.9$ is the best, i.e. the explicit momentum update at each iteration results in $\boldsymbol{m}_t \leftarrow 0.95 \cdot \boldsymbol{m}_{t-1} + 0.1 \cdot \boldsymbol{g}_t$; we found this configuration to be slightly better than $\tau = \beta = 0.95$. The same $\beta = 0.95$ is used in the "plain" `Signum` configuration. In all cases, the method with Nesterov momentum leads with a significant margin (for LLM pre-training) of $\sim 0.45$ PPL for 2.1**B** tokens run and $\sim 0.11$ PPL for long 16.8**B** tokens training over dampening and plain `Signum` variations. Interestingly, these margins vanish with the increased training horizon. We highlight the importance of Nesterov momentum for `Signum` runs in Takeaway 18. We also notice that Nesterov momentum slowdowns training, but not significantly, as our wall-clock time ablation reveals that `Signum`, with Nesterov momentum, is still the fastest method in various scenarios.

> **Takeaway 18.** `Signum` with Nesterov momentum (our PyTorch implementation) consistently outperforms both the dampening variant (EMA-like) and the basic version without Nesterov.

**MARS types.** In addition to the `MARS` optimizer that leverages Algorithm 14 to optimize 2D parameters, and `AdamW` to optimize 1D parameters and `lm_head`, we also study `MARS-Lion` and `MARS-Shampoo` methods—Algorithms 15 and 16 respectively. Before delving into the experimental details, we note that it is possible to use `MARS`-like methods for all parameters of LLM, however, this would be inefficient and in the original codebase [8], the default choice is to optimize all 1D parameters with `AdamW`. Therefore, we do the same in our experiments. For this ablation, we utilize 124**M** model and train for $T \in \{8, 16, 32, 48, 64, 128\}$**k** with batch size of $256 \times 512$ (we report plots only for this batch setting), varying $\gamma$-schedulers and $T_{\text{warmup}}$. We observe similar patterns regarding the impact of weight decay on these methods—for the majority of the training

---

[7]torch.optim.SGD

[8]https://github.com/AGI-Arena/MARS

the loss curves with $\lambda = 0$ look "convex" and lie below the curves corresponding to $\lambda = 0.1$, but then runs with the non-zero weight decay take the lead. Regarding tuning `MARS-Lion` and `MARS-Shampoo`, we found interesting observations related to our previous experience in hyperparameter tuning. `MARS-Lion`, despite optimizing only 1D parameters with `Lion`, is also sensitive to warmup, as the latter method, and benefits from longer warmup durations—see Figure 34 **(c)**. Similarly to `Lion`, `MARS-Lion` also prefers the WSD scheduler (Figure 34 **(b)**) that outperforms the corresponding runs with the cosine baseline. Notably, the best $(\beta_1, \beta_2)$ parameters of `MARS-Lion` coincide with those found for `Lion` in Table 10 and in (Chen et al., 2023). Of all the `MARS` versions, `MARS-Shampoo` performs the worst. We also note that this variant of `MARS` is not included in the original paper's (Yuan et al., 2024) experiments on LLMs. In our setup with batch size of $256 \times 512$ both `MARS` (`MARS-AdamW`) and `MARS-Lion` do not outperform the `AdamW` baseline. However, this may be due to the smaller batch size: in the original work, the authors use $480 \times 1024$ ($\equiv 492\mathbf{M}$ tokens) batch size, and our experiments with the larger batch size of $1984 \times 512$ ($\equiv 1\mathbf{B}$ tokens)— see Figure 1—reveal that both `MARS-AdamW` and `Lion` greatly benefit from the increased batch size. Therefore, we highlight that it may be the case that `MARS-Lion` can outperform `AdamW` in some cases.

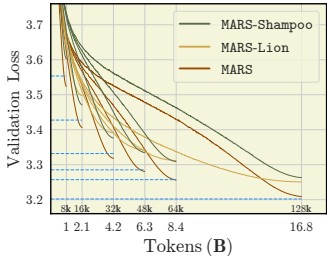

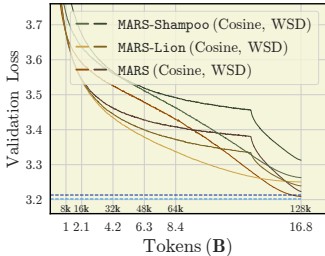

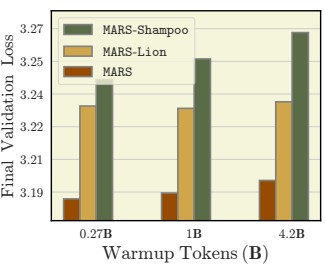

(a) `MARS` family of methods.     (b) Sensitivity to $\gamma$-schedulers.     (c) Warmup ablation.

Figure 34: **`MARS` family of optimizers.** We study three `MARS`-based algorithms: `MARS-AdamW` (just `MARS` in our work), `MARS-Lion`, and `MARS-Shampoo`. In this ablation, our goal is to complement our `MARS` runs with experiments for other similar methods, and support findings for these optimizer with our previous experience in tuning `Lion`. We replicate the setup from § 3 and train with the batch size of $256 \times 512$ for the same training durations as in Section 4.1. In **(a)**, we show that, indeed, `MARS-AdamW` outperforms other alike methods, as reported in (Yuan et al., 2024) regarding the `MARS-Lion` optimizer. Interestingly, in **(b)**, we show that the choice of $\gamma$-scheduler for `MARS`-based methods also depends on optimizer, as such, WSD runs of `MARS-Lion` outperform itself with cosine. Dashed blue and dark blue lines correspond to the `AdamW` baseline with cosine and WSD schedulers, respectively. Furthermore, in the same way as `Lion` benefits from larger warmup (Figure 7), `MARS-Lion` also improves with 8k steps ($\equiv 1\mathbf{B}$ tokens) warmup, however, this improvement is not as dramatic **(c)**.

> **Takeaway 19.** Among current `MARS` variants, `MARS-AdamW` is the best. Notably, other modifications—`MARS-Shampoo` and `MARS-Lion` are differently affected by $\gamma$-schedulers and warmup. `MARS-Lion` prefers the WSD scheduler over cosine, and shows the greatest stability to warmup sweep among all `MARS`-based methods.

**On learning rates of `Prodigy`.** Throughout our benchmarking results (Figures 1, 12 and 37), `Prodigy` consistently ranks among the top 6 optimizers, performing close to `AdamW` at smaller scales and maintaining strong performance even when applied to MoE architectures. Interestingly, when training 124**M** models with an increased batch size of $512 \times 512$ tokens, `Prodigy` outperforms the `AdamW` baseline, suggesting that its critical batch size (Erdil, 2024; Zhang et al., 2024a; Hoffmann et al., 2022) may be larger than that of `AdamW`. While highly efficient, `Prodigy` is generally easy to tune, except for its sensitivity to $\beta_2$ (Figure 30) in the small-batch setup. This robustness is attributed to its adaptive learning rate mechanism, which relies on two exponential moving average sequences

$$\begin{cases} r_t \leftarrow \sqrt{\beta_2} r_{t-1} + (1 - \sqrt{\beta_2}) \gamma_t d_t^2 \langle \boldsymbol{g}_t, \boldsymbol{x}_0 - \boldsymbol{x}_t \rangle, \\ \boldsymbol{s}_t \leftarrow \sqrt{\beta_2} \boldsymbol{s}_{t-1} + (1 - \sqrt{\beta_2}) \gamma_t d_t^2 \boldsymbol{g}_t, \end{cases}$$

that control the learning rate magnitude with a multiplier of

$$d_{t+1} \leftarrow \max\left\{ d_t, \frac{r_t}{\|\boldsymbol{s}_t\|_1} \right\}.$$

At first, we define the effective learning rate of `Prodigy` as:

$$\gamma_{\mathtt{t+1}}^{\mathrm{eff}} := \gamma_t d_t, \tag{2}$$

thus, when bias correction is applied—which we found necessary to ensure stability for small batches—Equation (2) becomes:

$$\gamma_{\mathtt{t+1}}^{\mathrm{eff}} = \gamma d_t \frac{\sqrt{1 - \beta_2^t}}{1 - \beta_1^t},$$

where $\gamma$—the learning rate—is usually set to 1 for `Prodigy`, which we confirmed to work the best through the sweeps in Figures 6 and 18.

When introducing the concept of the effective learning rate $\gamma^{\mathrm{eff}}$, we note that it depends on the momentum parameters ($\beta_1$, $\beta_2$), the base learning rate $\gamma$, and the EMA sequences $\boldsymbol{s}_t$, $r_t$. Moreover, applying a $\gamma$-scheduler further influences how $\gamma$ evolves over iterations. To study these interactions, we examine the dynamics of the effective learning rate (Equation (2)) under different schedulers (Equation (2)). For this purpose, we train a small $124\mathbf{M}$ model with the batch size of $256 \times 512$ tokens. The training horizon is short—8k steps—with a warmup of 1000 steps, and $\beta_2 = 0.999$—the best for `Prodigy` in this setup according to our tuning. We also set the learning rate of `Prodigy` to 1 as in our best benchmarking runs. As in previous experiments, we apply WSD and cosine $\gamma$-schedulers, and additionally show a run without any scheduler. For the WSD scheduler in this ablation, we do not rescale $\gamma$ to half the optimal value for cosine, as we are interested in the dynamics of $\gamma_t^{\mathrm{eff}}$ rather than the final performance; observing it without rescaling provides a clearer picture. Figure 35 **(a)** shows the dynamics of the effective learning rate $\gamma_t^{\mathrm{eff}}$, while **(b)** illustrates the effect of applying scheduling to $\gamma = 1$. The starting points of the curves differ slightly due to variations in the final learning rate—cosine decays $\gamma_t$ down to 0.01, whereas WSD decays it to zero using the $(1 - \sqrt{x})$ decay pattern—however, those differences do not affect the qualitative shape of the figures obtained.

Interestingly, across all schedulers, we observe a common pattern—the effective learning rate warmup is longer than $T_{\mathrm{warmup}} = 1000$ steps—meaning that `Prodigy` experiences an "implicit warmup" beyond the explicitly set value. Another notable observation is that when using the cosine scheduler with $\gamma = 1$, the maximal effective learning rate reaches $\gamma_{\mathrm{max}}^{\mathrm{eff}} \sim 1.08\times$ larger than the learning rate of `AdamW` we use in a similar setting (0.001). Consequently, setting `Prodigy`'s learning rate to the default value of 1 produces dynamics closely matching those of `AdamW`. This insight could be useful for practitioners as a proxy for tuning `Adam`-like optimizers: one can launch `Prodigy` with $\gamma = 1$, track the effective learning rate (Equation (2)), and then set the `AdamW` peak learning rate to $\gamma_{\mathrm{max}}^{\mathrm{eff}}$. We highlight this one more time in Takeaway 21.

> **Takeaway 20.** We explain the effectiveness of `Prodigy` in "learning rate-free" training through the concept of the effective learning rate (Equation (2)). Determined by two EMA sequences, the effective learning rate mimics the behavior and magnitude of the learning rate in `AdamW`-like methods. Importantly: (I) the magnitudes of the effective learning rate are close to those of `AdamW`; (II) effective learning rate ensures an implicit warmup that is longer than initially set.

> **Takeaway 21.** We point out that it might be interesting for researchers to try `Prodigy` as a proxy for learning rate tuning of `Adam`-like methods, e.g., (I) tune betas of `Prodigy`, (II) set $\gamma = 1$, (III) track $\gamma_t^{\mathrm{eff}}$, and (IV) look at the $\gamma_{\mathrm{max}}^{\mathrm{eff}}$ and set the learning rate of the `Adam`-like method to this value.

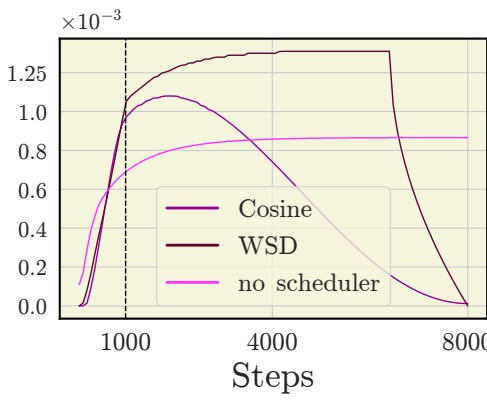

(a) Effective learning rate of `Prodigy`.

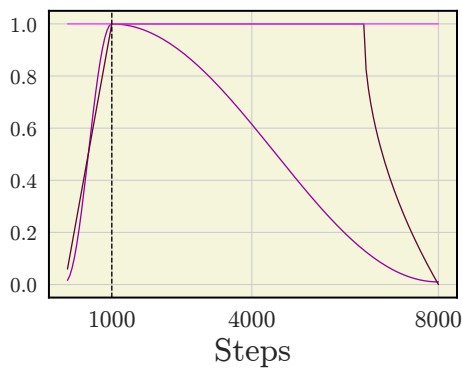

(b) Learning rate of `Prodigy`.

Figure 35: **EMA sequences of `Prodigy` result in the effective learning rate that emulates the dynamics of learning rate that we used to observe for `AdamW`.** Fixing the peak learning rate at $\gamma = 1$ (following Mishchenko & Defazio (2024)), the EMA sequences $r_t$ and $s_t$ (lines 8, 9 of Algorithm 13) result in the effective learning rate shown in **(a)**. The dashed line indicates the warmup duration. Across all schedulers and the run without a $\gamma$-scheduler, the warmup of $\gamma_t^{\text{eff}}$ **(a)** is consistently longer than that of $\gamma_t$ **(b)**, providing an implicit warmup. With cosine and WSD schedulers, the peak $\gamma_t^{\text{eff}}$ exceeds that of the run without a scheduler. Notably, the peak effective learning rates, especially for the cosine scheduler, are very close to the default value 0.001 used for `AdamW` at this model scale. This demonstrates that `Prodigy` may guide practitioners in tuning learning rates for `Adam`-like optimizers.

## F.2 ABLATIONS FOR 210M MODEL

**Results with a batch size of $256 \times 512$.** In this section, we verify if our selected hyperparameters from smaller 124M allow accurate transfer to a slightly larger model. We point out that the most important hyperparameters to be sweeped are learning rate and gradient clipping. Regarding the learning rate, we observe that it only becomes a sensitive choice for sign-based methods, while the optimal hyperparameters for `AdamW` remain the same. After retuning the learning rate for sign-based optimizers (see Appendix G.2), we replicate the setup from § 3: we stay in the "large" batch regime and train for the same number of steps (tokens) as in Figure 3 **(b)**. We report our benchmarking for 210M models in Figure 36 and the training dynamics of optimizers in Figure 37.

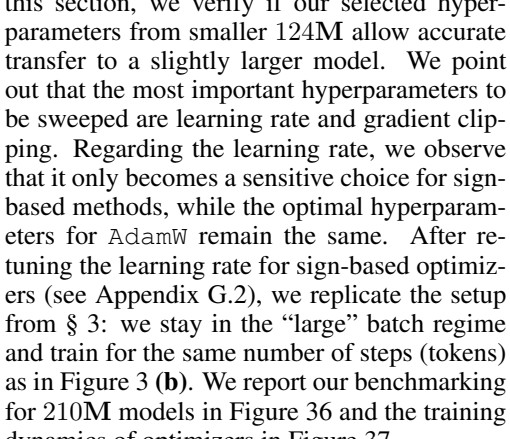

Figure 36: **Ranking of optimizers for 210M models with the batch size of $256 \times 512$ tokens.** Increasing a model size from 124M to 210M results in almost identical ranking of optimizers compared to Figure 3 **(b)**. At this scale, we observe a smooth transition in our benchmarking.

In this section, we complement our ablations from the main part with experiments specifically targeting 210M models. Compared to 124M ablations (§ F.1), we perform fewer studies here. We focus on two aspects: the sensitivity of `ADOPT` to its $\varepsilon$ hyperparameter, and the impact of weight initialization in LLMs and its interaction with the warmup.

**Complementing benchmarking results of 210M models.** In addition to results from the main part, we show the dynamics of the validation loss in Figure 37. The presented runs correspond to those in Figure 36.

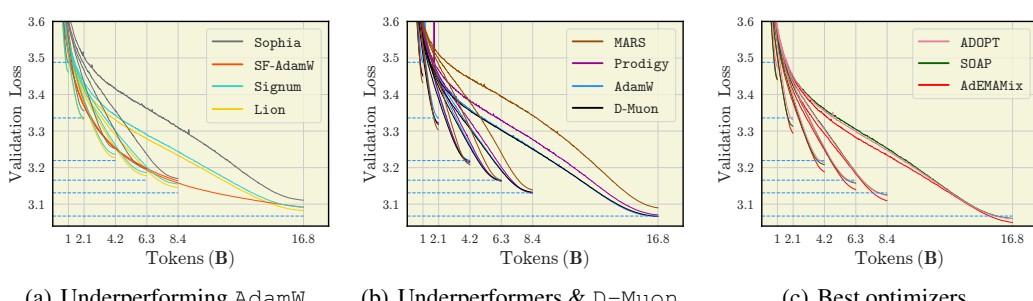

(a) Underperforming `AdamW`.     (b) Underperformers & `D-Muon`.     (c) Best optimizers.

Figure 37: **Comparing optimizers for training a 210M parameter LLM.** We plot the training dynamics of: **(a,b)** optimizers that underperform `AdamW` for pretraining a 210M model; **(c)** optimizers that outperform `AdamW` in this setup. A complete ranking of methods in this setting is in Figure 36.

**`ADOPT` is sensitive to the epsilon hyperapameter, but the suggested** $\varepsilon = 10^{-6}$ **is the best.** Among the many important hyperparameters, some receive less attention despite their influence. For `Adam`-like methods, one such parameter is $\varepsilon$ in the denominator of the update rule. While the default and widely accepted value for `AdamW` is $10^{-8}$, there is ongoing discussion in the community regarding other values that can significantly degrade training (Team OLMo, 2024a; He et al., 2024). The `ADOPT` optimizer also includes this hyperparameter—see line 6 of Algorithm 2. Interestingly, the authors recommend using a larger value of $\varepsilon = 10^{-6}$, which is higher than the conventional choice for `AdamW`. We perform a sweep over $\varepsilon$, keeping all other hyperparameters at their best values, and report the results in Figure 38. As suggested by Taniguchi et al. (2024), $\varepsilon = 10^{-6}$ outperforms all other tested values, with a noticeable margin for $\varepsilon \leq 10^{-5}$.

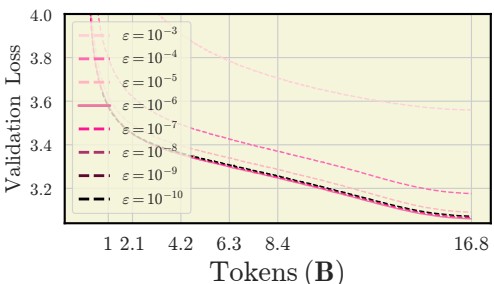

Figure 38: **`ADOPT`'s sensitivity to** $\varepsilon$**.** Interestingly, the suggested by the authors $\varepsilon = 10^{-6}$ is the best hyperparameter for this method. There is not a noticeable difference in convergence for $\varepsilon = \{10^{-6}, 10^{-7}, 10^{-8}, 10^{-9}, 10^{-10}\}$, but the values of $10^{-5}$ and above give a much morse results.

**Changing weight initialization and the effect on warmup.** A common approach to weight initialization in LLMs is the truncated Gaussian distribution with a predefined standard deviation (std). In popular codebases for scalable training (Shoeybi et al., 2020; Rasley et al., 2020; Team OLMo, 2024a), the default std is 0.02. Notably, in DeepSeek-V3 (DeepSeek-AI, 2024b), the default std is reduced to 0.006. Previously established connections between weight initialization and warmup report twofold results: ones (Huang et al., 2020; Zhu et al., 2021) state that with a smaller std, one can reduce or even eliminate the need for warmup, while others (Kalra & Barkeshli, 2024; Kosson et al., 2024a) highlight the importance of warmup for small weight initializations. In our experiments, we investigate how both initialization styles interact with the warmup duration and the batch size scaling. Specifically, we compare the DeepSeek style initialization (std = 0.006) with the conventional initialization (std = 0.02). We use two batch size settings: $512 \times 512$ tokens and $256 \times 512$ tokens, training Llama-based models for two horizons $T \in \{32, 128\}$k steps and sweeping $T_{\text{warmup}} \in \{50, 500, 1000, 2000\}$ iterations. For this ablation, we use only `AdamW` with all other hyperparameters set to the best values identified from tuning of 210M models. We report the results in Figure 39. Overall, we observe that smaller weight initialization favors longer warmup durations and performs significantly worse with short warmup. Increasing the batch size reduces this gap for shorter warmups, suggesting an interplay between initialization scale, warmup duration, and batch size.

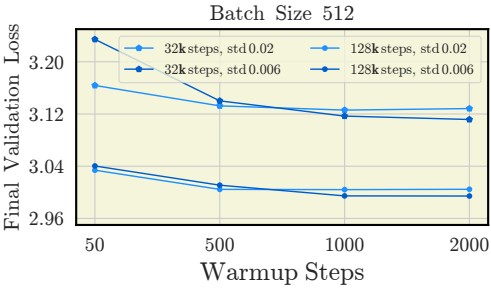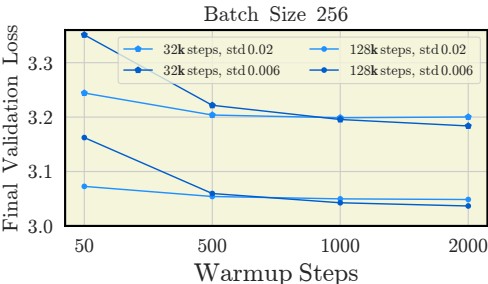

Figure 39: **Weight initialization with smaller std prefers longer warmup.** We compare final loss of models trained with `AdamW` using two weight initializations: the conventional std = 0.02 and a smaller std = 0.006 as in DeepSeek. We vary the training horizon, warmup duration, and batch size (without changing the number of iterations). Our results indicate that smaller initialization benefits from longer warmup, leading to better performance compared to std = 0.02. However, with very short warmup, the conventional initialization outperforms the smaller one. Interestingly, increasing the batch size reduces the performance gap between the two initializations for longer training runs.

> **Takeaway 22.** Weight initialization with smaller standard deviation, as in DeepSeek, benefits from longer warmup but underperforms with very short warmup. Increasing the batch size reduces the performance gap between small and conventional initializations.

### F.3 ABLATIONS FOR **720M** MODEL

**Complementing benchmarking results of 720M models.** In addition to results from the main part, we show the dynamics of the validation loss in Figure 40. The presented runs correspond to those in Figure 1.

We observe that sign-based methods and `Sophia` require careful re-tuning of the learning rate to converge on larger models. Notably, despite increasing the training horizon in terms of tokens, with larger batch size, the number of steps is reduced compared to our runs in § 4.1 and § 4.2; in this part of the benchmarking, we consider runs of $\{8, 16, 48\}$k iterations (the Chinchilla optimal duration at $\sim 14.4$k). This reduction in steps necessitates re-tuning optimizer-related hyperparameters such as $\beta_2$. We describe hyperparameter changes in Appendix G.4.

Studying the training dynamics (Figure 40), we find that `SF-AdamW`, and sign-based `Lion` and `Signum` scale poorly. `Sophia` can outperform our `AdamW` for short runs of 8k iterations, but then degrades significantly. Interestingly, `MARS` greatly benefits from this setup, emerging the second best-performing optimizer, closely following `AdEMAMix`: as it benefits from large batch size (see Figure 4 (*left*)), and does not degrade with increased model size unlike `Signum`, and `Lion`. On another hand, `Prodigy` was proven to be more beneficial at larger batch size, however, this setup it occured to be less performant. `D-Muon` is consistent across all settings we have tried, while `Muon` degrades when scaling model size (Figure 16 **(c)**).

As in (Vyas et al., 2024), we find that `SOAP` outperforms `AdamW` at the Chinchilla optimal duration and below. However, in longer training, `AdamW` narrows the gap and eventually surpasses `SOAP`. Another claim regarding the `SOAP` optimizer—that it is more beneficial, when the batch size is sufficiently large—remains quite questionable: (I) as Figure 10 (runs with 2M batch size) suggests that the matter of `SOAP` being better than `AdamW` is conditioned by the setup choice, which when properly tuned turns that `AdamW` becomes better even at Chinchilla optimal duration; (II) when considering 1M batch size setup in Figures 1 and 40, the performance gain of `SOAP` over `AdamW` is less pronounced than in our settings with smaller batches for 124M and 210M models (see Figures 12 **(b)** and 37 **(c)**).

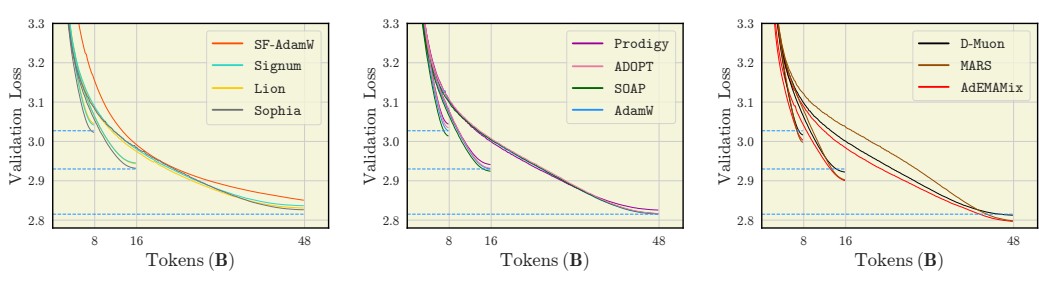

(a) Underperforming `AdamW` (I).  (b) Underperforming `AdamW` (II).  (c) Outperforming `AdamW`.

Figure 40: **Comparing optimizers for training a 720M parameter LLM.** We conduct runs with the batch size of 1M tokens. While previous ablations (see Figure 4) reveal that sign-based methods can outperform `AdamW` at sufficiently large batches, this advantage does not persist when scaling model size. On another hand, `MARS`, that also benefits from the increased batch size, along with `AdEMAMix` dominates over other optimizers with a huge gap.

### F.4 ABLATIONS FOR **520M** MoE MODEL

**Complementing ablations regarding transfer to MoE models.** In addition to results from the main part, we show the dynamics of the validation loss in Figure 40. The presented runs correspond to those in Figure 11.

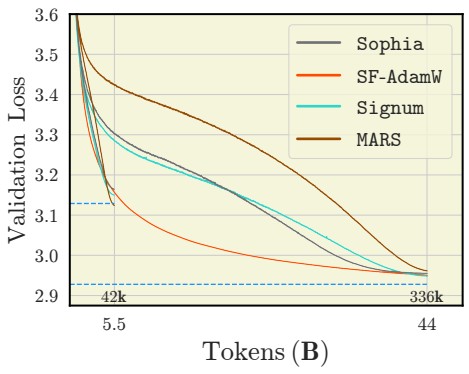 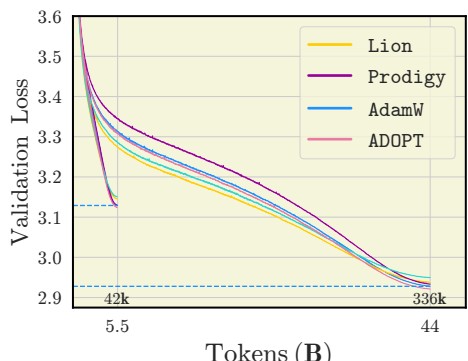

Figure 41: **Comparing optimizers for training a 520M parameter MoE.** Training dynamics of leading optimizers is in Figure 2. Results closely remind those in Figure 12 **(a,b)**. The `AdamW` baseline by far outperforms `Sophia`, `SF-AdamW`, `MARS`, and sign-based methods for 44**B** training horizon. Remarkably, in the same way as `Prodigy` followed `AdamW` in Figure 12 **(b)**, we observe a similar situation for the MoE model.

### F.5  WALL-CLOCK PERFORMANCE OF OPTIMIZERS ACROSS MODELS OF DIFFERENT SCALE

After conducting experiments for models of different sizes, we are ready to present the wall-time comparison for each method. For this purposes, we use a single GPU, and run each optimizer for 100 iterations on a small batch size of 16 without gradient accumulation and `torch.compile`. In this ablation, we consider a wider range of model sizes (30M–1B). We run each method 5 times with different seeds, compute the standard deviation, and report the mean wall-clock time per 100 iterations for each model configuration. We observe that all methods take the roughly the same or very close time to complete 100 iterations, with the exception of SOAP. We point out that wall-clock time for all optimizers exhibits a linear dependence on the model size ("model size" axis is rescaled in plots). However, SOAP slows down faster and we may expect a slowdown further, due to its preconditioner matrices operations which are fast only for certain matrices that are smaller than a predefined size. See details of this ablation in Appendix F.5, and Figures 43 and 44.

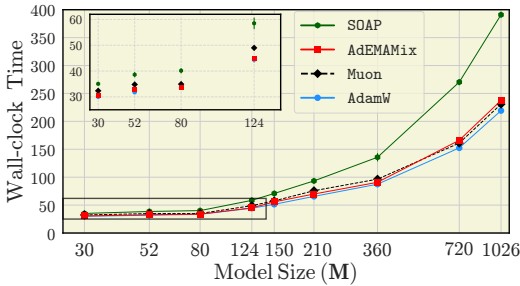

Figure 42: **Wall-clock time comparison.** SOAP slows down the most as model size increases.

> **Takeaway 23.** Most optimizers exhibit similar wall-time performance, with sign-based methods being slightly faster (Figure 43). SOAP is the main exception, showing a notable slowdown as model size increases.

We complement the wall-clock performance analysis from (Figure 42) by presenting complete results for all optimizers. The experimental setup is simple and consistent: we use a batch size of 16 (16 × 512 tokens), run for 100 iterations on a single GPU, without gradient accumulation, and we do not apply `torch.compile`. Precise model configurations for all scales (30M–1B) are reported in Table 2.

Table 2: **Configurations for our Llama-like models for the wall-clock experiments.**

| # Parameters | 30M | 52M | 80M | 124M | 150M | 210M | 360M | 720M | 1026M |
|---|---|---|---|---|---|---|---|---|---|
| Hidden size | 384 | 512 | 768 | 768 | 768 | 768 | 1024 | 2048 | 1792 |
| # Attention heads | 6 | 8 | 6 | 12 | 12 | 12 | 16 | 16 | 14 |
| # Layers | 8 | 8 | 6 | 12 | 16 | 24 | 24 | 12 | 24 |
| Init. std | 0.02 | 0.02 | 0.02 | 0.02 | 0.02 | 0.02 | 0.02 | 0.02 | 0.02 |
| Use bias | no | no | no | no | no | no | no | no | no |
| RMSNorm epsilon | 0.00001 | 0.00001 | 0.00001 | 0.00001 | 0.00001 | 0.00001 | 0.00001 | 0.00001 | 0.00001 |
| Positional encoding | RoPE | RoPE | RoPE | RoPE | RoPE | RoPE | RoPE | RoPE | RoPE |

Figure 43 shows a bar plot summarizing wall-clock time comparisons for all optimizers. Additionally, Figure 44 visualizes the per-optimizer behavior when scaling model size, omitting SOAP, AdEMAMix, Muon, and AdamW, as their results are already presented in the main part—see Figure 42.

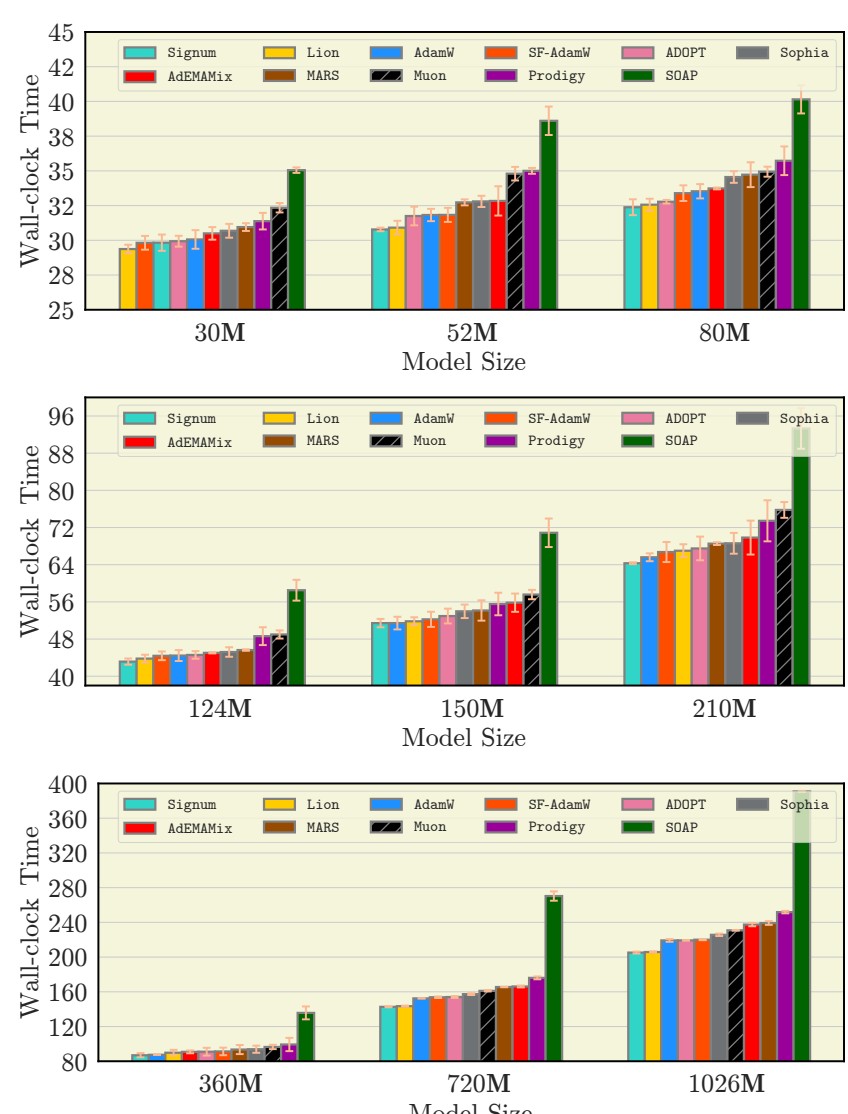

Figure 43: **Wall-clock time performance: gathered.** We report the wall-clock time (in seconds) for training each model for 100 iterations using a small batch size of $16 \times 512$ tokens on a single GPU, without gradient accumulation or `torch.compile`. Bars show the ranking of optimizers from fastest (`Signum`) to slowest (`SOAP`) gathered across all model scales. While the differences between most optimizers are small, `SOAP` is consistently slower. The absolute times may vary depending on the hardware, but the relative patterns remain consistent.

## G  HYPERPARAMETER TUNING

**How do we tune hyperparameters?** We perform systematic hyperparameter tuning for all algorithms, starting with smaller models (124**M**, 210**M**) and extrapolating to larger, 583**M** and 720**M** models. Our tuning process for 124**M** model focused on two primary settings: **"small" batch setting** (32 batch size) and **"large" batch setting** (256 batch size). For both settings, we use a sequence length of 512 tokens, resulting in 16**k** and 130**k** tokens per batch, respectively. If the batch cannot fit into memory, we use gradient accumulation steps, while maintaining the effective batch size.

We also include ablations on even larger batch size for 124**M** models, where we train on 512 batch size (260**k** tokens correspondingly). We train 583**M** models on the batch size of 3936, preserving the basic sequence length of 512, that is, $\sim 2$**M** tokens. And the larger models for benchmarking purposes—of 720**M**—were trained on the batch size of 1984, resulting in $\sim 1$**M** tokens.

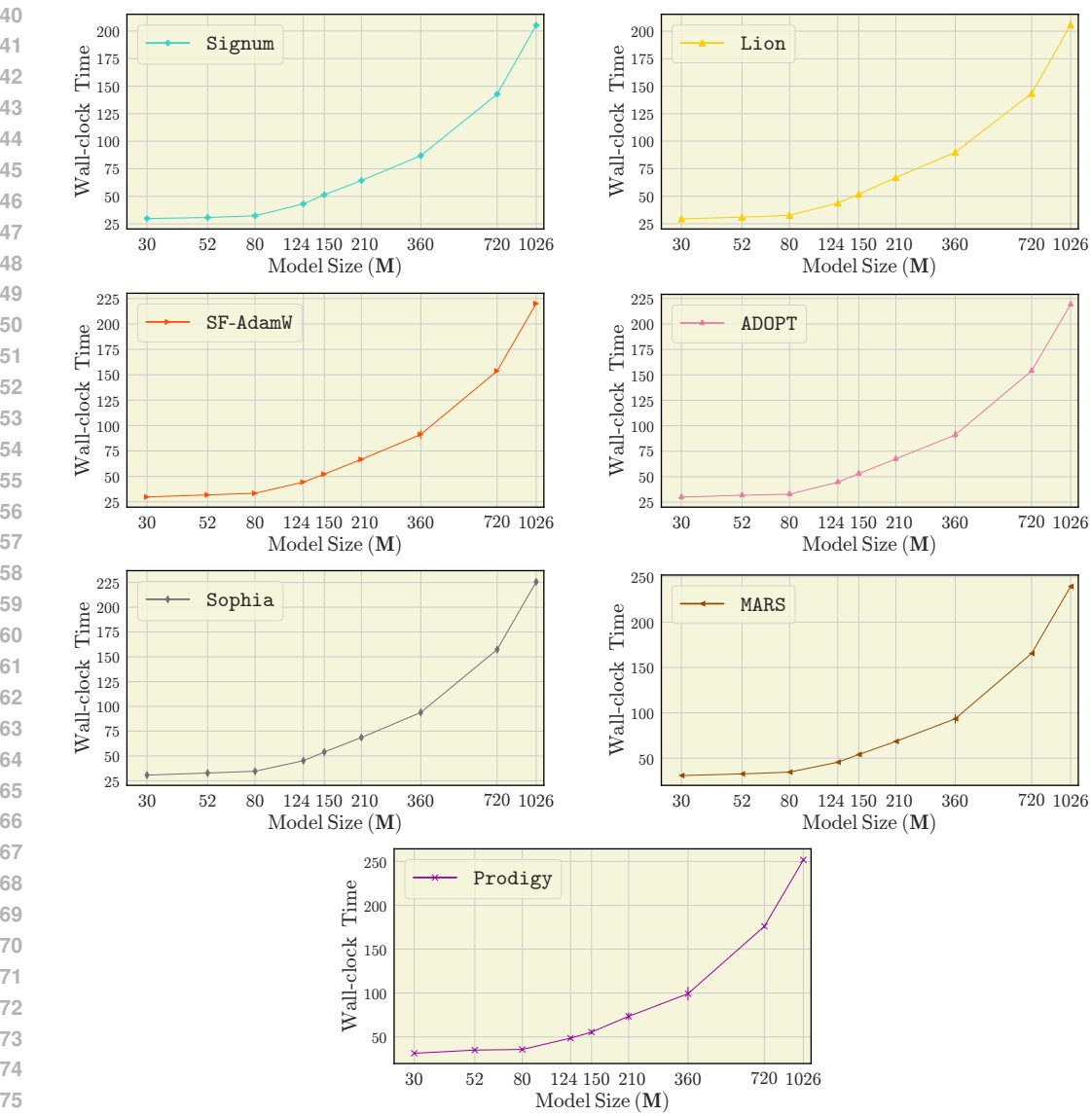

Figure 44: **Wall-clock time performance: individual.** Complementing Figures 42 and 43, this figure shows the evolution of wall-clock time per 100 iterations for each optimizer as model size increases. Optimizers already shown in the main part are omitted. To improve visualization, the abscissa is re-scaled to highlight the increase in wall-clock time with model size.

We first run multiple experiments, greed searching hyperparameters, on near Chinchilla optimal training length using *cosine learning rate scheduler* (except for `SF-AdamW`):

• for $124\text{M}$ models we tune at $2.1\text{B}$ tokens for both "small" (32) and "large" (256) batch size setting (see Appendix G.1),

• for $210\text{M}$ models we replicate training runs with the best hyperparameters found at $124\text{M}$ scale, except for the learning rate (see Appendix G.2),

• at $583\text{M}$ scale, we only ablate the effect of the $z$-loss regularizer while training with `AdamW` and `SOAP` on a near-Chinchilla optimal number of tokens (see Appendix G.3),

• for $720\text{M}$ models we tune at $16\text{B}$ tokens (see Appendix G.4),

• our MoE setting we discuss in-depth in Appendix G.5.

We present the configurations for different training horizons in Tables 3, 4, 6, 5.

Table 3: **Lengths of training for "Small" batch settings (32 × 512).**

| # Parameters | Tokens (Iterations) | | | | | | Chinchilla Tokens |
|---|---|---|---|---|---|---|---|
| 124M | 1B (64k) | 2.1B (128k) | 4.2B (256k) | 6.3B (384k) | 8.4B (512k) | 16.8B (1024k) | 2.5B |
| 210M | 1B (64k) | 2.1B (128k) | 4.2B (256k) | 6.3B (384k) | 8.4B (512k) | 16.8B (1024k) | 4.2B |

Table 4: **Lengths of training for "Large" batch settings (256 × 512).**

| # Parameters | Tokens (Iterations) | | | | | | Chinchilla Tokens |
|---|---|---|---|---|---|---|---|
| 124M | 1B (8k) | 2.1B (16k) | 4.2B (32k) | 6.3B (48k) | 8.4B (64k) | 16.8B (128k) | 2.5B |
| 210M | 1B (8k) | 2.1B (16k) | 4.2B (32k) | 6.3B (48k) | 8.4B (64k) | 16.8B (128k) | 4.2B |

Table 5: **Lengths of training for 2M (3936 × 512) batch size setting.**

| # Parameters | Tokens (Iterations) | Chinchilla Tokens |
|---|---|---|
| 583M | 13B (6.5k) | 11.7B |

Table 6: **Lengths of training for 1M (1984 × 512) batch size setting.**

| # Parameters | Tokens (Iterations) | | | Chinchilla Tokens |
|---|---|---|---|---|
| 720M | 8B (8k) | 16B (16k) | 48B (48k) | 14.4B |

Important to note, for larger models, we mostly kept the best hyperparameters found for the 124M model and re-tuned the learning rate, beta parameters, and gradient clipping. For dense LLMs, summarize this process in Appendices G.1, G.2, G.3, G.4, and cover the MoE setup in Appendix G.5.

Additionally, when we report the effect of a particular hyperparameter, we assume that the remaining hyperparameters of the algorithm have already been tuned. Thus, the results isolate and highlight only the impact of the chosen hyperparameter on overall performance.

**Hyperparameters used in our experiments with learning rate schedulers.** Once we found the best setting for each method using cosine learning rate scheduler, we are ready to obtain the optimal performance of our method with WSD (Hu et al., 2024) and linear schedulers. For the latter one, we use the same hyperparameters as for the cosine scheduler. However, for WSD, we follow the rule of thumb from (Hägele et al., 2024):

• use half the optimal learning rate for the cosine scheduler,

• use 20% of iterations for cooldown phase,

• use $(1 - \sqrt{x})$ decay shape for the cooldown phase,

the only difference is that we do not employ stochastic weight averaging (Izmailov et al., 2019).

Therefore, we maintain most hyperparameters across optimizers, only re-tuning the learning rate. For `Muon` and `MARS`, we reduce both `AdamW`'s learning rate and the learning rate for non-1D parameters. This approach ensures a fair comparison while accounting for the unique properties of each optimizer.

Importantly, the rule of thumb (Hägele et al., 2024) for using the decay shape $(1 - \sqrt{x})$ works better in our setting. We use exactly this shape during the cooldown phase of the WSD scheduler for all optimizers.

We report a series of comparisons between different schedulers in Figures 8, 19 and 20.

It has been shown (Hägele et al., 2024; Bergsma et al., 2025) that annealing the learning rate to smaller values than 10% of the maximum learning rate improves performance. We consider three mentioned schedulers, and report the ablation on the learning rate decay for the 210M models in Figure 9, and for the 124M models in Figure 25. In the tables that show the greed-search across hyperparameters we mention the learning rate decay factor (Final learning rate X × max LR) only for those optimizers, where we performed the corresponding ablation for. If this field is omitted from the table, we use $0.01 \times \gamma_{\max}$ for this method regardless of the learning rate scheduler applied.

## G.1 124M PARAMETERS MODEL

Below, we provide tables with complete information regarding hyperparameter tuning for 124M models including the important sweeps (weight decay, warmup, etc.) conducted for our ablations.

Table 7: **AdamW hyperparameter tuning for our 124M parameter large language models.** Bold hyperparameters are the best.

| Hyperparameter | "Small" batch setting | "Large" batch setting |
|---|---|---|
| Learning rate | $0.0001, \mathbf{0.0005}, 0.0008, 0.001, 0.002$ | $0.0001, 0.0003, 0.0005, \mathbf{0.001}, 0.002$ |
| Batch size | 32 | 256 |
| Sequence length | 512 | 512 |
| Number of warmup steps | $\mathbf{3000}, 5000, 8000$ | $500, 1000, \mathbf{2000}, 3000, 8000, 32000$ |
| Weight decay | 0.1 | no, $\mathbf{0.1}, 0.5, 0.7$ |
| Learning rate decay scheduler | WSD, **cosine** | WSD, **cosine**, linear |
| Gradient clipping | no, $\mathbf{0.5}, 1, 1.5$ | no, $\mathbf{0.5}, 1$ |
| AdamW $\beta_1$ | $0.5, \mathbf{0.8}, 0.9$ | $\mathbf{0.8}, 0.9$ |
| AdamW $\beta_2$ | $0.95, \mathbf{0.999}$ | $0.95, 0.99, \mathbf{0.999}, 0.9999$ |
| Final learning rate X × max cosine LR | — | $10^{-1}, \mathbf{10^{-2}}, 10^{-3}, 10^{-4}, 10^{-5}, 10^{-6}$ |
| Final learning rate X × max WSD LR | — | $10^{-1}, 10^{-2}, 10^{-3}, 10^{-4}, 10^{-5}, \mathbf{10^{-6}}$ |
| Final learning rate X × max linear LR | — | $10^{-1}, 10^{-2}, 10^{-3}, \mathbf{10^{-4}}, 10^{-5}, 10^{-6}$ |

Table 8: **ADOPT hyperparameter tuning for our 124M parameter large language models.** Bold hyperparameters are the best.

| Hyperparameter | "Small" batch setting | "Large" batch setting |
|---|---|---|
| Learning rate | 0.001 | $0.0001, 0.0003, 0.0005, \mathbf{0.001}, 0.002$ |
| Batch size | 32 | 256 |
| Sequence length | 512 | 512 |
| Number of warmup steps | $\mathbf{3000}, 8000$ | $\mathbf{2000}, 8000, 32000$ |
| Weight decay | 0.1 | no, $\mathbf{0.1}, 0.5$ |
| Learning rate decay scheduler | WSD, **cosine** | WSD, **cosine**, linear |
| Gradient clipping | 0.5 | no, $\mathbf{0.5}, 1$ |
| ADOPT $\beta_1$ | 0.9 | $0.8, \mathbf{0.9}$ |
| ADOPT $\beta_2$ | $\mathbf{0.999}, 0.9999$ | $0.5, \mathbf{0.999}, 0.9999$ |
| ADOPT $\varepsilon$ | $10^{-6}$ | $10^{-6}$ |

Table 9: **AdEMAMix hyperparameter tuning for our 124M parameter large language models.** Bold hyperparameters are the best.

| Hyperparameter | "Small" batch setting | "Large" batch setting |
|---|---|---|
| Learning rate | $0.0001, \mathbf{0.0005}, 0.0008, 0.001, 0.002$ | $0.0001, 0.0003, 0.0005, \mathbf{0.001}, 0.002$ |
| Batch size | 32 | 256 |
| Sequence length | 512 | 512 |
| Number of warmup steps | $\mathbf{3000}, 8000$ | $\mathbf{2000}, 8000, 32000$ |
| Weight decay | 0.1 | no, $\mathbf{0.1}, 0.5, 0.7$ |
| Learning rate decay scheduler | WSD, **cosine** | WSD, **cosine**, linear |
| Gradient clipping | no, $\mathbf{0.5}, 1, 1.5$ | no, $\mathbf{0.5}, 1$ |
| AdEMAMix $\beta_1$ | $0.5, \mathbf{0.8}, 0.9$ | $0.8, \mathbf{0.9}$ |
| AdEMAMix $\beta_2$ | 0.999 | $\mathbf{0.999}, 0.9999$ |
| AdEMAMix $\beta_3$ | $0.999, \mathbf{0.9999}, 0.99995$ | $\mathbf{0.999}, 0.9999$ |
| AdEMAMix $\alpha$ | $5, \mathbf{8}, 12$ | 8 |

Table 10: **Lion hyperparameter tuning for our 124M parameter large language models.** Bold hyperparameters are the best.

| Hyperparameter | "Small" batch setting | "Large" batch setting |
|---|---|---|
| Learning rate | $0.00005, \mathbf{0.0001}, 0.0005, 0.001$ | $0.0001, 0.0005, \mathbf{0.001}, 0.002$ |
| Batch size | 32 | 256 |
| Sequence length | 512 | 512 |
| Number of warmup steps | 3000 | $2000, 8000, \mathbf{32000}$ |
| Weight decay | $\text{no}, 0.1, 0.2, \mathbf{0.5}$ | $\text{no}, \mathbf{0.1}, 0.5, 0.7$ |
| Learning rate decay scheduler | WSD, **cosine** | WSD, cosine, **linear** |
| Gradient clipping | 0.5 | $\text{no}, \mathbf{0.5}, 1$ |
| Lion $\beta_1$ | $0.7, \mathbf{0.9}, 0.99$ | $0.5, \mathbf{0.9}$ |
| Lion $\beta_2$ | $0.9, \mathbf{0.99}, 0.999$ | $\mathbf{0.99}, 0.999$ |

Table 11: **Signum hyperparameter tuning for our 124M parameter large language models.** Bold hyperparameters are the best.

| Hyperparameter | "Small" batch setting | "Large" batch setting |
|---|---|---|
| Learning rate | $0.0003, 0.0005, \mathbf{0.001}$ | $0.0001, 0.00030.0005, 0.0003, \mathbf{0.001}, 0.002$ |
| Batch size | 32 | 256 |
| Sequence length | 512 | 512 |
| Number of warmup steps | $2000, \mathbf{3000}$ | $2000, \mathbf{8000}, 32000$ |
| Weight decay | $\text{no}, 0, \mathbf{0.1}, 0.5$ | $\text{no}, 0, \mathbf{0.1}, 0.5, 0.7$ |
| Learning rate decay scheduler | WSD, **cosine** | WSD, cosine, **linear** |
| Gradient clipping | $\text{no}, \mathbf{0.5}, 1$ | $\text{no}, \mathbf{0.5}, 1$ |
| Momentum | $\text{no}, 0.9, \mathbf{0.95}$ | $\text{no}, 0.9, \mathbf{0.95}, 0.99$ |
| Nesterov momentum | no, **yes** | no, **yes** |

Table 12: **Muon hyperparameter tuning for our 124M parameter large language models.** Bold hyperparameters are the best.

| Hyperparameter | "Small" batch setting | "Large" batch setting |
|---|---|---|
| Learning rate AdamW | $0.0001, 0.0003, 0.0005, \mathbf{0.001}, 0.002$ | $0.0001, 0.0003, 0.0005, \mathbf{0.001}, 0.002$ |
| Learning rate Muon | $0.001, \mathbf{0.01}, 0.02$ | $0.001, \mathbf{0.01}, 0.02$ |
| Batch size | 32 | 256 |
| Sequence length | 512 | 512 |
| Number of warmup steps | $\mathbf{3000}, 8000$ | $\mathbf{2000}, 8000, 32000$ |
| Weight decay | $\text{no}, \mathbf{0.1}, 0.5$ | $\text{no}, \mathbf{0.1}, 0.5$ |
| Learning rate decay scheduler | **WSD**, cosine | **WSD**, cosine, linear |
| Gradient clipping | $\text{no}, \mathbf{0.5}$ | $\text{no}, \mathbf{0.5}, 1.0$ |
| Momentum Muon | $0.9, 0.95, \mathbf{0.99}$ | $\mathbf{0.95}, 0.99$ |
| Optimizer for 1D layers | AdamW | AdamW |
| Optimizer for 1D layers, $\beta_1$ | $\mathbf{0.8}, 0.9$ | $\mathbf{0.8}, 0.9$ |
| Optimizer for 1D layers, $\beta_2$ | $0.99, \mathbf{0.999}, 0.9999$ | $0.99, \mathbf{0.999}, 0.9999$ |
| Newton-Schulz a | 3.4445 | 3.4445 |
| Newton-Schultz b | $-4.7750$ | $-4.7750$ |
| Newton-Schultz c | 2.0315 | 2.0315 |
| Newton-Schultz iterations | 5 | $1, \mathbf{5}, 10, 20$ |
| Nesterov momentum | no, **yes** | no, **yes** |

Table 13: **D-Muon hyperparameter tuning for our 124M parameter large language models.** Bold hyperparameters are the best.

| Hyperparameter | "Small" batch setting | "Large" batch setting |
|---|---|---|
| Learning rate | 0.001 | $0.0001, 0.0003, 0.0005, 0.001, \mathbf{0.002}$ |
| Batch size | 32 | 256 |
| Sequence length | 512 | 512 |
| Number of warmup steps | 3000 | $\mathbf{2000}, 8000, 32000$ |
| Weight decay | 0.1 | $\text{no}, \mathbf{0.1}, 0.5$ |
| Learning rate decay scheduler | WSD, **cosine** | WSD, **cosine**, linear |
| Gradient clipping | $\text{no}, \mathbf{0.5}$ | $\text{no}, \mathbf{0.5}, 1.0$ |
| Momentum D-Muon | 0.95 | 0.95 |
| Optimizer for 1D layers | AdamW | AdamW |
| Optimizer for 1D layers, $\beta_1$ | $\mathbf{0.8}, 0.9$ | $\mathbf{0.8}, 0.9$ |
| Optimizer for 1D layers, $\beta_2$ | $0.99, \mathbf{0.999}, 0.9999$ | $0.99, \mathbf{0.999}, 0.9999$ |
| Newton-Schulz a | 3.4445 | 3.4445 |
| Newton-Schultz b | $-4.7750$ | $-4.7750$ |
| Newton-Schultz c | 2.0315 | 2.0315 |
| Newton-Schultz iterations | 5 | 5 |
| Nesterov momentum | yes | yes |

Table 14: **SOAP hyperparameter tuning for our 124M parameter large language models.** Bold hyperparameters are the best.

| Hyperparameter | "Small" batch setting | "Large" batch setting |
|---|---|---|
| Learning rate | $0.005, \mathbf{0.001}$ | $0.0001, 0.0003, 0.0005, 0.001, \mathbf{0.002}$ |
| Batch size | 32 | 256 |
| Sequence length | 512 | 512 |
| Number of warmup steps | $\mathbf{3000}, 8000$ | $\mathbf{2000}, 4000, 8000, 12000, 16000, 32000$ |
| Weight decay | 0.1 | $\text{no}, \mathbf{0.1}, 0.5$ |
| Learning rate decay scheduler | WSD, **cosine** | WSD, **cosine**, linear |
| Gradient clipping | 0.5 | $\text{no}, \mathbf{0.5}, 1$ |
| Preconditioner dimension | 10000 | 10000 |
| Preconditioning frequency | $1, 5, \mathbf{10}$ | $1, 5, \mathbf{10}$ |
| SOAP $\beta_1$ | $0.8, \mathbf{0.9}$ | $0.8, \mathbf{0.9}, 0.95$ |
| SOAP $\beta_2$ | $0.95, 0.99, \mathbf{0.999}, 0.9999$ | $0.95, 0.99, \mathbf{0.999}, 0.9999$ |

Table 15: **Sophia hyperparameter tuning for our 124M parameter large language models.** Bold hyperparameters are the best.

| Hyperparameter | "Small" batch setting | "Large" batch setting |
|---|---|---|
| Learning rate | $0.0001, \mathbf{0.0003}, 0.0005, 0.001, 0.002$ | $0.0001, 0.0003, 0.0005, \mathbf{0.001}, 0.002, 0.01$ |
| Batch size | 32 | 256 |
| Sequence length | 512 | 512 |
| Number of warmup steps | $\mathbf{2000}, 3000$ | $2000, 8000, \mathbf{32000}$ |
| Weight decay | 0.1 | $\text{no}, \mathbf{0.1}, 0.5$ |
| Learning rate decay scheduler | WSD, **cosine** | WSD, **cosine**, linear |
| Gradient clipping | 0.5 | $\text{no}, \mathbf{0.5}, 1$ |
| Estimator | Gauss-Newton-Bartlett | Gauss-Newton-Bartlett |
| Estimator frequency | 10 | 10 |
| Sophia $\beta_1$ | 0.9 | $0.8, \mathbf{0.9}$ |
| Sophia $\beta_2$ | $0.95, \mathbf{0.999}, 0.9999, 0.99999$ | $0.95, \mathbf{0.999}, 0.9999, 0.99999$ |
| Sophia $\rho$ | $0, 0.03, \mathbf{0.04}$ | $0, 0.03, \mathbf{0.04}$ |

Table 16: **Schedule-Free AdamW hyperparameter tuning for our 124M parameter large language models.** Bold hyperparameters are the best.

| Hyperparameter | "Small" batch setting | "Large" batch setting |
|---|---|---|
| Learning rate | $0.0001, 0.0003, 0.0005, \mathbf{0.001}, 0.005$ | $0.0001, 0.0003, 0.0005, 0.001, \mathbf{0.002}, 0.005$ |
| Batch size | 32 | 256 |
| Sequence length | 512 | 512 |
| Number of warmup steps | $\mathbf{3000}, 8000$ | $2000, 4000, \mathbf{8000}, 12000, 16000, 32000$ |
| Weight decay | no, $0.05, \mathbf{0.1}, 0.5$ | no, $0.05, \mathbf{0.1}, 0.5$ |
| Learning rate decay scheduler | no | no |
| Gradient clipping | no, $\mathbf{0.5}$ | no, $\mathbf{0.5}, 1$ |
| Schedule-Free AdamW $\beta_1$ | $\mathbf{0.9}, 0.95, 0.98$ | $\mathbf{0.9}, 0.95, 0.98$ |
| Schedule-Free AdamW $\beta_2$ | $0.95, 0.99, 0.999, \mathbf{0.9999}, 0.99999$ | $0.95, 0.99, 0.999, \mathbf{0.9999}, 0.99999$ |

Table 17: **Prodigy hyperparameter tuning for our 124M parameter large language models.** Bold hyperparameters are the best.

| Hyperparameter | "Small" batch setting | "Large" batch setting |
|---|---|---|
| Learning rate | $0.5, \mathbf{1}$ | $0.5, \mathbf{1}, 2, 10, 100$ |
| Batch size | 32 | 256 |
| Sequence length | 512 | 512 |
| Number of warmup steps | $\mathbf{3000}, 8000$ | $\mathbf{2000}, 4000, 8000, 12000, 16000, 32000$ |
| Weight decay | no, $\mathbf{0.1}, 0.5$ | no, $\mathbf{0.1}, 0.5$ |
| Learning rate decay scheduler | no, WSD, **cosine** | no, WSD, **cosine**, linear |
| Gradient clipping | no, $\mathbf{0.5}, 1$ | no, $\mathbf{0.5}, 1$ |
| Prodigy $\beta_1$ | $0.9$ | $0.8, \mathbf{0.9}$ |
| Prodigy $\beta_2$ | $0.99, \mathbf{0.999}, 0.9999$ | $\mathbf{0.999}, 0.9999$ |
| Prodigy bias correction | no, **yes** | no, **yes** |

Table 18: **MARS (MARS-AdamW) hyperparameter tuning for our 124M parameter large language models.** Bold hyperparameters are the best.

| Hyperparameter | "Small" batch setting | "Large" batch setting |
|---|---|---|
| Learning rate AdamW | $0.0001, 0.0005, \mathbf{0.001}, 0.003$ | $0.0001, 0.0005, \mathbf{0.001}, 0.003$ |
| Learning rate MARS | $0.001, \mathbf{0.003}$ | $0.001, \mathbf{0.003}$ |
| Batch size | 32 | 256 |
| Sequence length | 512 | 512 |
| Number of warmup steps | $2000, \mathbf{3000}$ | $\mathbf{2000}, 8000, 32000$ |
| Weight decay MARS | no, $\mathbf{0.1}$ | no, $\mathbf{0.1}, 0.5$ |
| Weight decay for 1D layers | 0.1 | 0.1 |
| Learning rate decay scheduler | WSD, **cosine** | WSD, **cosine**, linear |
| Gradient clipping | 0.5 | 0.5 |
| Optimizer for 1D layers | AdamW | AdamW |
| Optimizer for 1D layers $\beta_1$ | $\mathbf{0.8}, 0.9$ | $\mathbf{0.8}, 0.9, 0.95$ |
| Optimizer for 1D layers $\beta_2$ | $0.95, 0.99, \mathbf{0.999}$ | $0.95, 0.99, \mathbf{0.999}$ |
| MARS $\beta_1$ | $0.9, \mathbf{0.95}$ | $0.9, \mathbf{0.95}$ |
| MARS $\beta_2$ | $0.95, \mathbf{0.99}$ | $0.95, \mathbf{0.99}$ |
| VR scaling factor $\eta$ | $0.023, 0.024, \mathbf{0.025}$ | $0.023, 0.024, \mathbf{0.025}$ |

Table 19: **MARS-Lion hyperparameter tuning for our 124M parameter large language models.** Bold hyperparameters are the best.

| Hyperparameter | "Small" batch setting | "Large" batch setting |
|---|---|---|
| Learning rate Lion | **0.0001**, 0.0005, 0.001, 0.003 | **0.0001**, 0.0005, 0.001, 0.003 |
| Learning rate MARS | **0.0001**, 0.001, 0.003 | **0.0001**, 0.001, 0.003 |
| Batch size | 32 | 256 |
| Sequence length | 512 | 512 |
| Number of warmup steps | 2000, **3000** | **2000**, 8000, 32000 |
| Weight decay MARS | no, **0.1** | no, **0.1**, 0.5 |
| Weight decay for 1D layers | 0.1 | 0.1 |
| Learning rate decay scheduler | **WSD**, cosine | **WSD**, cosine |
| Gradient clipping | 0.5 | 0.5 |
| Optimizer for 1D layers | Lion | Lion |
| Optimizer for 1D layers $\beta_1$ | 0.8, **0.9** | 0.8, **0.9**, 0.95 |
| Optimizer for 1D layers $\beta_2$ | 0.95, **0.99**, 0.999 | 0.95, **0.99**, 0.999 |
| MARS $\beta_1$ | 0.9, **0.95** | 0.9, **0.95** |
| MARS $\beta_2$ | 0.95, **0.99** | 0.95, **0.99** |
| VR scaling factor $\eta$ | 0.024, **0.025** | 0.024, **0.025** |

Table 20: **MARS-Shampoo hyperparameter tuning for our 124M parameter large language models.** Bold hyperparameters are the best.

| Hyperparameter | "Small" batch setting | "Large" batch setting |
|---|---|---|
| Learning rate Shampoo | 0.0001, 0.0005, **0.001**, 0.003 | 0.0001, 0.0005, **0.001**, 0.003 |
| Learning rate MARS | 0.001, **0.003** | 0.001, **0.003** |
| Batch size | 32 | 256 |
| Sequence length | 512 | 512 |
| Number of warmup steps | 2000, **3000** | **2000**, 8000, 32000 |
| Weight decay MARS | no, **0.1** | no, **0.1**, 0.5 |
| Weight decay for 1D layers | 0.1 | 0.1 |
| Learning rate decay scheduler | WSD, **cosine** | WSD, **cosine** |
| Gradient clipping | 0.5 | 0.5 |
| Optimizer for 1D layers | Shampoo | Shampoo |
| Optimizer for 1D layers $\beta_1$ | 0.8, **0.9** | 0.8, **0.9**, 0.95 |
| Optimizer for 1D layers $\beta_2$ | 0.95, 0.99, **0.999** | 0.95, 0.99, **0.999** |
| MARS $\beta_1$ | 0.9, **0.95** | 0.9, **0.95** |
| MARS $\beta_2$ | 0.95, **0.99** | 0.95, **0.99** |
| VR scaling factor $\eta$ | 0.024, **0.025** | 0.024, **0.025** |

## G.2 210M PARAMETERS MODEL

For 210M models we perform training runs only with the batch size of $256 \times 512$ tokens, utilizing the same training durations as for 124M model with this batch size, i.e., $\{8\mathbf{k}, 16\mathbf{k}, 32\mathbf{k}, 48\mathbf{k}, 64\mathbf{k}, 128\mathbf{k}\}$, which corresponds to the following counts in tokens: $\{1\mathbf{B}, 2.1\mathbf{B}, 4.2\mathbf{B}, 6.3\mathbf{B}, 8.4\mathbf{B}, 16.8\mathbf{B}\}$.

We also replicate almost identical hyperparameters to those of the training of the 124M model to verify whether the smooth transition Takeaway 3 in the final ranking of optimizers and their sensitivity to hyperparameters will be observed.

Table 21: **AdamW hyperparameter tuning for our 210M parameter large language models.** Bold hyperparameters are the best.

| Hyperparameter | "Large" batch setting |
|---|---|
| Learning rate | 0.001 |
| Batch size | 256 |
| Sequence length | 512 |
| Number of warmup steps | $50, 500, 1000, \mathbf{2000}$ |
| Weight decay | no, $\mathbf{0.1}$ |
| Learning rate decay scheduler | WSD, **cosine**, linear |
| Gradient clipping | 0.5 |
| AdamW $\beta_1$ | $0.8, \mathbf{0.9}$ |
| AdamW $\beta_2$ | $0.95, 0.99, \mathbf{0.999}, 0.9999$ |
| Final learning rate X $\times$ max cosine LR | $10^{-1}, \mathbf{10^{-2}}, 10^{-3}, 10^{-4}, 10^{-5}, 10^{-6}$ |
| Final learning rate X $\times$ max WSD LR | $10^{-1}, \mathbf{10^{-2}}, 10^{-3}, 10^{-4}, 10^{-5}, 10^{-6}$ |
| Final learning rate X $\times$ max linear LR | $10^{-1}, 10^{-2}, \mathbf{10^{-3}}, 10^{-4}, 10^{-5}, 10^{-6}$ |

Table 22: **ADOPT hyperparameter tuning for our 210M parameter large language models.** Bold hyperparameters are the best.

| Hyperparameter | "Large" batch setting |
|---|---|
| Learning rate | 0.001 |
| Batch size | 256 |
| Sequence length | 512 |
| Number of warmup steps | 2000 |
| Weight decay | 0.1 |
| Learning rate decay scheduler | cosine |
| Gradient clipping | no, $\mathbf{0.5}, 1$ |
| ADOPT $\beta_1$ | 0.9 |
| ADOPT $\beta_2$ | $0.5, \mathbf{0.999}, 0.9999$ |
| ADOPT $\varepsilon$ | $10^{-3}, 10^{-4}, 10^{-5}, \mathbf{10^{-6}}, 10^{-7}, 10^{-8}, 10^{-9}, 10^{-10},$ |

Table 23: **AdEMAMix hyperparameter tuning for our 210M parameter large language models.** Bold hyperparameters are the best.

| Hyperparameter | "Large" batch setting |
|---|---|
| Learning rate | 0.001 |
| Batch size | 256 |
| Sequence length | 512 |
| Number of warmup steps | 2000 |
| Weight decay | 0.1 |
| Learning rate decay scheduler | cosine |
| Gradient clipping | 0.5 |
| AdEMAMix $\beta_1$ | 0.9 |
| AdEMAMix $\beta_2$ | 0.999 |
| AdEMAMix $\beta_3$ | 0.999 |
| AdEMAMix $\alpha$ | 8 |

Table 24: **Lion hyperparameter tuning for our 210M parameter large language models.** Bold hyperparameters are the best.

| Hyperparameter | "Large" batch setting |
|---|---|
| Learning rate | $0.0001, \mathbf{0.0005}, 0.001$ |
| Batch size | 256 |
| Sequence length | 512 |
| Number of warmup steps | 2000 |
| Weight decay | 0.1 |
| Learning rate decay scheduler | cosine |
| Gradient clipping | 0.5 |
| Lion $\beta_1$ | 0.9 |
| Lion $\beta_2$ | 0.99 |

Table 25: **Signum hyperparameter tuning for our 210M parameter large language models.** Bold hyperparameters are the best.

| Hyperparameter | "Large" batch setting |
|---|---|
| Learning rate | $0.0001, \mathbf{0.0005}, 0.001$ |
| Batch size | 256 |
| Sequence length | 512 |
| Number of warmup steps | 2000 |
| Weight decay | 0.1 |
| Learning rate decay scheduler | cosine |
| Gradient clipping | 0.5 |
| Momentum | $0.9, \mathbf{0.95}, 0.99$ |
| Nesterov momentum | yes |

Table 26: **Muon hyperparameter tuning for our 210M parameter large language models.** Bold hyperparameters are the best.

| Hyperparameter | "Large" batch setting |
|---|---|
| Learning rate AdamW | 0.001 |
| Learning rate Muon | 0.01 |
| Batch size | 256 |
| Sequence length | 512 |
| Number of warmup steps | 2000 |
| Weight decay | 0.1 |
| Learning rate decay scheduler | cosine |
| Gradient clipping | 0.5 |
| Momentum Muon | 0.95 |
| Optimizer for 1D layers | AdamW |
| Optimizer for 1D layers, $\beta_1$ | 0.8 |
| Optimizer for 1D layers, $\beta_2$ | 0.999 |
| Newton-Schulz a | 3.4445 |
| Newton-Schultz b | $-4.7750$ |
| Newton-Schultz c | 2.0315 |
| Nesterov momentum | yes |

Table 27: **D-Muon hyperparameter tuning for our 210M parameter large language models.** Bold hyperparameters are the best.

| Hyperparameter | "Large" batch setting |
|---|---|
| Learning rate | 0.001 |
| Batch size | 256 |
| Sequence length | 512 |
| Number of warmup steps | 2000 |
| Weight decay | 0.1 |
| Learning rate decay scheduler | cosine |
| Gradient clipping | 0.5 |
| Momentum D-Muon | 0.95 |
| Optimizer for 1D layers | AdamW |
| Optimizer for 1D layers, $\beta_1$ | **0.8**, 0.9 |
| Optimizer for 1D layers, $\beta_2$ | 0.99, **0.999** |
| Newton-Schulz a | 3.4445 |
| Newton-Schultz b | $-4.7750$ |
| Newton-Schultz c | 2.0315 |
| Newton-Schultz iterations | 5 |
| Nesterov momentum | yes |

Table 28: **SOAP hyperparameter tuning for our 210M parameter large language models.** Bold hyperparameters are the best.

| Hyperparameter | "Large" batch setting |
|---|---|
| Learning rate | 0.001 |
| Batch size | 256 |
| Sequence length | 512 |
| Number of warmup steps | 2000 |
| Weight decay | 0.1 |
| Learning rate decay scheduler | cosine |
| Gradient clipping | 0.5 |
| Preconditioner dimension | 10000 |
| Preconditioning frequency | 10 |
| SOAP $\beta_1$ | 0.9 |
| SOAP $\beta_2$ | 0.999 |

Table 29: **Sophia hyperparameter tuning for our 210M parameter large language models.** Bold hyperparameters are the best.

| Hyperparameter | "Large" batch setting |
|---|---|
| Learning rate | 0.001 |
| Batch size | 256 |
| Sequence length | 512 |
| Number of warmup steps | 2000 |
| Weight decay | 0.1 |
| Learning rate decay scheduler | cosine |
| Gradient clipping | 0.5 |
| Estimator | Gauss-Newton-Bartlett |
| Estimator frequency | 10 |
| Sophia $\beta_1$ | 0.9 |
| Sophia $\beta_2$ | 0.999 |
| Sophia $\rho$ | 0.04 |

Table 30: **Schedule-Free AdamW hyperparameter tuning for our 210M parameter large language models.** Bold hyperparameters are the best.

| Hyperparameter | "Large" batch setting |
|---|---|
| Learning rate | 0.001 |
| Batch size | 256 |
| Sequence length | 512 |
| Number of warmup steps | 2000, **8000** |
| Weight decay | 0.1 |
| Learning rate decay scheduler | no |
| Gradient clipping | 0.5 |
| Schedule-Free AdamW $\beta_1$ | 0.9 |
| Schedule-Free AdamW $\beta_2$ | 0.9999 |

Table 31: **Prodigy hyperparameter tuning for our 210M parameter large language models.** Bold hyperparameters are the best.

| Hyperparameter | "Large" batch setting |
|---|---|
| Learning rate | 1 |
| Batch size | 256 |
| Sequence length | 512 |
| Number of warmup steps | 2000 |
| Weight decay | 0.1 |
| Learning rate decay scheduler | cosine |
| Gradient clipping | 0.5 |
| Prodigy $\beta_1$ | 0.9 |
| Prodigy $\beta_2$ | 0.999 |
| Prodigy bias correction | yes |

Table 32: **MARS (MARS-AdamW) hyperparameter tuning for our 210M parameter large language models.** Bold hyperparameters are the best.

| Hyperparameter | "Large" batch setting |
|---|---|
| Learning rate AdamW | 0.001 |
| Learning rate MARS | 0.003 |
| Batch size | 256 |
| Sequence length | 512 |
| Number of warmup steps | 2000 |
| Weight decay MARS | 0.1 |
| Weight decay for 1D layers | 0.1 |
| Learning rate decay scheduler | cosine |
| Gradient clipping | 0.5 |
| Optimizer for 1D layers | AdamW |
| Optimizer for 1D layers $\beta_1$ | **0.8**, 0.9 |
| Optimizer for 1D layers $\beta_2$ | 0.999 |
| MARS $\beta_1$ | 0.95 |
| MARS $\beta_2$ | 0.99 |
| VR scaling factor $\eta$ | 0.024, **0.025** |

### G.3 583M PARAMETERS MODEL

For models of 583M scale, we ablate the difference between our setup and the one from Vyas et al. (2024). The main changes compared to our setup include: learning rate decay down to $10\%$ of the maximum, usage of $z$-loss regularizer in addition to the cross-entropy loss, smaller decoupled weight decay of $0.0001$. We also point out that SOAP performance in (Vyas et al., 2024) was measured on the Chinchilla optimal number of tokens and with 2M tokens batch size. Thus, in Section 4.3 we ablate the differences between our settings on the same training horizons. A complete list of hyperparameters used for our AdamW and SOAP models in this ablations are presented in Table 33.

Table 33: **AdamW hyperparameter tuning for our 583M parameter large language models.** Bold hyperparameters are the best.

| Hyperparameter | 2M batch setting |
|---|---|
| Learning rate | 0.001, **0.005** |
| Batch size | 3936 |
| Sequence length | 512 |
| Number of warmup steps | 1200 |
| Weight decay | 0.0001, **0.1** |
| Learning rate decay scheduler | cosine |
| Gradient clipping | 0.5 |
| AdamW $\beta_1$ | 0.9, **0.95** |
| AdamW $\beta_2$ | **0.95**, 0.99 |
| Final learning rate X $\times$ max cosine LR | $10^{-1}$, $\mathbf{10^{-2}}$ |
| $z$-loss regularization | **no**, 0.0001 |

Table 34: **SOAP hyperparameter tuning for our 583M parameter large language models.** Bold hyperparameters are the best.

| Hyperparameter | 2M batch setting |
|---|---|
| Learning rate | 0.001, **0.005** |
| Batch size | 3936 |
| Sequence length | 512 |
| Number of warmup steps | 1200 |
| Weight decay | 0.0001, **0.1** |
| Learning rate decay scheduler | cosine |
| Gradient clipping | 0.5 |
| Preconditioner dimension | 10000 |
| Preconditioning frequency | 10 |
| SOAP $\beta_1$ | 0.9, **0.95** |
| SOAP $\beta_2$ | **0.95**, 0.99, 0.999 |
| Final learning rate X $\times$ max cosine LR | $10^{-1}$, $\mathbf{10^{-2}}$ |
| $z$-loss regularization | **no**, 0.0001 |

### G.4  720M PARAMETERS MODEL

In this section, we provide a complete information about the hyperparameter search for the largest models used in our benchmarking experiments. Deriving insights from our ablations (Figures 13, 30 and 31) on the smaller scale, we suggest to re-tune beta parameters of optimizers as changing the training iterations—see Takeaways 9 and 17 for this conclusions.

Tables below cover our tuning outcomes for all methods. We highlight that, when training with large batches of 1M tokens, we use the smaller number of iterations for our runs: $T \in \{8, 16, 48\}\mathbf{k(B)}$ steps (tokens)—see Table 6. Thus, according to Takeaway 9, we find that smaller $\beta_2$ parameter gives better results for SOAP, D-Muon (for 1D parameters), and Prodigy.

Table 35: **AdamW hyperparameter tuning for our 720M parameter large language models.** Bold hyperparameters are the best.

| Hyperparameter | 1M batch setting |
|---|---|
| Learning rate | $0.0001, 0.0003, 0.0005, \mathbf{0.001}$ |
| Batch size | 1984 |
| Sequence length | 512 |
| Number of warmup steps | 2000 |
| Weight decay | 0.1 |
| Learning rate decay scheduler | cosine |
| Gradient clipping | $\mathbf{0.1}, 0.5$ |
| AdamW $\beta_1$ | $0.8, \mathbf{0.9}, 0.95$ |
| AdamW $\beta_2$ | $0.95, 0.99, \mathbf{0.999}$ |

Table 36: **ADOPT hyperparameter tuning for our 720M parameter large language models.** Bold hyperparameters are the best.

| Hyperparameter | 1M batch setting |
|---|---|
| Learning rate | 0.001 |
| Batch size | 1984 |
| Sequence length | 512 |
| Number of warmup steps | 2000 |
| Weight decay | 0.1 |
| Learning rate decay scheduler | cosine |
| Gradient clipping | 0.1 |
| ADOPT $\beta_1$ | $0.9, \mathbf{0.95}$ |
| ADOPT $\beta_2$ | $0.95, 0.99, \mathbf{0.999}$ |
| ADOPT $\varepsilon$ | $10^{-6}$ |

Table 37: **AdEMAMix hyperparameter tuning for our 720M parameter large language models.** Bold hyperparameters are the best.

| Hyperparameter | 1M batch setting |
|---|---|
| Learning rate | $\mathbf{0.001}, 0.002$ |
| Batch size | 1984 |
| Sequence length | 512 |
| Number of warmup steps | 2000 |
| Weight decay | 0.1 |
| Learning rate decay scheduler | cosine |
| Gradient clipping | 0.1 |
| AdEMAMix $\beta_1$ | 0.9 |
| AdEMAMix $\beta_2$ | $0.95, \mathbf{0.999}$ |
| AdEMAMix $\beta_3$ | $\mathbf{0.999}, 0.9999$ |
| AdEMAMix $\alpha$ | 8 |

### G.5 520M PARAMETERS MoE MODEL

We extend our comparison of optimizers beyond dense models to include Mixture of Experts (MoE) architectures. Starting from our Llama-like transformer with tied embeddings, we construct an MoE variant following the Switch-Transformer implementation (Fedus et al., 2022). The model employs classical linear gating with softmax and top-$k$ routing ($k = 2$) over 8 experts. We retain the SwiGLU activation functions (Shazeer, 2020), RMSNorm layers (Zhang & Sennrich, 2019), and RoPE embeddings (Su et al., 2023) exactly as in our dense LLMs. Keeping the same hidden size, number of layers, and attention heads as the 124M dense model, this results in a $\sim 520$M parameter MoE architecture. A detailed specification of this model is provided in Table 47.

Table 38: **Lion hyperparameter tuning for our 720M parameter large language models.** Bold hyperparameters are the best.

| Hyperparameter | 1M **batch setting** |
|---|---|
| Learning rate | $0.00005, 0.0001, \mathbf{0.0002}, 0.0003, 0.0005, 0.001$ |
| Batch size | 1984 |
| Sequence length | 512 |
| Number of warmup steps | 2000 |
| Weight decay | 0.1 |
| Learning rate decay scheduler | cosine |
| Gradient clipping | $0.1, \mathbf{1}$ |
| Lion $\beta_1$ | 0.9 |
| Lion $\beta_2$ | 0.99 |

Table 39: **Signum hyperparameter tuning for our 720M parameter large language models.** Bold hyperparameters are the best.

| Hyperparameter | 1M **batch setting** |
|---|---|
| Learning rate | $0.0001, \mathbf{0.0002}, 0.0003, 0.0005, 0.001$ |
| Batch size | 1984 |
| Sequence length | 512 |
| Number of warmup steps | 2000 |
| Weight decay | 0.1 |
| Learning rate decay scheduler | cosine |
| Gradient clipping | $\mathbf{0.1}, 1$ |
| Momentum | $0.9, \mathbf{0.95}, 0.99$ |
| Nesterov momentum | yes |

Table 40: **Muon hyperparameter tuning for our 720M parameter large language models.** Bold hyperparameters are the best.

| Hyperparameter | 1M **batch setting** |
|---|---|
| Learning rate AdamW | $0.0005, \mathbf{0.001}, 0.002$ |
| Learning rate Muon | 0.01 |
| Batch size | 1984 |
| Sequence length | 512 |
| Number of warmup steps | 2000 |
| Weight decay | 0.1 |
| Learning rate decay scheduler | cosine |
| Gradient clipping | 0.1 |
| Momentum Muon | 0.95 |
| Optimizer for 1D layers | AdamW |
| Optimizer for 1D layers, $\beta_1$ | $\mathbf{0.8}, 0.9, 0.95$ |
| Optimizer for 1D layers, $\beta_2$ | $0.95, 0.99, \mathbf{0.999}$ |
| Newton-Schulz a | 3.4445 |
| Newton-Schultz b | $-4.7750$ |
| Newton-Schultz c | 2.0315 |
| Nesterov momentum | yes |

For training, we use a batch size of $256 \times 512$. Optimizer hyperparameters are taken directly from Appendix G.2, with one adjustment: the learning rate for Sophia is set to $0.0005$ instead of $0.001$. The purpose of this ablation is to evaluate how optimizers, tuned on dense models, perform when directly transferred to MoE models. In practical scenarios, practitioners often reuse well-established hyperparameters tuned on dense LLMs; hence, we argue that our comparison on the 520M MoE model reflects realistic small-scale deployment settings.

We report our configurations for training runs in Table 48.

Table 41: **D-Muon hyperparameter tuning for our 720M parameter large language models.** Bold hyperparameters are the best.

| Hyperparameter | 1M batch setting |
|---|---|
| Learning rate | $0.0005, \mathbf{0.001}, 0.002, 0.003, 0.005$ |
| Batch size | 1984 |
| Sequence length | 512 |
| Number of warmup steps | 2000 |
| Weight decay | 0.1 |
| Learning rate decay scheduler | cosine |
| Gradient clipping | 0.1 |
| Momentum D-Muon | 0.95 |
| Optimizer for 1D layers | AdamW |
| Optimizer for 1D layers, $\beta_1$ | $0.8, \mathbf{0.9}, 0.95$ |
| Optimizer for 1D layers, $\beta_2$ | $0.95, \mathbf{0.99}, 0.999$ |
| Newton-Schulz a | 3.4445 |
| Newton-Schultz b | $-4.7750$ |
| Newton-Schultz c | 2.0315 |
| Newton-Schultz iterations | 5 |
| Nesterov momentum | yes |

Table 42: **SOAP hyperparameter tuning for our 720M parameter large language models.** Bold hyperparameters are the best.

| Hyperparameter | 1M batch setting |
|---|---|
| Learning rate | 0.001 |
| Batch size | 1984 |
| Sequence length | 512 |
| Number of warmup steps | 2000 |
| Weight decay | 0.1 |
| Learning rate decay scheduler | cosine |
| Gradient clipping | 0.1 |
| Preconditioner dimension | 10000 |
| Preconditioning frequency | 10 |
| SOAP $\beta_1$ | $0.9, \mathbf{0.95}$ |
| SOAP $\beta_2$ | $\mathbf{0.95}, 0.99, 0.999$ |

Table 43: **Sophia hyperparameter tuning for our 720M parameter large language models.** Bold hyperparameters are the best.

| Hyperparameter | 1M batch setting |
|---|---|
| Learning rate | $0.0001, \mathbf{0.0005}, 0.001$ |
| Batch size | 1984 |
| Sequence length | 512 |
| Number of warmup steps | 2000 |
| Weight decay | 0.1 |
| Learning rate decay scheduler | cosine |
| Gradient clipping | 0.1 |
| Estimator | Gauss-Newton-Bartlett |
| Estimator frequency | 10 |
| Sophia $\beta_1$ | $0.9, \mathbf{0.95}$ |
| Sophia $\beta_2$ | $0.95, 0.99, \mathbf{0.999}$ |
| Sophia $\rho$ | 0.04 |

Table 44: **Schedule-Free AdamW hyperparameter tuning for our 720M parameter large language models.** Bold hyperparameters are the best.

| Hyperparameter | 1M batch setting |
|---|---|
| Learning rate | 0.001 |
| Batch size | 1984 |
| Sequence length | 512 |
| Number of warmup steps | 2000, **8000** |
| Weight decay | 0.1 |
| Learning rate decay scheduler | no |
| Gradient clipping | no, **0.1** |
| Schedule-Free AdamW $\beta_1$ | **0.9**, 0.95 |
| Schedule-Free AdamW $\beta_2$ | 0.95, 0.99, 0.999, **0.9999** |

Table 45: **Prodigy hyperparameter tuning for our 720M parameter large language models.** Bold hyperparameters are the best.

| Hyperparameter | 1M batch setting |
|---|---|
| Learning rate | 0.5, **1**, 2 |
| Batch size | 1984 |
| Sequence length | 512 |
| Number of warmup steps | 2000 |
| Weight decay | 0.1 |
| Learning rate decay scheduler | cosine |
| Gradient clipping | 0.1 |
| Prodigy $\beta_1$ | 0.9, **0.95** |
| Prodigy $\beta_2$ | 0.95, **0.99**, 0.999 |
| Prodigy bias correction | yes |

Table 46: **MARS (MARS-AdamW) hyperparameter tuning for our 720M parameter large language models.** Bold hyperparameters are the best.

| Hyperparameter | 1M batch setting |
|---|---|
| Learning rate AdamW | 0.001 |
| Learning rate MARS | 0.003 |
| Batch size | 1984 |
| Sequence length | 512 |
| Number of warmup steps | 2000 |
| Weight decay MARS | 0.1 |
| Weight decay for 1D layers | 0.1 |
| Learning rate decay scheduler | cosine |
| Gradient clipping | 0.1 |
| Optimizer for 1D layers | AdamW |
| Optimizer for 1D layers $\beta_1$ | **0.8**, 0.9, 0.95 |
| Optimizer for 1D layers $\beta_2$ | 0.95, 0.99, **0.999** |
| MARS $\beta_1$ | 0.95 |
| MARS $\beta_2$ | 0.99 |
| VR scaling factor $\eta$ | 0.024, **0.025** |

Table 47: **Configurations for our Llama-based MoE model.**

| # Parameters | 520M |
|---|---|
| Hidden size | 768 |
| # Attention heads | 12 |
| # Layers | 12 |
| Init. std | 0.02 |
| Use bias | no |
| RMSNorm epsilon | 0.00001 |
| Positional encoding | RoPE |
| MoE router loss | load balancing loss (Fedus et al., 2022) (Eq. 4) & router $z$-loss (Zoph et al., 2022) (Eq. 5) |
| # Experts per layer | 8 |
| # Shared experts | 0 |
| Top-$k$ routing ($k$) | 2 |
| MoE softmax order | top-$k \rightarrow$ softmax |

Table 48: **Lengths of training for the MoE model in "Large" batch size setting ($256 \times 512$).**

| # Parameters | Tokens (Iterations) | | Chinchilla Tokens |
|---|---|---|---|
| 520M | 5.5B (42k) | 44B (336k) | 10.4B |

