# OpenReview forum: "Benchmarking Optimizers for Large Language Model Pretraining"
_ICLR.cc/2026/Conference — Submitted to ICLR 2026_

### Official Review · Reviewer_Sd77 · 2025-10-16

**Soundness:** 2
**Presentation:** 3
**Contribution:** 3
**Rating:** 4
**Confidence:** 3

**Summary:**

This paper points out that during the standard pre-training phase of Large Language Models (LLMs), a comprehensive evaluation of recent optimization techniques is conducted. Specifically, the paper systematically varies model scale, batch size, and training duration, and through careful tuning, provides appropriate optimizer selections for various scenarios.

**Strengths:**

1. This paper is well-written, with clear logic and concise readability.

2. It conducts extensive experiments, including comparative experiments on models of different types and scales.

**Weaknesses:**

1. First, the paper claims to provide the most suitable optimizer selections for various scenarios. However, as is widely known, different optimizers not only differ in performance but also have significant disparities in computational resources and time required. Conducting a detailed comparison of the optimizer selection schemes provided in the paper from multiple perspectives would enhance the paper’s comprehensiveness.

2. Second, the paper uses relatively small-scale models for experiments. The scale of existing Large Language Models (LLMs) is all larger than that of the models used in the paper’s experiments. This increases the uncertainty of the paper’s conclusions in practical applications, making a broader range of experimental comparisons necessary.

3. Finally, from the perspective of experimental results, the optimizer selection schemes proposed in the paper are appropriate under the current experimental settings. Nevertheless, these selection schemes are heuristic and lack theoretical guidance or selection principles—this fails to address the paper’s core question: "What are the key factors for selecting optimizers in different scenarios?" More descriptions should be added regarding future work or limitations.

**Questions:**

See Weaknesses.

---

> ### Author Response · Authors · 2025-11-21
>
> We thank the Reviewer for the time, comments and your expertise. Below, we answer mentioned weaknesses.
>
> > different optimizers not only differ in performance but also have significant disparities in computational resources and time required
>
> We do agree on that point. That’s why in the initial version of our manuscript we also ablated wall-clock time performance (Figs. 42, 43, 44) and reported the consumed resources (see Lines 210 and 1838). For some optimisers, we studied the particular hyperparameters that might affect the runtime or its stability, e.g., the number of Newton-Schulz iterations for Muon (Fig. 32), effective LR of Prodigy (Fig. 35), why Sophia diverges (Figs. 26 and 27) and more.
>
> > the paper uses relatively small-scale models for experiments
>
> We do not use large, production-scale models, which we, of course, state as our limitation (see Lines 1171-1172). However, we think that even small-scale experimental results (when models ranging from nanoGPT-like 124M to near 1B size) are very relevant for the community. The ability to look and play with hands-on level in a controlled benchmark is meaningful. Additionally, some aspects of the pretraining best practices we revisit in this work are, perhaps, scale-independent. For example, the importance of large weight decay in very short runs and the importance of increased beta parameters (mainly $\beta_2$), when the number of iterations is large. And, while training paradigms for large models tend to be more conservative in some aspects (for example, the learning rate decaying below 10% of the maximum is a well-adopted technique across models of practical scales), it is always beneficial to keep in mind the small-scale ablations.
>
> > "What are the key factors for selecting optimizers in different scenarios?"
>
> We do provide some guidance on the hyperparameter and optimizer selection for different training needs. For example, in very short, almost speed run regimes, we suggest trying to increase not only the learning rate but also the weight decay. Cruelly, weight decay needs to be applied to all parameters for matrix-based optimizers such as Muon; otherwise, it would cause a huge gap in performance. Perhaps surprisingly, Lion and other sign-based optimizers prefer longer warmup; our Fig. 7 even demonstrates the results where Lion with a super long warmup of up to 25% of training outperforms the AdamW baseline with a moderate warm-up — in our opinion, this is a very overlooked phenomenon that also should be taken into consideration. Other phenomena, described in the “Our advice on tuning each method” paragraph, are also valuable and are still a hot topic for discussions, especially everything related to the learning rate decaying and questions whether or not cosine decay down to zero can harm the SFT. As a choice for an optimizer, we encourage the community to try AdEMAMix, Muon, SOAP / MARS — strong baselines in every setting we have tried. Additionally, the Prodigy optimizer might be a good option for tuning learning rates of Adam-like methods as we describe in more detail in Fig. 35.
>
> We believe we addressed Reviewer’s concerns. We stay at their disposal to answer any further questions.

---

### Official Review · Reviewer_yoQR · 2025-10-24

**Soundness:** 2
**Presentation:** 2
**Contribution:** 2
**Rating:** 2
**Confidence:** 4

**Summary:**

This paper presents a benchmark of 11 optimization methods for LLM pre-training across varying model sizes, batch sizes, and training durations, challenging AdamW's long-standing dominance in the field. The study reveals that newer optimizers like AdEMAMix and MARS can significantly outperform AdamW, particularly at larger scales, while batch size critically affects optimizer performance. The research provides insights for practitioners regarding learning rate schedules, momentum parameters, and weight decay that substantially impact both absolute performance and relative optimizer rankings.

**Strengths:**

- Developing a better understanding of optimizer dynamics beyond the go-to AdamW optimizer is important for building more efficient training pipelines
- I do appreciate the overview of recent developments in the optimizer field. This can be valuable for practitioners in building more effective training pipelines.

**Weaknesses:**

- I am generally missing deep insights. All highlights in the paper appear to be trivial to come up with by systematically comparing the optimizers with one another. I would have hoped to read more about _why_ certain optimizers exhibit their unique dynamics rather than just a a vs b comparison (e.g., Takeaway 1: ... sign-based methods (Signum, Lion), and MARS greatly benefit from the increased batch size; (III) Sophia diverges in the small-batch setting, when trained beyond the Chinchilla optimal horizon...). What are the general insights that will help a practitioner or researcher build better (more sample-efficient) training routines? Also, some of the Takeaways are well known and not new (e.g., Takeaway 5).
- I am missing the "so what?" The paper ends with results and does not provide a discussion of what might be the best optimizer going forward (incl. hyperparameter ranges that work well in a wide number of settings) or even edge cases where certain optimizers fail. Also, in the end the optimizer performance depends on the underlying data. FineWeb is considered to be a curated pre-training dataset. Exploring the performance under data heterogeneity would yield more meaningful insights, I believe.
- I do see a major issue with the chosen model sizes. Typically, ablation studies use models that are twice or even 10x the size of what is being discussed in the paper (e.g., GneissWeb uses 1B and 7B models to explore data dynamics, https://arxiv.org/pdf/2502.14907).

**Formatting concerns**
- The paper appears to make too much use of vertical spaces. Some figures overlap the text, making some section a bit difficult to read (e.g., l. 294)

**Questions:**

- How do you make sure you are actually providing an apples-to-apples comparison when you need to tune multiple hyperparameters at once (e.g., line 266f.) where you cannot be sure how the changes individually affect the optimizer dynamics?
- How do the optimizers behave with noisy data or with datasets that contain samples with very little relevant information?
- How do the computational cost of all the optimizers compare?

---

> ### Author Response · Authors · 2025-11-21
>
> We thank the Reviewer for their expertise, time taken to review, and for the raised questions. Below, we answer your concerns
>
> > I am generally missing deep insight
>
> We do provide takeaways throughout the text, both in the main part and in the Appendix. Overall, we highlight 23 takeaways of different content. Each figure with ablation is clearly described. Some takeaways are indeed well-known; however, we do not think that this is the case for takeaway 5. Some takeaways need to be revisited also. For example, we do think that takeaways 2, 5, 7, 8, 9, 10, 11, 16, 20 are highly relevant, in general. And they are not affected by the final optimizer ranking in any case.
>
> > "so what?"
>
> Due to the space limits, we left the Discussion section, finalizing the paper, in Appendix. This section briefly highlights the most important ablations. The Discussion is provided on page 22, along with the limitations. It was added since the first edition of the PDF
>
>
> **Q1**
>
> Once the batch size setting is fixed, we, throughout almost every ablation, fixed all hyperparameters except for one and did a sweep across this hyperparameter, starting with the learning rate as the most important one. In Section 4.1 we report interesting patterns found across optimisers, dissecting how one particular hyperparameter affects training. We believe this is a thorough approach given the fact that we are unable to tune everything simultaneously. Also, for all model scales, we tune hyperparameters at near Chinchilla optimal horizons and then extrapolate. Not surprisingly, after this approach some hyperparameters remain untuned, that’s why we additionally performed a sweep for the long 16.8B and 33.6B token runs for 124M models.
> For our larger models, we also re-tuned some hyperparameters (mostly LR and betas) for the longest training runs.
> The only major experiment where we sweep many hyperparameters at once is the comparison with the setting of the SOAP paper. In this ablation, our goal was to compare their hyperparameters against ours in the setting where they trained SOAP. It occurred that our choice for the hyperparameters not only improved their results, but also swapped the ranking of optimisers — Takeaway 5. In general, we do not claim that  the hyperparameters found are the best, we just ensure that the overall setting for tuning is self-consistent and no key hyperparameter is omitted.
>
> **Q2** and `…optimizer performance depends on the underlying data` and `….data dynamics`
>
> Thanks for pointing us towards this issue. We would like to clarify that our setup is positioned as controlled and standard, for ease of use. Therefore, we did not pursue a goal of comparing optimisers on different data mixtures. Though they all work, we have tried running only on the FineWeb, OWT2, and the FineWeb-Edu. Separately, we performed some ablations on C4 as well. All of the mentioned corpora are curated; therefore, we did not study the essentially noisy data. Other corpora for LLM pretraining, especially at scale, are also curated in some way, and recent findings suggest popular optimisers perform on different datasets with close relative performance [1, 2]. Generally, investigating optimisers’ behaviour under different data mixes is another challenging task. However, it is worth mentioning in the manuscript that such a topic needs further investigation.
>
> **Q3**
>
> We present the wall-clock time ablation in Figs. 42, 43, 44. Additionally, we mention the compute spent for all our experiments (see Lines 210 and 1838) and the number of total runs. Generally, we find SOAP being the slowest optimizer in terms of wall-clock time. However, in terms of the final val loss reached, SOAP performs consistently well (around top 3-4 on average) in different settings.
>
> [1] "Apertus: Democratizing Open and Compliant LLMs for Global Language Environments"
>
> [2] "Muon is Scalable for LLM Training"
>
> We hope we addressed Reviewer's concerns. We hope the Reviewer will consider increasing their score, and we stay at their disposal to answer any further questions.

---

> > ### Comment · Reviewer_yoQR · 2025-11-24
> >
> > Thank you for your concise rebuttal. Regarding Q3, I agree that there are hints on optimizers working reasonably well with noisy data but we still see notable performance improvements when doing token or sample level reweighing or selection [1, 2]. This is indicative of a relevant sensitivity nonetheless. I think, the benchmark would really benefit from insights related to noise in data. It would also benefit the line of argumentation in the paper if there was a thorough discussion of insights along with a set of guiding principles for overall more effective training pipelines. I maintain my rating.
> >
> > Sources
> > [1] Rho-1: Not All Tokens Are What You Need, Lin et al., 2024
> > [2] Dynamic Loss-Based Sample Reweighting for Improved Large Language Model Pretraining, Sow et al., 2025

---

### Official Review · Reviewer_Tb1Y · 2025-10-29

**Soundness:** 3
**Presentation:** 3
**Contribution:** 3
**Rating:** 6
**Confidence:** 3

**Summary:**

The paper provides a comprehensive evaluation of 12 optimizers for LLM pretraining. With systematic experiments on varying model size, model type (dense/MoE), training duration, batch size, etc, the paper reports rankings across regimes, and releases code/configs to encourage reproducibility. Comprehensive ablation on hyperparameters provides practical insights.

**Strengths:**

1. The topic is highly relevant and significant to LLM pretraining. While AdamW is the de facto optimizer, numerous new optimizers have been proposed, claiming to outperform AdamW. It is worthwhile to conduct controlled benchmark experiments to evaluate these optimizers and compare their performance.

2. Experiments seem comprehensive and rigorous. It covers a wide range of optimizers, model size, batch size, training duration, and other configurations. The experiments are standardized, and hyperparameters are carefully tuned for each optimizer.

3. Releasing the full benchmarking toolkit greatly enhances reproducibility and facilitates future research.

**Weaknesses:**

1. The experiments are limited in scale. The largest model in experiments is only 720M, while concurrent work [1] scales up to 1.2B. This could affect the results and claims, as larger models can be less sensitive to optimizers.

2. The evaluation metric is limited to validation loss. Downstream performances like MMLU and GLUE are not included.

3. The presentation is a bit crowded for now (due to page limit).


[1] Wen, Kaiyue, et al. "Fantastic pretraining optimizers and where to find them." arXiv preprint arXiv:2509.02046 (2025).

**Questions:**

1. In your experiments, MARS and AdEMAMix are promising. However, in [1], MARS performs worse than Muon and SOAP. What could be the reason for this difference?

2. Have you considered performing even limited downstream evaluations (e.g., on GLUE, MMLU) for models trained with the top-performing optimizers vs. AdamW to confirm if the validation loss improvements translate to task performance gains in your setting?

[1] Wen, Kaiyue, et al. "Fantastic pretraining optimizers and where to find them." arXiv preprint arXiv:2509.02046 (2025).

---

> ### Author Response · Authors · 2025-11-21
>
> We thank the Reviewer for their expertise, time taken to review, and for the positive feedback. Below, we proceed to the mentioned questions and weaknesses.
>
> > The evaluation metric is limited to validation loss.
>
> We explicitly state this as a limitation of ours in the corresponding section in the **Limitations** paragraph in the **Discussion** section.
> Additionally, it is common in many to analyse training dynamics relying mostly on loss and other training-related metrics, leaving the downstream evals aside, and prior research works argue that loss might be a good proxy for downstream scores [1, 2, 3]. Of course, the contrary examples exist. However, when models are small enough, e.g., smaller than 3B, even trained for a sufficient amount of tokens, they can barely answer simple MMLU tasks [4]. Additionally, small models (around 1B-1.5B) need special preparation for multiple-choice questions, e.g., removing (A, B, C, …) from the problem description.
>
> > The presentation is a bit crowded for now
>
> With an additional page included, we will accommodate the Discussion section into the text, and organize a bit more space for ablations.
>
> **Q1**
>
> In our opinion, the difference is in the batch size — both in the global BS and in the sequence length. We use a much smaller sequence length (512 vs their 4096), but they do have a slightly larger batch size, allowing them to train with larger learning rates (e.g., following the sqrt{2} scaling rule for the LR when doubling BS). Thus, we suspect that Muon with a large enough learning rate can outperform MARS in a similar setting. While the LR is definitely the main reason, other hyperparameters also matter. We think that our choice of betas for MARS is slightly better. The authors of the mentioned paper do not use AdEMAMix, so we  would be curious to try and see if it can outperform Muon in their setup as well.
>
> **Q2**
>
> We did a separate runs with AdEMAMix and AdamW at 1.5B scale with evals, however, at this scale of models downstream evals are not so meaningful metric due to the reason from the previous questions.
>
> [1] “Establishing task scaling laws via compute-efficient model ladders”, 2024
>
> [2] “Understanding emergent abilities of language models from the loss perspective”, 2025
>
> [3] “Language models scale reliably with over-training and on downstream tasks”, 2024
>
> [4] Phi-1.5 technical report, 2023
>
>
> We stay here at your disposal and are happy to answer any further questions.

---

### Official Review · Reviewer_mntc · 2025-10-29

**Soundness:** 2
**Presentation:** 3
**Contribution:** 3
**Rating:** 4
**Confidence:** 4

**Summary:**

This paper conducted a systematic benchmark of optimizers for LLM pretraining, evaluating over many choices (e.g. AdamW, Muon, D-Muon, AdEMAMix, SOAP, MARS, etc) across multiple model sizes and architectures, ranging from 124 M to 720 M parameters. For each optimizer and model, the author finetune the hyperparameters, schedulers, erc based on the scaling law, and rank optimizers based on the final validation ppl using FineWeb dataset.

**Strengths:**

This paper conduct systematic evaluation of the optimizer's performance with finetuned hyperparameters. There approach seems to be solid (**under their setup**). But I have concerns regarding their setup, and I will elaborate about this later. From these benchmarking, it reveals several interesting findings for practitioners, e.g. AdamW can catchup in the end when iterations is longer. There rankings and empirical findings are useful for the community.

**Weaknesses:**

I have several concerns regarding the setups.

First, the choice of context length (as a benchmarking paper) is not practical. $512$ is too short for any production-level LLM, even in the pretraining phrase. Changing the context length can have effects on the conclusions. In practice, $1K$ to $4K$ or even $8K$ context sizes are often adopted.

Second, the training iterations (or equivalently the token batch size) is too long. This may also alter the conclusions. Note that training with longer iterations and compare the checkpoints is **NOT** the same as training directly with short iterations due to the lr scheduling and varied batch sizes. For example, for 124M model, you uses upto $1024$K iterations to finish 16.8B token. However, even for industrial-level model training, it does not use that many iterations. For example, [1] trains a MoE model with roughly 35K steps with 1.2T data. In this setup, Muon does show a clear advantages over AdamW.

Third, the current setup does not consider severe over-train regime. For 124M model, only $\approx 6.7$x overtrain is considered. It would be good to test one sever over-train regime (like 10x?) with relatively short training iterations (within 10K steps).

Last but not least, I wonder is it possible that some optimizers just use a larger effective learning rate? Therefore, without fine-grained tuning of lr, we may have different conclusions. One possible solution is to align their RMS norm. For an optimizer output $\Delta W$, we can post-process it to be $\frac{\Delta W}{||\Delta W||} * \rho$, where $\rho$ is some target norm. This eliminates the different effective learning rate. Will this change the claims?

**Questions:**

My questions are in the weakness section.

---

> ### Author Response · Authors · 2025-11-21
>
> We are grateful for the time that the Reviewer invested in our work and for the positive feedback. Below we answer the Reviewer’s question.
>
> > training iterations (or equivalently the token batch size) is too long
>
> We train with a cosine scheduler for a few fixed training horizons; depending on the batch size, it results in a different number of iterations.
> For the very small BS (~16.4K tokens), we train 124M models for 1024K steps; for, e.g., a BS of 260K tokens, we train shorter.
> I would not say that this number of iterations is too long. Taking into account the tokens-per-parameter ratio, we train up to ~6.7x Chinchilla horizons. Production models are being trained for dozens or hundreds of Chinchillas [1, 3, 4].
>
> > the current setup does not consider severe over-train regime …. For 124M model, only x overtrain is considered
>
> From the context, it seems to me that by “x overtrain” you mean the Chinchilla training length in tokens. If yes, then for 124M models we do consider overtrain up to 33.6B tokens, which is roughly ~13.6x Chinchilla horizons. In this regard, we already include the results for more than “10x overtrain”. Please refer to Figures 4 and 13.
>
> > is it possible that some optimizers just use a larger effective learning rate?
>
> Yes, matrix-based optimizers such as SOAP and (D-)Muon allow the use of larger learning rates than AdamW and sign-based methods. From our observations, the effective LR of D-Muon is roughly twice larger than that of AdamW. Notably, for Signum, Lion, and Sophia, the effective LR is even smaller than that for AdamW. We connect this to the fact that the $\mathtt{sign}(\cdot)$ operator is sensitive to gradient noise.
>
> > One possible solution is to align their RMS norm
>
> Thanks for the suggestion! I see that some formulae did not type well, but we got your idea. We did not try to align RMS of all optimizers, however, we find this approach interesting and admit that this is one of the key improvements of the D-Muon over Muon. However, we think that with hand-tuned LR one can still get better performance, but the idea of matching RMS for all optimizers sounds promising. The question here is which RMS (if not AdamW’s) one would like to preserve.
>
> We believe we addressed Reviewer’s concerns. We stay at their disposal to answer any further questions.

---

### Meta-Review · Area_Chair_WVbB · 2026-01-07

**Summary:**

The paper presents an empirical study on the effect of different optimizers on LLM training, which is a good starting point for understanding the training dynamics of LLMs. The reviewers have concerns about the lack of practicality and insufficient in-depth discussions. Although a number of takeaway observations have been presented, a discussion on **why** such observations are seen is desirable but largely missing. As such, this paper is not yet ready for publication. For future submissions, the authors are suggested to reorganize the paper and experiment design, by focusing each section on the problem being studied and potential root causes of observations.

**Reviewer Concerns:**

Some concerns have been addressed, but some major issues related to practicality and in-depth discussion have not been addressed.

**Reviewer Scores:**

I'm not sure whether the reviewers will change their scores.

---

### Decision · Program_Chairs · 2026-01-26

Reject